

# A 20-year record (1998-2017) of permafrost, active layer, and meteorological conditions at a High Arctic permafrost research site (Bayelva, Spitsbergen): an opportunity to validate remote sensing data and land surface, snow, and

**permafrost models**

**Julia Boike[1,2], Inge Juszak[1], Stephan Lange[1], Sarah Chadburn[3,4], Eleanor Burke[5], Pier Paul Overduin[1], Kurt Roth[6], Olaf Ippisch[7], Niko Bornemann[1], Lielle Stern[6], Isabelle Gouttevin[8,9], Ernst Hauber[10], Sebastian Westermann[11]**

[1] Alfred Wegener Institute Helmholtz Center for Polar and Marine Research, Telegrafenberg A43, 14473 Potsdam, Germany

[2] Geography Department Humboldt-University, Unter den Linden 6, 10099 Berlin, Germany

[3] University of Leeds, School of Earth and Environment, Leeds LS2 9JT, U.K.

[4] University of Exeter, College of Engineering, Mathematics and Physical sciences, Exeter EX4 4QF, U.K.

[5] Met Office Hadley Centre, FitzRoy Road, Exeter EX1 3PB, U.K.

[6] Institute of Environmental Physics, INF 229, University of Heidelberg, 69120 Heidelberg, Germany

[7] Institute for Mathematics, Erzstr. 1,Erzstr. 1, 38678 Clausthal-Zellerfeld, Germany



[8] Irstea, UR HHLY, centre de Lyon-Villeurbanne, 5 rue de la Doua, BP 32108, 69616 Villeurbanne Cedex, France

[9] Université Grenoble Alpes, Irstea, UR ETGR, Centre de Grenoble, 2 rue de la Papeterie-BP 76, 38402 St-Martin-d'Hères, France

[10] Institute of Planetary Research, German Aerospace Center (DLR), Rutherfordstr. 2, 12489 Berlin, Germany

[11] Department of Geosciences, University of Oslo, P.O. Box 1047, Blindern, 0316 Oslo, Norway

Correspondence to: Julia Boike (Julia.Boike@awi.de)





**Abstract**

Most permafrost is located in the Arctic, where frozen organic carbon makes it an important component of the global climate system. Despite the fact that the Arctic climate changes more rapidly than the rest of the globe, observational data density in the region is low. Permafrost

thaw and carbon release to the atmosphere are a positive feedback mechanism that can exacerbate climate warming. This positive feedback functions via changing land-atmosphere energy and mass exchanges. There is thus a great need to understand links between the energy balance, which can vary rapidly over hourly to annual time scales, and permafrost, which changes slowly over long time periods. This understanding thus mandates long-term observa-

tional data sets.

Such a data set is available from the Bayelva Site at Ny-Ålesund, Svalbard, where meteorology, energy balance components and subsurface observations have been made for the last 20 years. Additional data include a high resolution digital elevation model and a panchromatic image. This paper presents the data set produced so far, explains instrumentation, calibration,

processing and data quality control, as well as the sources for various resulting data sets. The resulting data set is unique in the Arctic and serves a baseline for future studies. Since the data provide observations of temporally variable parameters that mitigate energy fluxes between permafrost and atmosphere, such as snow depth and soil moisture content, they are suitable for use in integrating, calibrating and testing permafrost as a component in Earth System

Models. The data set also includes a high resolution digital elevation model that can be used together with the snow physical information for snow pack modeling.

The presented data are available in the supplementary material for this paper and through the PANGAEA website (https://doi.pangaea.de/10.1594/PANGAEA.880120).



## 1  Introduction

Permafrost, which is defined as ground that has remained frozen continuously for two years or
more, covers large parts of the land surface in the northern hemisphere, amounting to about 15
million km$^2$ (Brown et al., 1998). The range in temperatures and water and ice content of the
upper surface layer of seasonally freezing and thawing ground (the "active layer") determine
the biological and hydrological processes that operate in these areas. Thermal degradation of
permafrost over the last few decades has been reported from many circum-Arctic boreholes
(Romanovsky et al., 2010). Warming and thawing of permafrost and an overall reduction in
the area that it covers under future climate change scenarios have been predicted in all recent
climate models, but at widely varying rates (Koven et al., 2012). Continued observations, not
only of permafrost thermal data but also of the multiple other types of data required to
understand the changes to permafrost, are therefore of great importance. The data required
includes information on the upper boundary condition of the soil (specifically on snow cover),
on atmospheric conditions, and on various subsurface state variables (for example, volumetric
liquid water content and temperature). The seasonal snow cover in Arctic permafrost regions
can blanket the permafrost surface for many months of the year and has an important effect on
the thermal regime of permafrost (Langer et al., 2013; López-Moreno et al., 2016). The soil's
water content determines not only its hydrologic and thermal properties, but also the amount
of latent heat that is either required for the seasonal thaw in spring, or produced during fall. In
view of these dependencies, the resulting datasets (including snow cover and the thermal state
of the soil and permafrost), ideally at the same resolution as any meteorological input data,
will be of great value for evaluating permafrost models (or land surface models intended for

permafrost regions). In this paper we present data that incorporate subsurface components of heat and mass flux, properties of the snow cover, and weather data from the Bayelva High Arctic permafrost site.

The Bayelva research site on Spitsbergen Island in the Svalbard archipelago (78.551° N; 11.571° E) has been investigated by the AWI in collaboration with academic partners since 1998, with the original objective of developing and testing permafrost process models (Ippisch, 2001; Stern, 2017). Major developments in earth system models, for example

through the European PAGE21 project (http://www.page21.org/), Permafrost Carbon Network projects (http://www.permafrostcarbon.org), and satellite calibration and validation (cal/val) missions, have subsequently led to sustained interest in the data produced from a wider modeling community. This publication provides information on the site and a full documentation of the data set collected between 1998 and 2017 that may be required for input

into earth system models (Chadburn et al., 2015; Ekici et al., 2014; Ekici et al., 2015).

## 2   Site description

The Bayelva site is located in the European High Arctic, on western Spitsbergen in the Svalbard archipelago. It is situated about 3 km from the small village of Ny-Ålesund, which

serves as a permanent hub for researchers and their logistical support. Ny-Ålesund was a coal mining area from 1916 to 1962, since then the village has been gradually transformed into a research community. Long term meteorological monitoring was initiated in 1969 by the Norwegian Sverdrup Station but research topics now also include terrestrial ecosystems, glacier monitoring, (high Level) atmospheric research, and the Kongsfjorden ocean system,

coordinated by the Ny-Ålesund Science Managers Committee (http://nysmac.npolar.no/).



The West Spitsbergen Ocean Current, a branch of the North Atlantic Current, warms this area to an average air temperature of about −13°C in January and +5°C in July. It also provides about 400 mm of precipitation annually, which falls mostly as snow between September and May. Significant warming of air temperatures has been detected since 1960, which has generally been attributed to changes in the radiation budget and atmospheric circulation patterns (Førland et al., 2012; Hanssen-Bauer and Førland, 1998). This warming is also reflected in the permafrost temperatures, as recorded from deep boreholes (up to 102 m deep) in the mountains at Janssonhaugen (Isaksen et al., 2007a; Isaksen et al., 2001; Isaksen et al., 2007b). Continuous permafrost underlies the un-glaciated coastal areas to a depth of about 100 m and the active layer thickness at the end of summer ranges between 1 and 2 m (Humlum, 2005). The region has experienced increases in cloudiness, precipitation, and the number and intensity of cyclones in recent years, especially during the winter months (Hanssen-Bauer and Førland, 1998; Sepp and Jaagus, 2011). The increase in cloudiness (Maturilli and Kayser, 2016) has led to an increase in incoming long-wavelength radiation, resulting in a major change to the winter radiation budget for this region, as measured at the German-French (AWIPEV) research station, which is located within the village of Ny-Ålesund (Maturilli et al., 2014). This research station carries out long term monitoring of radiation (Baseline Surface Radiation Network: http://bsrn.awi.de/) and meteorological data (http://www.awipev.eu/).

The data presented herein were collected from the High Arctic Bayelva River catchment area (Figure 1), away from Ny-Ålesund. A legacy from past mining activities is the physical disturbance of the ground in and around Ny-Ålesund (for example, compaction and reworking of the soil). Traffic in the village (people, cars, snow mobiles) also affects the surface





conditions, especially in winter. The Bayelva catchment area lies between two mountains
(Zeppelinfjellet and Scheteligfjellet), with the glacial Bayelva River originating from the two
branches of the Brøggerbreen glacier. The terrain flattens out to the north of the Bayelva site
and the Bayelva River flows into the Kongsfjorden fjord and the Arctic Ocean about 1 km
from the site (Figure 1). Over the past three decades the Bayelva catchment area has been the
focus of intensive investigations into fluvial hydrology (sediment transfer and geochemistry;
(Hodson et al., 2002)), soil and permafrost conditions (Boike et al., 2008b; Roth and Boike,
2001; Westermann, 2010; Westermann et al., 2011b), the surface energy balance (Boike et al.,
2003b; Westermann et al., 2009), and the micrometeorological processes controlling the
surface gas and energy exchanges (Lloyd, 2001b; Lloyd et al., 2001; Lüers et al., 2014). The
data from this Bayelva site have also been used in earth system modeling (Ekici et al., 2014;
Ekici et al., 2015). Nearby investigations by Japanese and Italian researchers include
vegetation analysis with respect to periglacial and glacial landforms and topography (Cannone
et al., 2004; Ohtsuka et al., 2006), and investigations into the plant and plot scale dependence
of $CO_2$ emissions on biotic and abiotic factors at the start and end of the growing season
(Cannone et al., 2016; Uchida et al., 2006).

The Bayelva site is located on top of the Leirhaugen hill (25 m a.s.l.), on permafrost patterned
ground (Figure 2). The hill consists mainly of rock but is partly covered by a mixture of
sediments that consist of glacial till, together with fine grained glacio-fluvial sediments and
clays from the last glacial advance. The gray color of these sediments suggests that they
derived from the Kongsfjorden glacier and not the adjacent Brøggerbreen glacier, whose
sediments have a more reddish color. Since the hill has a maximum altitude of about 38 m,

which is below the maximum height for marine sedimentation, these sediments may also include marine deposits.

The vegetation cover in the vicinity of the Bayelva site (i.e. the area of the hill) has been estimated to be approximately 50-60 %, with the remainder being bare soil with a small proportion of stones (cobbles and gravel) (Lloyd et al., 2001). Within the rest of the Bayelva River catchment area, sparse vegetation alternates with bare soil and sand, or with rock fields.

The area of the Bayelva site is covered with non-sorted circles, also known as mudboils (Figure 2), which are only present on the hill itself, with other patterned ground phenomena (such as sorted circles and stripes) occurring on and in the area surrounding Leirhaugen hill. These non-sorted circles formed under localised favourable conditions following the last glacial period. The bare soil centers of these circles are about 1 m in diameter, surrounded by a vegetated rim that consists of a mixture of low vascular plants (including various species of grass and sedge such as Carex spp., Deschampsia spp., Eriophorum spp., Festuca spp., and Luzula spp.), catchfly, saxifrage, willow, various other locally common species (Dryas octopetala, Oxyria digyna, Polygonum viviparum), and some unclassified species of moss and lichen (Ohtsuka et al., 2006; Uchida et al., 2006).

The soils on the hill generally range from "silty loam" to silty clay" with a few large stones. The silt content decreases from more than 50 % at the top of the soil profile to less than 30 % at the bottom of the soil profile at about 1 meter depth, while the clay content increases to more than 50 %. Concentrations of total organic carbon, total nitrogen, and total sulphur are highest at the bottom of the soil profile, peaking below the centers of the non-sorted circles. The organic carbon concentration is high (> 6 % weight) at the bottom of the soil profile. Nitrogen is elevated beneath the vegetated rim of the non-sorted circles (Boike et al., 2008a).

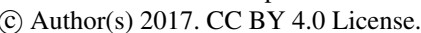

The Bayelva site can be temporarily shaded from direct solar radiation by the mountains on

170    either side. Snow is present for up to 9 months of the year and strong winds can encourage

drifting and snow redistribution. The depth of the snow cover varies within the Bayelva

catchment area, as do its physical characteristics, such as the snow-water-equivalent and the

internal stratigraphy and density of the snow pack. A 15 m x 10 m fence surrounds the

Bayelva site to prevent reindeer from damaging the equipment, and a small wooden container

175    (about 0.90 m x 1.40 m, and 2.3 m high) houses the data acquisition and transfer system

(Figure 2). Warm weather events during the winter can interrupt the build-up of snow cover

resulting in the formation of either internal ice lenses or basal ice, depending on the heat

content of the snow pack. Table 1 summarizes the characteristics of the site retrieved from

both previous publications and data included in this paper.

180

### 3    Data description

The Bayelva site was powered by a solar panel and wind generator during the period from

1998 to 2013 and data retrieved manually through site visits every 4 to 6 weeks, when visual

inspections were made of the sensors.

Since 2013 the site has been connected to the mains electricity supply and the Ny-Ålesund

data network, and data has since then been transferred automatically resulting in much

improved data collection with almost no data gaps. The data gaps prior to 2013 resulted

mainly from problems with the energy supply for the site, such as problems with the

solar/wind charge regulator. The current online data transfer and real time data visualization

enables real time data checking and identification of possible sensor failure, as well as remote

programming. Major problems that have affected the continuous data collection or the data

quality include reindeer disturbance (e.g. reindeer pulling cables out of the ground, or destroying sensors by rubbing against the stations, as occurred in 2003) and the gradual uplift of sensors as a result of freeze-thaw processes, which affected soil temperature measurements.

Details of the sensors are provided in the following sections, as well as descriptions of the data quality and cleaning routine (Section 4). The instruments are divided into above-ground (meteorological) and below-ground sensors (for example, soil sensors). Further detailed information on the sensors can be found in Table 2, which summarizes all of the parameters and instruments (Table 2), as well as in Appendix C and Appendix D (metadata and

description of instruments). Figure 3 presents time series of parameters between 1998 and 2017.

### 3.1  Weather station data

Meteorological data have been recorded at the AWIPEV Research Base in Ny-Ålesund since

1992, including radiation component data for the worldwide Baseline Surface Radiation Network (BSRN). The AWIPEV meteorological data are described by (Maturilli et al., 2013) and can be downloaded from PANGAEA (http://bsrn.awi.de/data/data-retrieval-via-pangaea/). Climate records covering a longer period of time (since 1989) are available from the airport in Longyearbyen, which is about 110 km from Ny-Ålesund (Nordli et al., 2014).

Since the Bayelva site is located close to the leading edge of the Brøggerbreen glacier, a different local micro-climate can be expected from Ny-Ålesund. A climate station was therefore installed in 1998, within the fenced area. The temporal resolution of the meteorological measurements was increased in October 2009, from 60 minutes to 30 minutes.



### 3.1.1 Air temperature, relative humidity, and snow temperature

Air temperature and relative humidity (hourly averages, since 2009 half hourly averages)
are measured at a height of 2 m above the ground surface using PT100 platinum resistance
temperature sensors and Rotronic MP103A/Vaisala HMP45 capacity sensors, protected by
unventilated shields. The heights of the shields are not adjusted during periods of snow cover
accumulation or ablation. The uncertainty in the temperature measurements ranges between
0.03 and 0.5 °C, depending on the sensors used; the uncertainty in the relative humidity
measurements ranges between 2 and 3%. The PT00 temperature sensors were calibrated in
ice-slush water for at least 12 hours to determine the absolute accuracy at 0° C. The precision
of the temperature sensors was calculated from the variation in values recorded during the
calibration period, at constant temperatures in ice-slush water, and can be found in Table 2.

The change of sensors for relative humidity in 2009 resulted in 7 percentage points lower
average relative humidity between 2009 and 2017 as compared to the period before,
independent of the season.

*Snow temperature*

PT100 temperatures sensors were installed at the weather station in 1998, close to the ground
surface. When covered by snow these sensors record the snow temperature, while during the
snow-free period they record (unshielded) air temperatures. The temperature cables were orig-
inally attached to a small (vertical) bamboo stick, with sensors sticking out a few cm from the
stick (Appendix C, fig. C5). Visual inspections during late spring (May 2016) revealed that
melting occurred around the bamboo stick, leaving an air gap around the stick. The tempera-
ture sensors were nevertheless well away from the stick (in a horizontal direction) and re-



mained embedded in the snow. The height of the sensors was changed on a number of occasions following damage by reindeer. Sensors were originally mounted at heights of 0.35 and 0.48 m to record snow temperatures, but these heights were changed during subsequent years

(Table 2). Information concerning the changes in sensor height is included in the data description that accompanies the data (Appendix C, fig. C5).

### 3.1.2  Wind speed and direction

The wind speed and direction were measured with a R.M. Young propeller anemometer

(Model 05103). The wind direction, its standard deviation was recorded at hourly (1998-2009) or half-hourly (2009-present) intervals.

### 3.1.3  Radiation

Net radiation was first measured in 1998 using a CSI (Campbell Scientific Ltd.) Q7 net radi-

ometer, which was replaced in 2000, reinstalled between May 2002-Sept 2003 with an NR Lite net radiometer (Kipp & Zonen) and in 2009 with a NR01 four-component sensor (Hukseflux). A wind correction was performed according to the manufacturer's suggestion for the Q7 and NRLite net radiometers (Campbell Scientific Ltd.; details in Appendix C & D). An SP 1110 (Skye Instruments) pyranometer was also installed in 1998 and a CG1 (Kipp and

Zonen, Appendix C3) downward-looking pyrgeometer (to detect reflected longwave radiation) in 2003. Regular site visits were made during which the pyranometers were checked for condensation, dirt, physical damage, or hoar frost. Nevertheless, because the instruments were argely unattended most of the time, we estimate the field accuracy to have been between

±10 % (for the Q7 sensor) and ±20 % (for the NR Lite). The Q7 sensor was destroyed by

reindeer in September 2003 and was not replaced. Details on the various net radiometer set

ups and correction are provided in Appendix D.

### 3.1.4 Rainfall

An un-heated and un-shielded tipping bucket rain gauge (R M Young, 52203) was installed

on a pole in August 1999, with the top of the bucket about 1.6 m above ground Level. The

instrument measures only liquid precipitation (rainfall) and not winter snowfall. The tipping

bucket was regularly checked in summer by pouring a known volume of water into the bucket

and by frequent visual inspections for dirt or snow during each site visit. At wind speeds

above 12.5 m/s (hourly or half hourly average) the rain gauge was found to record

precipitation even when there was none, due to vigorous shaking of the pole; these data were

flagged accordingly within the data series. Until 2010, precipitation was only recorded as

hourly totals, but the provided data set uses 30 minute temporal resolution (half hours are

presented as "NaN", since the hourly total is given at full hour). Total daily precipitation

figures from a gauge in Ny-Ålesund are available through the The Norwegian Meteorological

Institute (formerly: www.eklim.no, https://www.met.en/).

### 3.1.5 Snow depth

The snow depth around the station has been continuously monitored since 1998 with an SR50

sonic ranging sensor (Campbell Scientific Ltd.) and with an additional SR50 sensor at a

nearby (about 85 m; Figure 2) eddy station since 2006. The sensor recorded raw distance data



from the sensor to the object (in this case the ground or snow surface). The data obtained from the sonic sensor was corrected using the speed of sound at 0°C and the air temperature measured at the Bayelva site, using the formula provided by the manufacturer (Campbell Scientific Ltd.; Appendix D, D4). To obtain the snow depth the distance of the sensor from

the ground surface was recorded annually and subtracted from the corrected distance data. Snow depths are provided in the data supplement for this paper. Due to seasonal thawing of the ground, the surface can subside over the season (Overduin and Kane, 2006), thus negative distance rates of a few cms are computed (and not set to zero). The data are not removed from the series since the opposite process, frost heave, takes place during fall freeze back and thus

the surface bounces back to the original height. Since the ground surface has little or no vegetation cover the signal is returned with a high degree of accuracy (Table 2). An additional snow depth laser distance sensor (Jenoptik SHM30) was installed at the Bayelva site climate tower in August 2013. The data from the three different snow depth sensors was combined to obtain a continuous record and fill any gaps left by the failure of individual instruments

("Level 2 data" containing all original snow data from the three sites "merged" into one data product, can be found in Section 4). It should be noted that two of the sensors (an SR50 sensor and the laser sensor) were located within 5 m of each other, while the second SR50 (at the eddy covariance site) was located 85 m away, downhill from the Bayelva site (Figure 2). Any differences in snow depth between these sensors may therefore be due to

microtopographic and microclimatic variations between these two locations.

### 3.1.6  Time lapse photography of snow cover

In order to monitor the timing and pattern of snow melt an automated camera system (K1; UIT) was set up in August 2013 to photograph the land surface in those areas in which the instruments were located (Figure C2). The camera recorded one image every hour; during the polar night a light was switched on automatically while each image was being recorded.

### 3.1.7  Dielectric number of snow

The dielectric number of the snow was measured using a vertically installed time-domain reflectometry (TDR) probe. The length of the TDR probe was 0.5 m between 1998 and 2007 and 0.33 m from 2009 onwards. Dielectric numbers are provided in the data set for both the snow-covered and the snow-free periods (recording only air dielectric numbers). When the snow cover depth exceeds the length of the TDR probe the dielectric number data can be used to infer dry snow densities using empirical relationships (Schneebeli et al., 1998). In wet snow, the measured snow dielectric number can be used for the calculation of the snow liquid water content using the composite approach with ice added as an additional phase (for example, Roth et al., 1990) and measured snow densities (Appendix E, table. E1).

### 3.1.8  Vertical profiling of the physical properties of snow

Snow depth, together with vertical profiles for temperature, density, and dielectric numbers were collected from the site and surrounding areas during a number of end-of-snow (spring) seasons. These profiles, which were collected at irregular intervals between 2000 and 2016, provide additional information on the snowpack. They typically reveal the presence of a number of ice layers (including a basal ice layer) and snow densities of between 220 and 440 kg/m$^3$. The typical end-of-spring average snow density before snow melts is 350 kg/m$^3$. The

end of spring season snow-water-equivalent varies between the years due to variations in

snow depth (Figure 4). Details of these profiles, including some stratigraphic information, are

provided in Appendix E (Table E1).

### 3.2 Subsurface data on permafrost and the active layer

Data have been collected from a variety of installations that were changed over time because

of the deterioration of sensors or disturbance by reindeer. As a result, there are a number of

data series from different profiles rather than one continuous data series. Some measurements

were also discontinued (for example, soil heat flux).

### 3.2.1 Instrument installation and soil sampling

In order to take into account any possible effects of small variations in vegetation and

microtopography at the site (e.g. due to the presence of non-sorted circles), instruments were

installed in a number of different positions within a profile. Deterioration of data quality and

the failure of some sensors also led to the installation of new sensors in new profiles.

*Instrument installation and soil sampling 1998*

The first set of instruments was installed in 1998, within one of the non-sorted circles

(Appendix C, Figure C7), for which a trench 2 m wide and 1.4 m deep was excavated across

the circular feature (Appendix C, Figure C8). The surface was carefully cut and the excavated

soil stockpiled separate according to depth and soil horizon, in order to be able to restore the

original profile following installation of the instruments. The unfrozen layer was 1.2 m deep

at the time of installation. The entire profile was quite heterogeneous but consisted mainly of

silty clay, with a stone content of less than 10 %. The stone content increased to about 50 %

345 in a slightly inclined layer approximately 20 cm thick that occurred at a depth of 50 cm; the stones were quite large (including cobbles in excess of 6 cm in diameter) but showed no preferred orientation. Below 85 cm depth the soil was coal-rich and contained massive coal lenses.

Sensors were installed horizontally and vertically in the undisturbed wall of the trench profile

350 (Appendix C Figure C8b) to obtain a 2 D profile instrumentation, which was then backfilled. Soil samples were collected during the installation so that physical parameters could be analyzed. The stratigraphic and textural information from the soil profile, including soil organic carbon (C), nitrogen (N) and sulfur (S) concentrations, can be found in Appendix F.

*Soil sampling 2007*

355 A number of additional soil profiles in the vicinity of the Bayelva site were sampled in 2007 (Figure 2). Data for the soil texture, bulk density and carbon, nitrogen and sulphur (C, N, S) content of these profiles is also included in Appendix F (Table F1).

*Instrument installation and soil sampling 2009*

A new set of measurement profiles was established within a second non-sorted circle close to

360 the 1998 profiles (Appendix C, Figures. C9). The site was chosen for its similarity to the 1998 site, in order to provide an analogous continuation of the long term subsurface data from the 1998 profiles. Data on the soil texture, its bulk density, and carbon, nitrogen and sulphur content can be found in Appendix F.

All profiles show cryoturbation features at the surface through patterned ground (non-sorted

365 circles) and also within the soil profile, for example, roots that are transported downwards through cryoturbation (Figures C8&C9).



### 3.2.2 Ground temperature

*Soil profile sensor installation, 1998*

Pairs of soil temperature and soil water content sensors were installed over a vertical 2D

profile in 1998; their positions are shown in Appendix C (Figures C7 & C8). The probes

automatically recorded temperatures from that time until January 2012. The temperatures

were measured using CSI (Campbell Scientific Ltd.) thermistors connected to a CR10X

datalogger with an AM416 multiplexer. The voltage measured for each sensor was stored in a

database. Temperatures were calculated from the raw data using the Steinhart-Hart equation

(Steinhart and Hart, 1968) and sensor calibration at 0°C (Table D2). The average accuracy

determined through calibration was 0.01°C with a precision better than 0.002°C. However,

the max deviation at 0°C was 0.1°C. After installation, the sensors cannot be re-calibrated.

However, temperatures during spring thaw and fall refreezing are stable at the phase change

temperature (the zero-curtain effect in permafrost soils). Assuming that freezing point

depression (i.e. soil type and soil water composition) does not change significantly from year

to year, these periods can be used to evaluate sensor stability. During the first two years

(1998-1999) the temperatures during these periods were stable to within 0.1°C. Subsequently,

temperature readings from nine of the 32 sensors subsequently started to drift and showed a

positive offset relative to the ~0°C zero-curtain period. The temperature sensor at 19 cm depth

for example, showed correct temperatures during the fall freeze-back of 1998 and 1999, but in

the fall of 2000 the measured soil temperatures shifted to positive values by 0.5°C (offset

from the fall zero-curtain). This offset increased further to about 6°C in 2009. All sensors

with temperature offsets of 0.5°C or more were flagged accordingly within the data set. The

positive temperature offset was most likely due additional resistance, either in the cables due

to leakage currents or in the multiplex due to corrosion. In addition to the temperature offset,

the data quality of the years 2002 to 2007 was strongly affected by irregular spikes caused by

the multiplexer. In the worst year, 2002, these spikes occurred for 5 to 7% of all measured

values in all soil temperature series of the 2D profile. Spikes were flagged automatically and

manually within the data set. The external charge controller regulating the solar and wind

energy supply was changed in 2002 following a power failure, resulting in increased noise

Levels in the measured temperatures between 2002 and 2012 and subsequent discontinuation

of this data series.

*Soil profile sensor installation, 1999*

Temperature sensors were installed in 1999 in one additional vertical profile, 2 m apart,

within the fenced off area, below vegetation (b in Appendix C, Figure C6). A small-diameter

(vertical) hole was drilled by a hand drill into the ground in the fall of 1999. The PT100

temperature sensors were calibrated in ice-slush water for at least 12 hours to determine the

offset at 0°C. The max deviation at 0°C was 0.1°C. Similar to the 107 probes, the average

accuracy determined through calibration was 0.01°C with a precision better than 0.002°C.

Wooden sticks to which sensors had been attached were then inserted into the holes down to

1.50 m into the permanently frozen soil beneath the active layer in such a way that the sensors

formed a tight fit in the hole and no air was able to pass. The temperature sensors were

attached at regular intervals to bamboo sticks. The upper sensors were affected on September

11, 2003 by reindeer disturbance. Furthermore, the upper two sensors moved upwards by

about 10 cm between 2003 and 2016 due to frost heave and they were removed from the data
series.

*Soil profile sensor installation, 2009*

Pairs of soil temperature (CS107) and water content (TDR) probes were installed during

August 2009, close to the first profile, that was installed in 1998 (Appendix C, Figure C9).
Soil temperatures were measured at 11 depths in a profile from the surface down to a depth of
1.4 m. The data quality from the 2009 system improved considerably to the 1998 system,
probably due to the data logging and multiplexer system used (CR10X from 1998 to 2011;
CR1000 since 2009). Data collection is ongoing.


### 3.2.3 Soil dielectric number, volumetric liquid water content, and bulk electrical conductivity

*Soil profile sensor installations 1998 and 2009*

Time-domain reflectometry (TDR) probes were installed were installed in horizontally

adjacent to the temperature probes installed in soil profiles in 1998 and 2009 (see above and
Appendix C (Table C5).

These TDR probes automatically recorded hourly measurements of bulk electric conductivity
and the dielectric number, obtained by measuring the amplitude at very long times and the
$L_a/L$ ratio (ratio of the apparent to real probe length, corresponding to the square root of the

dielectric number). From 1998 to 2008 the measurements were obtained using a Tektronix
1502B cable tester connected to a CR10X data logger and custom made triple wire 24 cm

probes. Since 2009 a CSI TDR100 reflectometer has been used, connected to a CR1000 datalogger and 30 cm TDR CS605 probes (Campbell Scientific Ltd.). All TDR probes were calibrated for probe offset following the method described in Heimovaara and de Water,

(1993) and the CSI TDR 100 manual (http://www.campbellsci.de/tdr100). The dielectric number data and computed volumetric liquid water in frozen and unfrozen soil are provided with the data set. The calculation for volumetric liquid water content takes into account four phases of the soil medium (air, water, ice, mineral) and uses the mixing model from Roth et al. (1990; Appendix D, D1).

Time-domain reflectometry can also be used to measure the impedance Z ($\Omega$) of the bulk soil, which is related to the soil's bulk electrical conductivity (BEC). These data were used to infer the electrical conductivity of soil water and solute transport in the non-sorted circle over a period of one year (Boike et al., 2008a). The impedance can be determined from the attenuation of the electromagnetic wave traveling along the TDR probe after all multiple

reflections have ceased and the signal stabilized. The bulk electronic conductivities were recorded hourly using the TDR setup described in this section. The equipment installed in 1998 was calibrated in pure water, in air, and in NaCl solutions with various known conductivities, and is corrected for temperature (25°C), following the method by Heimovaara et al., (1995). The TDR equipment installed in 2009 was not calibrated for BEC and a probe

constant of 1 was used for BEC waveform retrieval; Campbell Scientific Ltd. suggests a probe constant ($K_p$) for the CS605 probes of 1.74. The lack calibration explains the differences of the data sets recorded between the time 1998-2009 and 2009 to present (Figure 3i). Measurements of electric conductivity and the dielectric number were affected by irregular spikes and possibly by sensor drift similarly to the soil temperature measurements.

Spikes were flagged automatically and manually, while drifts could not be flagged due to the

lack of reference data. Overall, the quality of soil data instrumented in 1998 is of reduced

quality since July 2005. Values for dielectric number, computed volumetric liquid water

content, and bulk soil electrical conductivity can be found in the data set.

### 3.2.4  Soil heat flux

Two heat flux plates (Hukseflux HFP01), one installed at 0.18 m depth below the unvegetated

center of the non-sorted circle and the second at 0.24 m depth below the vegetated rim of the

non-sorted circle, recorded ground heat flux between September 1998 and September 2009

(Figure 3). The manufacturer's calibration values were used to record heat flux in W m$^{-2}$.


### 3.2.5  Permafrost temperature

A 9.3 m borehole was drilled on March 30, 2009 and cased with PVC. Ten thermometers

were installed, one above the ground surface and nine from the surface down to 9 m depth (at

0, 0.5, 1.0, 1.5, 2.5, 3.5, 5.5, 7.5, 9 m below ground surface, Appendix C, Figure C10). The

casing was left open (not refilled) so that the thermometer chain could be retrieved or

replaced. Because of instrument failure, thermometers were retrieved and replaced several

times. From August 2009, a Geoprecision-M datalogger and sensor chain were used.

Temperatures were recorded at hourly intervals, with no averaging; no data was recorded

between 30 January 2011 and 29 March 2011, due to a low battery voltage. Geoprecision

claims an accuracy of +/- 0.05°C (at 0°C) and a resolution of 0.01°C, suitable for

measurements in the range from -50 to +120°C. However, comparison measurements using an

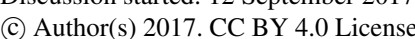

PT100 thermometer at the same depths in the borehole showed a deviation of up to 0.2°C. In 2014, a wooden shield was installed over the casing to prevent warming due to radiation and to facilitate natural ventilation (Appendix C, figure C10). It is recommended that the

temperature data from sensors installed above ground and down to a depth of 1.5 m should be used with caution because of air exchange within the casing. These data are flagged in the data series.

## 4   Data quality control and availability

The primary purpose of quality control of observational data is to detect missing data, errors in the data, and possibly to correct errors in order to ensure the highest possible standard of accuracy and optimal use of the data by the broadest range of possible users. We differentiate between different data Levels, from online data (http://www.awi.de/im-fokus/permafrost/direkter-draht-in-den-permafrost.html) to Level 2 data (Table 3). Data

supplied with this paper are Level 1 and Level 2 data. Level 1 data have undergone an extensive quality-control and are flagged with regards to maintenance periods, physical plausibility, spike/constant value detection, sensor drifts (Table 3). Level 2 data are compiled for special purposes such as a combination of data series from multiple sensors and gap-filling of data (documentation of source data is supplied in the PANGAEA data archive). Examples

in this paper of Level 2 data are the snow cover, soil temperature, and soil volumetric water content data series that have been combined from various stations into a single data series, in order to obtain a long term picture. There are 8 quality control types (Table 3). These flags include information on the system's maintenance, no data availability, system error, as well as

on consistency checks based on physical limits, gradients, and plausibility. Due to the failure

of some sensors that cannot be retrieved for repair or re-calibration (for example, sensors

installed in the ground), the initial accuracy and precision of the sensors is not maintained.

In the case of soil temperature, the accuracy can be estimated by analysis of temperatures

relative to the fall zero-curtain effect assuming soil water composition is similar from year to

year. As an example, our temperature data have been checked through the fall zero-curtain

effect and thus provide information on any reduction in accuracy (see Table 3, Flag 7). These

checks are mandatory when small warming trends are to be estimated and interpreted. The

selection of flagged data depends on its application and the required accuracies.

The data from this Bayelva site have been widely used for the development and evaluation of

land surface models. The newly collated dataset will allow multi-year model runs, with

improved quality control and checks. The dataset presented herein is freely available, either as

download from PANGAEA (https://doi.org/10.1594/PANGAEA.880120) or from the

supplementary material of this paper. The data are provided as ASCII files. Other available

data sets include a flux data set in the European Fluxnet Database (Lüers and Boike, 2013)

and aerial imagery of the snow cover (Westermann et al., 2015). Volumetric soil liquid water

content and snow cover data are transferred in real time to the NASA SMAP mission

(https://smap.jpl.nasa.gov/) and are uploaded to the PANGAEA archives on an annual basis.

## 5   Outlook

Permafrost around the Arctic is thawing and warming (Christiansen et al., 2010; Romanovsky

et al., 2008), and this is true for Svalbard (Isaksen et al., 2007b) and the Bayelva site as well.

Mean annual, summer and winter soil temperature data at all depths have been warming over
the period of record (Figure 4, three depths are shown) as well as in the deeper permafrost
(Figure 4 h). Interannual to sub-decadal variability is evident in the data and results mostly
from differences during the winter months. Future analysis is required to detangle the

relationship between potential meteorological drivers and permafrost degradation at this site.
The data set described and distributed in this paper provides a basis for analyzing this
relationship at one site and a means of calibrating Earth System Modelling efforts over a long
observational period. Developing predictive capacity for permafrost warming will be key to
understanding the role of the permafrost feedback in the global climate system.


Acknowledgements: The logistical support provided by the AWIPEV Research Base at
Ny-Ålesund is gratefully acknowledged. Field support, including data collection, was also

provided by Konstanze Piel, Christian Wille, Steffen Frey, Conrad Kopsch, Günther Stoof
and Peter Schreiber. We appreciate the support of Klaus-Dieter Matz and Frank Scholten for
the preparation of the GeoTIFF versions of the HRSC-AX data products. The authors
acknowledge the financial support provided through the European Union's FP7-ENV
PAGE21 project under contract number GA282700.




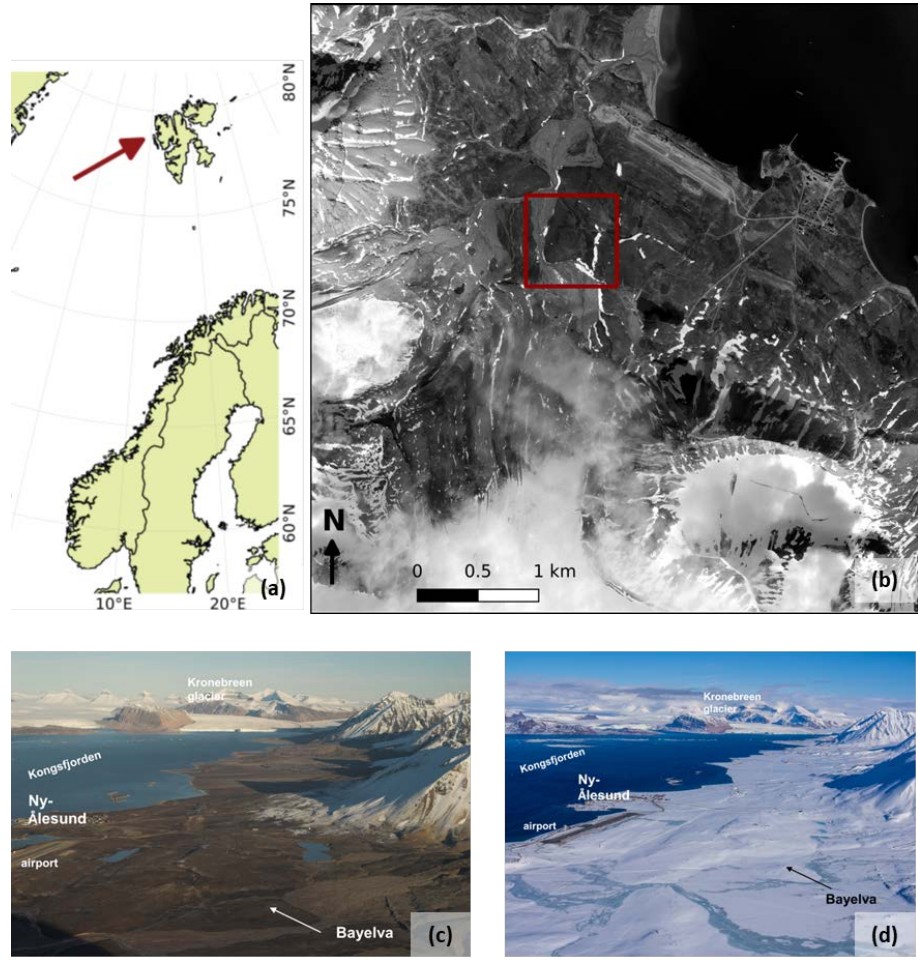

**Figure 1. (a)** Location of the Bayelva study site on Spitsbergen, western Svalbard. **(b)** The

site is located about 3 km from Ny-Ålesund in the Bayelva River catchment area, between

two mountains (Zeppelinfjellet and Scheteligfjellet), and in front of the Brøggerbreen glacier.

Aerial image captured in August 2008 using a high resolution HRSC-AX camera (Hauber et

al., 2011a; Hauber et al., 2011b): data and metadata for the high resolution the HRSC-AX

image covering the entire area shown in Figure 1b are provided in Appendix B **(c)** & **(d).** The

area of the site under summer conditions (August 2008) and spring conditions (April 2016).



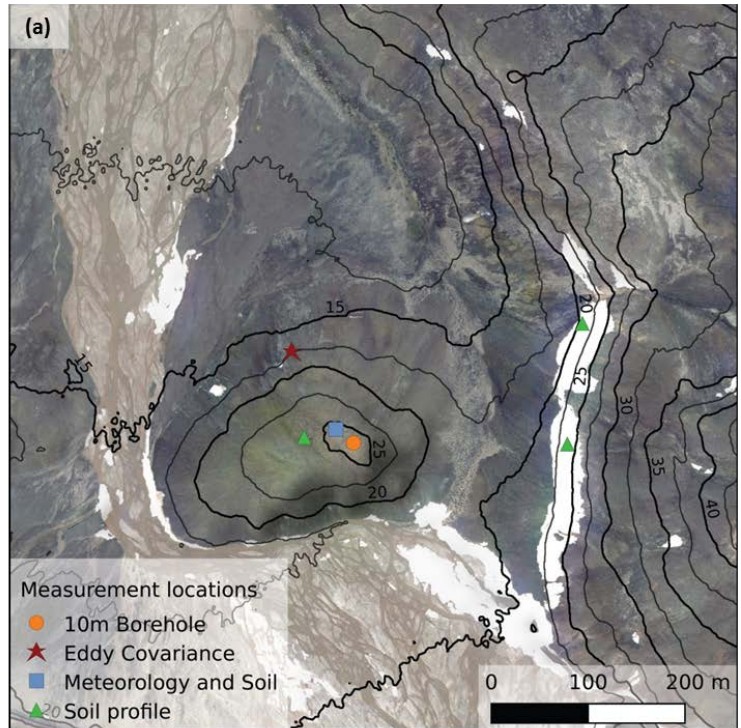

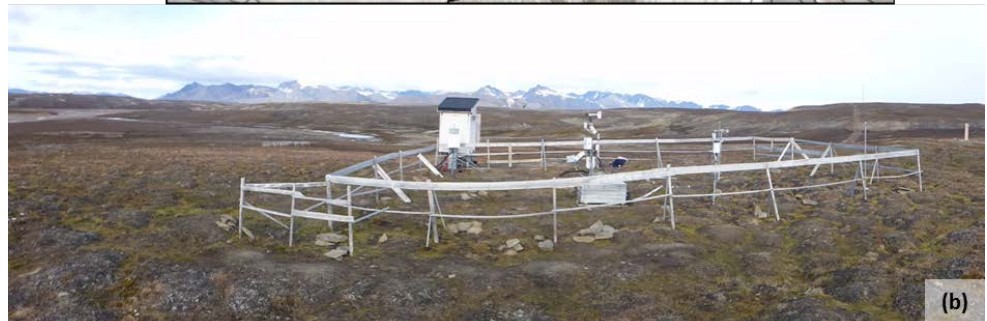

**Figure 2.** Detailed images of the Bayelva study site on Spitsbergen. **(a)** Aerial orthoimage
(20 cm/px) and topographic information (2.5 m contour lines) derived from a corresponding
Digital Elevation Model (DEM) with a cell size of 0.5 m. The locations of the instrumental
and sampling sites are marked by coloured symbols. The site is located on a small hill; white
areas are snow fields remaining late in the season. The colour image and DEM were obtained
with an HRSC-AX camera in August 2008 (data and metadata for the high resolution digital
elevation model covering the entire area shown in Figure 1b are provided in Appendix B).



**(b)** Most of the instruments are within a fenced-off area to protect them from reindeer dam-
age. Note that the eddy covariance station and permafrost borehole are located outside the
fenced area.









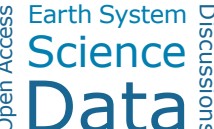

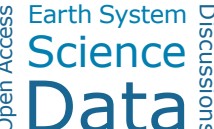



Figure 3. Time series of Bayelva data provided in this paper divided into three plots f. a-f: meteorological data, g-l: soil/subsurface data, m-p: snow data. The data are organized following the structure of Appendix G. Further details on the sensors and periods of operation are given in Table 2.


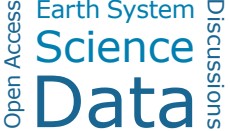

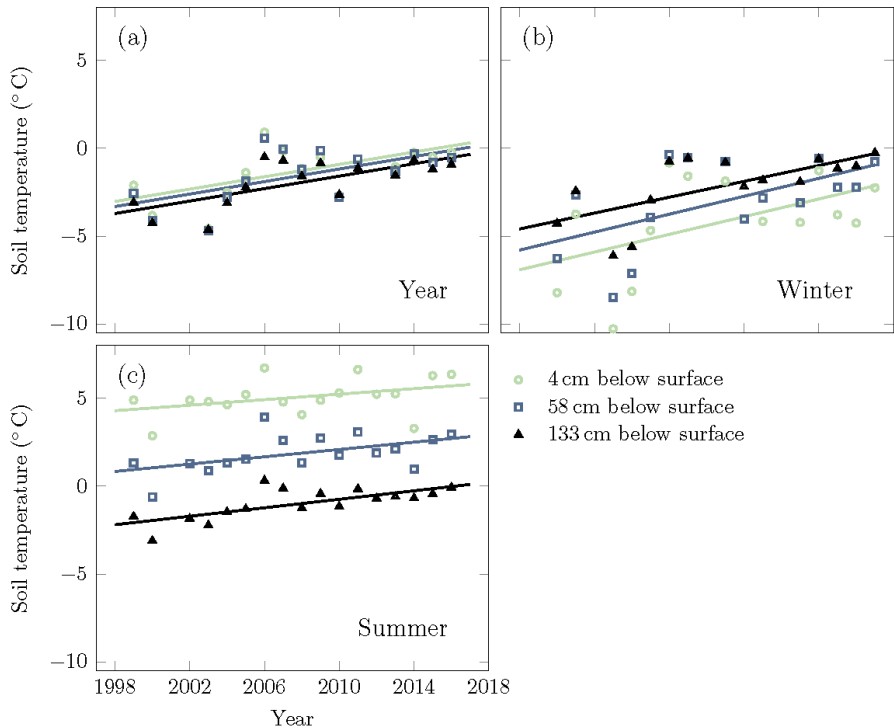

Figure 4. Bayelva soil temperature trends for three depths from 1998- September 2017 using

Level 2 soil temperatures. a) yearly trends at all three depths of 4, 58, 138 cms are

0.18°C/year (standard error of trends: ± 0.07, 0.06, 0.05 °C/year) b) winter trends (months

December, January, February) are 0.25, 0.25, 0.23°C/winter (standard error of trends: ± 0.12,

0.11, 0.07 °C/winter) and c) summer trends (months June, July, August) are 0.08, 0.1,

0.12°C/summer (standard error of trend ± 0.05, 0.04, 0.03 °C/summer). Years in which data

gaps of more than 48 hours exceeded 5% of data were included. Data gaps were interpolated

linearly.





| Variable | Value | Source |
|---|---|---|
| **Surface characteristic** | | |
| Summer albedo | 0.15 | Westermann et al. (2009) |
| Summer Bowen ratio | 1 (0.25–2) | Westermann et al. (2009) |
| Summer roughness length [mm] | 1 (by fitting an energy balance model) | Westermann (2010) |
| | 7 (eddy covariance) | |
| **Snow properties** | | |
| Snow albedo | Average for snow covered period prior to melt: 0.81 (2009–2016) | This paper |
| End of the snow ablation | 8 June–11 July | This paper (1999–2016) |
| Maximum snow depth (end of season before ablation) [m] | range: 0.65–1.42 m | This paper (1999–2016) |
| | average 0.9 | |
| End of season snow density [kg m$^{-3}$] | 350 ± 50 | Westermann et al. (2009) |
| | | (Boike et al., 2003a)Boike et al. (2003) |
| | | Appendix E for individual profiles |
| Snow thermal conductivity [W m$^{-1}$ K$^{-1}$] | 0.34 | Computed from snow density of 370 kg m$^{-3}$ using Yen (1981) |
| Snow thermal conductivity [W m$^{-1}$ K$^{-1}$] | 0.45 ± 0.15 | Westermann et al. (2009) |
| **Soil properties** | | |
| Organic layer thickness | 0–5 cm (bare to vegetated areas; up to 15 cm in wetter areas) | This paper |
| Thawed soil thermal conductivity [W m$^{-1}$ K$^{-1}$] | 1.3 ± 0.4 at volumetric soil liquid water content 20%–40% | Westermann et al. (2009) |
| | (soil profile 1998: profile average) | |
| Soil heat capacity thawed [10$^6$ J K$^{-1}$ m$^{-3}$] | 2.3 ± 0.5 at volumetric liquidwater content 20%–40% | Westermann et al. (2009) |
| Frozen soil thermal conductivity [W m$^{-1}$ K$^{-1}$] | 2.0–2.5 | Westermann et al. 2011(Westermann et al., 2011a) based on thermal diffusivity provided by Roth and Boike (2001) |
| Soil texture | Silty clay to sandy silt, , stone content < 10% at surface with up to 50% | Boike et al., 2008a<br>Appendix F for individual pedons |



| Variable | Value | Source |
|---|---|---|
| Soil bulk density [kg m$^{-3}$] | Average: 1.9 x 10$^3$ kg m$^{-3}$ | Boike et al. (2008a) |
| | | Roth and Boike (2001) |
| | | Appendix F for individual pedons |
| Soil carbon density [kg C m$^{-3}$] | 6 kg m$^{-3}$ (down to 78 cm) | Chadburn et al. (2017) |
| | 0–60 kg m$^{-3}$ (range) | Appendix F for individual pedons |
| | 2–22 kg m$^{-3}$ (range over aver-aged vertical profile) | |
| Organic carbon stock [kg C m$^{-2}$] | 4.3 (range: 1.1–7.9 kg C m$^{-2}$ for 0–100 cm | Area between flood plain and sea (Yoshitake et al., 2011). |
| | | Appendix F for individual pedons |
| Saturated hydraulic conductivity [m s$^{-1}$] | 10.9 x 10$^{-6}$ | Weismüller et al. (2011) |
| | 7.11 x 10$^{-6}$ | Ekici et al. (2015) |
| Clapp-Hornberger exponent (b factor) | 5 (for silty loam soil) | Chadburn et al. (2015) |
| porosity (volumetric water content at saturation) | 0.36–0.5 | Roth and Boike (2001) |
| | | Weissmüller et al., 2005 |
| Van Genuchten Parameters: Alpha [1 mm$^{-1}$] | Alpha = 0.002 | Weismüller et al. (2011) |
| Van Genuchten Parameters: n [unit-free] | 1.2–1.5 | Weismüller et al. (2011) |
| **Vegetation characteristics** | | |
| | | |
| Vegetation height | 0–20 cm (bare soil to vegetated areas) | This paper |
| Vegetation fractional coverage | 50–60 area %, vascular plants cover up to 30% | This paper |
| | | Lloyd (2001a) |
| Vegetation type | "Lichen heath" | Brattbakk (1981) |
| | "High Arctic tundra in mesic–xeric conditions" | Cannone et al. (2004) |
| | "A mixed community of bryo-phytes (e.g., Sanionia uncinata, Aulacomnium turgidum and Dicranoweissia sp.) and vascu-lar plants (e.g. Salix polaris, Saxifraga oppositifolia, Oxyria digyna and Dryas octopetala) covered the well developed vegetation among topograph- | Uchida et al. (2009) |



| Variable | Value | Source |
|---|---|---|
| | ical undulation." | |
| Leaf Area Index (LAI) | 0 (no vegetation areas) | Cannone et al. (2016) |
| | 0.3–0.7 | |
| Root depth | ~19 cm | This paper |
| Leaf dry biomass | 26–195 g m$^{-2}$ for small patches of 100% vegetation cover of three different species | Muraoka et al. (2008) |
| **Landscape** | | |
| Landscape type | Glacial outwash flood plain | This paper |
| Bioclimate subzones | Subzones A and B, characterized by barren ground and scattered vegetation. The vegetation mainly consists of non-vascular plants (40% cover) and a few vascular plants (5% cover). The average total phytomass in this subzone is lower than 3 t ha$^{-1}$ and the net annual production is lower than 0.3 t ha$^{-1}$. | CAVM-Team (2005) |

**Table 1.** Site description parameter for earth system model input. Values have been computed and compiled for the Bayelva site and surrounding areas.




**Meteorological sensors**

| Variable | Sensor | Period of operation from | to | Height [m] | Unit | Measuring interval | Integration method | Spectral range | (Field) accuracy [±] |
|---|---|---|---|---|---|---|---|---|---|
| Air temperature | Rotronic MP103A | Apr 2000 | Aug 2009 | 2 | °C | 20 s | avg[1] 1 h | | 0.5 °C (at -40 °C to +60 °C) |
| Air temperature | Vaisala HMP45 | Aug 2009 | … | 2 | °C | 60 s | avg 30 min | | 0.2 °C (at 20 °C) |
| Air temperature | 1x PT100 | Aug 1999 | Aug 2009 | 8/1999[2]: 2 4/2000: 1 | °C | 15 min | avg 1 h | | 0.03 °C |
| Air temperature | 1x PT100 | Aug 2009 | … | 1 | °C | 60 s | avg 30 min | | 0.03 °C |
| Relative humidity | Rotronic MP103A | Apr 2000 | Aug 2009 | 2 | % | 20 s | avg 1 h | | 3 % (at 20 °C) |
| Relative humidity | Vaisala HMP45 | Aug 2009 | … | 2 | % | 60 s | avg 30 min | | 2 % (at 20 °C, 0 to 90 % RH); 3 % (at 20 °C, 90 to 100 % RH) |
| Wind direction | R.M. Young anemometer 05103 | Sep 1998 | Aug 2009 | 3 | deg | | avg 1 h | | 3° |
| Wind direction | R.M. Young anemometer 05103 | Aug 2009 | … | 3 | deg | | avg 30 min | | 3° |

[1] average
[2] from [M/YYYY]



| Variable | Sensor | Period of operation from | to | Height [m] | Unit | Measuring interval | Integration method | Spectral range | (Field) accuracy [±] |
|---|---|---|---|---|---|---|---|---|---|
| Windspeed | R.M. Young anemometer 05103 | Sep 1998 | Aug 2009 | 3 | ms⁻¹ | | avg 1 h | | 0.3 ms⁻¹ |
| Windspeed | R.M. Young anemometer 05103 | Aug 2009 | … | 3 | ms⁻¹ | | avg 30 min | | 0.3 ms⁻¹ |
| Net radiation | CS³ Q7 | Sep 1998 | Apr 2000 | 1.14 | Wm⁻² | 20 s | avg 1 h | 0.25 to 60 µm | 6-10 % |
| Net radiation | Kipp & Zonen NR-LITE | Apr 2000 | May 2002 | 4/2000: 1.14 9/2003: 1.6 | Wm⁻² | 20 s | avg 1 h | 0 to 100 µm | 3-20 % |
| Net radiation | CS Q7 | May 2002 | Sep 2003 | 1.14 | Wm⁻² | 20 s | avg 1 h | 0.25 to 60 µm | 6-10 % |
| Net radiation | Kipp & Zonen NR-LITE | Sep 2003 | Aug 2009 | 4/2000: 1.14 9/2003: 1.6 | Wm⁻² | 20 s | avg 1 h | 0 to 100 µm | 3-20 % |
| Incoming shortwave radiation | Skye Pyranometer SP1110 | Sep 1998 | Aug 2009 | 2 | Wm⁻² | 20 s | avg 1 h | 350 nm to 1100 nm | 5 % (typically <±3 %) |
| Outgoing longwave radiation | Kipp & Zonen Pyrgeometer CG1 | May 2002 | Aug 2009 | 5/2002: 2.0 9/2003: 1.6 | Wm⁻² | 20 s | avg 1 h | 4.5 to 42 µm (50% points) | 10 % for daily totals |
| Four components radiation | Huxeflux NR01 | Aug 2009 | … | 1.56 | Wm⁻² | 60 s | avg 30 min | 305 to 2800 nm⁴; 4500 to 50000 nmᶜ | 10 % for daily sums |

³ Campbell Scientific Ltd.
⁴ 50% transmission points



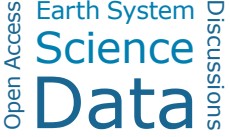

| Variable | Sensor | Period of operation from | to | Height [m] | Unit | Measuring interval | Integration method | Spectral range | (Field) accuracy [±] |
|---|---|---|---|---|---|---|---|---|---|
| Precipitation | Young 52203 unheated Tipping Bucket Rain Gauge | Sep 1998 | July 2010 | 1.68 | mm | | sum 1 h | | 2 % (up to 25 mm/hour) |
| Precipitation | Young 52203 unheated Tipping Bucket Rain Gauge | July 2010 | … | 1.68 | mm | | sum 30 min | | 2 % (up to 25 mm/hour) |
| **Snow sensors** | | | | | | | | | |
| Snowdepth | CS SR50 ultrasound | Sep 1998 | Aug 2009 | 9/1999: 2.00 7/2001: 1.60 9/2003: 1.45 | m | | single 12 h | | 1 cm |
| Snowdepth | CS SR50 ultrasound | Aug 2009 | … | 1.45 | m | | single 30 min | | 1 cm |
| Snowdepth @eddy covariance site | CS SR50 ultrasound | Mar 2007 | … | 2.4 | m | 3 min | avg 1 h | | 1 cm |
| Snowdepth | Jenoptik SHM30 laser distance | Aug 2013 | … | | m | | single 0.5 h | | 5 mm |
| Air/snow temperature | 3x PT100 | Aug 1999 | Aug 2009 | 8/1999: 0.20; 0.35; 0.48 9/2003: 0.01; 0.20; 0.48 9/2004: 0.01; 0.20; 0.35 | °C | 15 min | avg 1 h | | 0.1 °C |



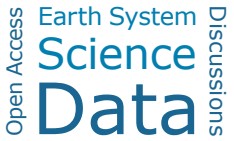

| Variable | Sensor | Period of operation from | to | Height [m] | Unit | Measuring interval | Integration method | Spectral range | (Field) accuracy [±] |
|---|---|---|---|---|---|---|---|---|---|
| Air/snow temperature | 2x PT100 | Aug 2009 | … | 0.04; 0.2 | °C | 60 s | avg 30 min | | 0.1 °C |
| Snow dielectric number | TDR Tektronix 1502B, SDM50 Multiplexer, 1x TDR triple wire 0.5m | Sep 1998 | Dec 2009 | 0 to 0.5 (vertical) | | | single 1 h | | |
| Snow dielectric number | TDR100, SDM50 Multiplexer, 1x CS605 probes, triple wire 0.3 m | Aug 2009 | … | 0 to 0.3 (vertical) | | | single 1 h | | |
| **Soil sensors** | | | | | | | | | |
| Soil temperature (2D), profile (a) | 32x CS 107 | Sep 1998 | Jan 2012 | -0.045 to -1.25[5] | °C | | single 1 h | | 0.1 °C |
| Soil temperature (1D), profile (c) | 8x CS 107 | Aug 2009 | … | -0.01 to -1.41[6] | °C | 60 s | avg 1 h | | 0.1 °C |
| Soil temperature (1D), profile (b) | 6x PT100 | Aug 1999 | Aug 2009 | -0.02 to -1.43[7] | °C | 15 min | avg 1 h | | 0.1 °C |

[5] See appendix C, figure C8
[6] Exact heights [m]: -0.01; -0.11; -0.21; -0.37; -0.55; -0.71; -0.89; -1.41
[7] Exact heights [m]: -0.02; -0.05; -0.24; -0.53; -0.93; -1.43



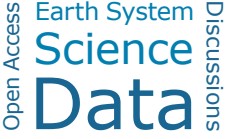

| Variable | Sensor | Period of operation from | to | Height [m] | Unit | Measuring interval | Integration method | Spectral range | (Field) accuracy [±] |
|---|---|---|---|---|---|---|---|---|---|
| Soil heat flux | 2x Hukseflux HFP01 | Sep 1998 | Dec 2012 | 0.18; 0.24 | Wm⁻² | | single 1 h | | -15 % to +5 % |
| Soil electrical conductivity | TDR Tektronix 1502B, SDM50 Multiplexer, 29x TDR triple wire 0.24 m long probes | Sep 1998 | Dec 2009 | 0 to -1.2 | dSm⁻¹ | | single 1 h | | |
| Soil electrical conductivity | TDR100, SDM50 Multiplexer, 7x CS605 probes, triple wire 0.3 m | Aug 2009 | ... | -0.01 to -0.89 | dSm⁻¹ | | single 1 h | | |
| Soil liquid volumetric water content | TDR Tektronix 1502B, SDM50 Multiplexer, 29x TDR triple wire 0.24 m long probes | Sep 1998 | Dec 2009 | 0 to -1.2 | | | single 1 h | | |
| Soil liquid volumetric water content | TDR100, SDM50 Multiplexer, 7x CS605 probes, triple wire 0.3 m | Aug 2009 | ... | -0.01 to -0.89 | | | single 1 h | | |
| Permafrost temperature | Geoprecision Temperature Chain with 10 sensors, M-Log | Aug 2009 | Sep 2015 | 0.5; 0; -0.5; -1; -1.5; -2.5; -3.5; -5.5; | °C | | single 1 h | | 0.2 °C (absolute accuracy) |



| Variable | Sensor | Period of operation from | to | Height [m] | Unit | Measuring interval | Integration method | Spectral range | (Field) accuracy [±] |
|---|---|---|---|---|---|---|---|---|---|
| | | | | -7.5; -9 | | | | | |
| **Generic sensors** | | | | | | | | | |
| Snow/surface observation | Camera K1 | Aug 2013 | … | 2 | px | | single 1 h | | |

**Table 2.** List of sensors, parameters, and instrument characteristics for the automated time series data from the Bayelva site, 1998-2017. Additional (not automated) data on snow cover and soil profiles can be found in Appendix E and Appendix F. Positive heights are above surface, negative heights are below surface.





| Flag | Meaning | Description |
|---|---|---|
| **ONL** | Online data | Data from online stations, daily download, used for online status check |
| **RAW** | Raw data | Base data from offline stations, 3-monthly backup of online data, used for maintenance check in the field |
| **LV0** | Level 0 | Standardised data with equal time steps, without gaps and in a standard data format |
| **LV1** | Level 1 | Quality-controlled data including flags; quality control includes maintenance periods, physical plausibility, spike/constant value detection, sensor drifts and snow on sensor detection |
| **LV2** | Level 2 | Modified data compiled for special purposes such as combined data series from multiple sensors and gap-filled data |

| Flag | Meaning | Description |
|---|---|---|
| **0** | Good data | All quality tests passed |
| **1** | No data | Missing value |
| **2** | System error | System failure led to corrupted data, e.g. when the power supply broke down, sensors were removed from their proper location, sensors broke or the data logger saved error codes |
| **3** | Maintenance | Values influenced by the installation, calibration and cleaning of sensors or programming of the data logger; information from field protocols of engineers |
| **4** | Physical limits | Values outside the physically possible or likely limits, e.g. relative humidity should be in a range of 0-100% |
| **5** | Gradient | Values unlikely because of prolonged constant periods or high/low spikes; test within each single series |
| **6** | Plausibility | Values unlikely in comparison with other series or for a given time of the year; flagged manually by engineers |
| **7** | Decreased accuracy | Values with decreased sensor accuracy, e.g. identified when freezing soil does not have a temperature of 0°C |
| **8** | Snow-covered | Good data, but the sensor is snow-covered |


**Table 3.** Description of data Level and quality control for data flags. Most flags are run automatically, few are done manually (for example, 3-maintainance, 6-plausibility).





# Appendices

**Appendix A:** Symbols and abbreviations

**Appendix B:** Description of HRSC-AX images

**Appendix C:** Metadata description and pictures of climate, soil and permafrost stations, and instruments

**Appendix D:** Calculation and correction of soil and meteorological parameters

**Appendix E:** Description and data of snow profiles

**Appendix F:** Description and data of soil profiles

**Appendix G:** Names of the variables and units for data files



# Appendix A: Symbols and abbreviations


$R_T$ = measured resistance

$\delta_O$ = resistance offset at 0°C

$\frac{L_a}{L}$ = apparent length of the TDR probes (TDR datalogger output)

$\varepsilon_b$ = bulk dielectric number (Ka)

$\varepsilon_l$ = temperature-dependent dielectric number of liquid water

$\varepsilon_i$ = dielectric number of ice

$\varepsilon_s$ = dielectric number of soil matrix

$\varepsilon_a$ = dielectric number of air

$\theta_l$ = volumetric liquid water content

$\theta_i$ = volumetric ice content

$\theta_s$ = volume fraction of soil matrix

$\theta_a$ = volume fraction of air

$\theta_{tot}$ = total volumetric water content (liquid water and ice)

$\alpha$ = geometry of the medium in relation to the orientation of the applied electrical field (see

Roth et al. (1990))

$\Phi$ = porosity

Z= impedance

BEC= bulk electrical conductivity

$NetRad$ = radiation netto all [W m$^{-2}$]

$NetRad\_raw$ = uncorrected radiation netto all [W m$^{-2}$]

$wind\_v$ = wind speed [m s$^{-1}$]

$TOC$ = total organic carbon [%]

$SOCC$ = soil organic carbon content [kg m$^{-2}$]





$CD_{bulk}$ = bulk carbon density [kg m$^{-3}$]

$\overline{\rho_{bulk}}$ = average dry bulk density [kg m$^{-3}$]

$z$ = layer thickness [m]



## Appendix B. Description of HRSC-AX images


HRSC is a multisensor pushbroom instrument with 9 CCD line sensors mounted in parallel (Figure B1) that has been in orbit around Mars since January 2004 on ESA's *Mars Express spacecraft* (Gwinner et al., 2016). It simultaneously obtains high-resolution stereo, multicolor, and multiphase images. Digital photogrammetric techniques are used to reconstruct the topog-

raphy on the basis of five stereo channels, which provide five different views of the ground.

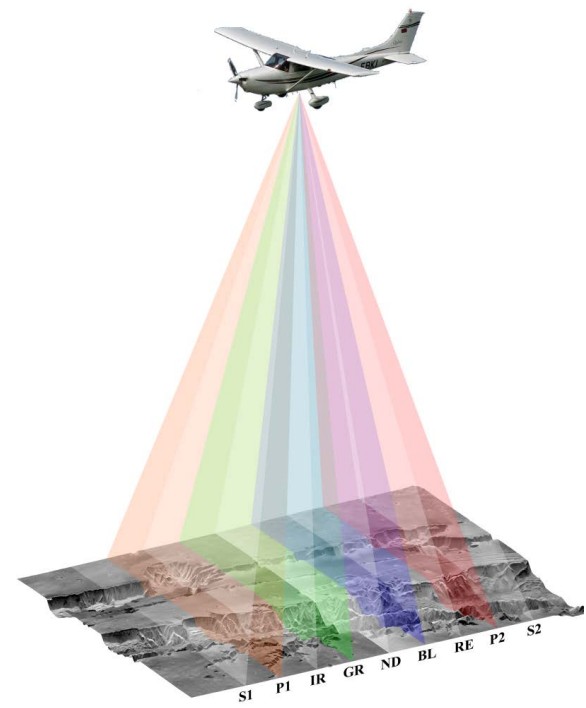

**Figure B1.** Operating principle of the airborne High-Resolution Stereo Camera (HRSC-AX), and viewing geometry of the individual Charge-Coupled Device (CCD) sensors. ND—nadir channel; S1, S2—stereo 1 and stereo 2; P1 and P2—photometry 1 and photometry 2; IR—

near-infrared channel; GR—green channel; BL—blue channel; RE—red channel. All nine line sensors have a crosstrack field of view of ± 6 °.



The four color channels (blue, green, red, and near-infrared; Figure B2) are used to make true
orthophotos in color and false color. The particular value of HRSC is the stereo capability,
which allows the systematic production of high-resolution DEMs with grid sizes between 50
and 100 m (Wewel et al. (2000); Scholten and Gwinner (2004); Scholten et al. (2005);
Gwinner et al. (2005, 2010).

Since 1997 different airborne versions of HRSC have been developed, one of which (HRSC-
AX) was used to acquire stereo color images over Svalbard. The principles of HRSC-AX data
processing are described by Gwinner et al. (2006). Data on the camera orientation are recon-
structed from a global positioning system inertial navigation system (GPS INS). HRSC-AX
has been used in diverse technical and scientific applications (e.g. Gwinner et al. (1999);
Gwinner et al. (2000)). The aerial survey covering the Brøgger peninsula took place on 17
July 2008 at around noon, acquiring data over most of the northern part of Brøggerhalvoya
(Figure B3). A Dornier Do228 aircraft from the German Aerospace Center (DLR) was used
for the survey, flying at an altitude of ~2,800 m. A comprehensive description of the initial
results is given by Hauber et al. (2011a, 2011b).

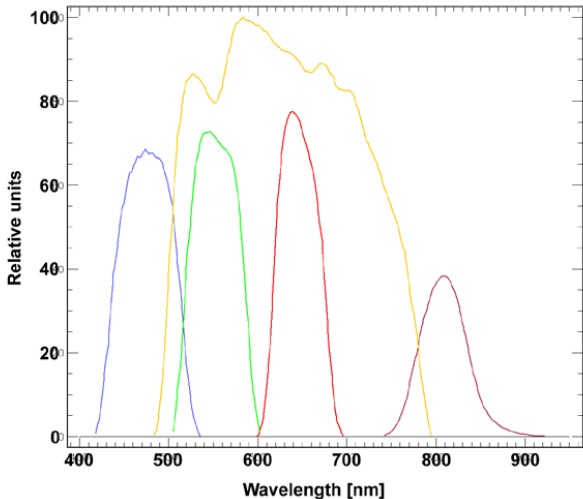

**Figure B2.** Spectral response of HRSC-AX panchromatic color filters.



Data processing from the raw images to the final data products, including digital photogram-

metric processing, was performed with the VICAR (Video Image Communication and Re-

trieval; http://www.mipl.jpl.nasa.gov/external/vicar.html) software developed at the JPL (Jet

Propulsion Laboratory, Pasadena, USA) and the DLR. We provide HRSC-AX data in the

form of a digital elevation model (DEM) and as individual channels (panchromatic nadir

channel, red, green, and blue color channels, and CIR false-color channels). The CIR (color-

infrared) channels are computed by merging a false-color image (an RGB image where R, G,

and B correspond to the original infrared, red, and green channels, respectively) with the nadir

channel. Table B1 lists important key properties of the individual HRSC-AX image files.

Metadata for HRSC-AX data are contained in "image labels", which we provide as two XML

files (one for the panchromatic and (false) color images, and one for the DEM). The label

entries consist of keyword-value pairs; essential keywords are defined below in Table B2. The

elevations recorded in the DEM are ellipsoid heights, i.e. they are not computed with respect

to a geoid but to a mathematically defined reference surface, which is a rotational ellipsoid

with the equatorial A and B axes both having a radius of 6378.14 km and the polar C axis

having a radius of 3356.75 km. This results in an offset of about 36.5 m with respect to geoid

heights, i.e. sea Level in the HRSC-AX DEM is not at 0 m, but at ~36.5 m.

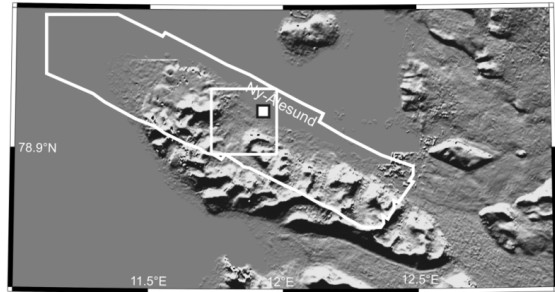



**Figure B3.** Context map of Brøgger peninsula, with the thick white outline showing the total coverage of the HRSC-AX survey and with the white square indicating the location of the image tile provided in this publication (base map: hillshade version of ASTER DEM).

| Image name | Number of lines | Number of samples | Ground pixel size |
|---|---|---|---|
| 430-8765_5.0x5.0km.pan | 25,000 | 25,000 | 0.2 m |
| 430-8765_5.0x5.0km.dsm | 10,000 | 10,000 | 0.5 m |
| 430-8765_5.0x5.0km.re | 25,000 | 25,000 | 0.2 m |
| 430-8765_5.0x5.0km.gr | 25,000 | 25,000 | 0.2 m |
| 430-8765_5.0x5.0km.bl | 25,000 | 25,000 | 0.2 m |
| 430-8765_5.0x5.0km.cir_re | 25,000 | 25,000 | 0.2 m |
| 430-8765_5.0x5.0km.cir_gr | 25,000 | 25,000 | 0.2 m |
| 430-8765_5.0x5.0km.cir_bl | 25,000 | 25,000 | 0.2 m |


**Table B1.** Image size and pixel size of individual HRSC-AX images.





| Name | Definition | Dimension | Type | Label Group |
|---|---|---|---|---|
| File_Name | Name of the data file | | string | |
| FORMAT | Image format (BYTE: 8 bit per pixel, HALF: 16 bit per pixel; REAL: 32 bit per pixel) | | string | |
| TYPE | Type of data file | | string | |
| ORG | Order of image file (BSQ = band sequential) | | string | |
| NL | Number of lines | | int | |
| NS | Number of samples | | int | |
| NB | Number of bands | | int | |
| TARGET_NAME | name of the target | | string | MAP |
| A_AXIS_RADIUS | The a_axis_radius element provides the value of the semimajor axis of the ellipsoid that defines the approximate shape of a target body. 'A' is usually in the equatorial plane. | km | real | MAP |
| B_AXIS_RADIUS | The b_axis_radius element provides the value of the intermediate axis of the ellipsoid that defines the approximate shape of a target body. 'B' is usually in the equatorial plane. | km | real | MAP |
| C_AXIS_RADIUS | The c_axis_radius element provides the value of the c_axis of a solar system body. For tri-axial ellipsoidal objects, the c_axis is the semiminor axis of the ellipsoid which defines the approximate shape of the body. | km | real | MAP |
| BODY_LONG_AXIS_ LONGITUDE | The BODY_LONG_AXIS_LONGITUDE element represents the offset between the longest axis of the triaxial ellipsoid used to model a body and the prime meridian of the body. Its value is the sum of the offset added to the prime meridian. This term is the positi | deg | real | MAP |
| CARTESI- AN_AZIMUTH | The cartesian_azimuth element provides the clockwise rotation, in degrees, of the line and sample coordinates with respect to the center of the pixel at the map projection origin (*i.e.* where line_projection_offset and sample_projection_offset are measured). | deg | real | MAP |
| CENTER_LATITUDE | The center_latitude element provides a reference latitude for certain map projections. In many projections, the center_latitude along with the center_longitude defines the point or | deg | real | MAP |



| Name | Definition | Dimension | Type | Label Group |
|------|-----------|-----------|------|-------------|
| | tangency between the sphere of the planet and the plane of the projection. | | | |
| CEN-TER_LONGITUDE | The center_longitude element provides a reference longitude for certain map projections. In many projections, the center_longitude along with the center_latitude defines the point or tangency between the sphere of the planet and the plane of the projection. | deg | real | MAP |
| COORDINA-TE_SYSTEM_NAME | Defines whether the CENTER_ LAT-ITUDE is geocentric or geodetic. | | string | MAP |
| LINE_PROJECTION_ OFFSET | The line_projection_offset element provides the line offset value of the map projection origin position from the center of the pixel at line and sample position 1,1 (line and sample 1,1 is considered the upper left corner of the digital array). | pixel | real | MAP |
| SAMP-LE_PROJECTION_OF FSET | The sample_projection_offset element provides the sample offset value of the map projection origin position from the center of the pixel line and sample 1,1 (line and sample 1,1 is considered the upper left corner of the digital array). Note that the posi | pixel | real | MAP |
| MAP_PROJECTION_ TYPE | The map_projection_type element identifies the type of projection characteristic of a given map. | | string | MAP |
| MAP_SCALE | The map_scale element identifies the scale of a given map. The scale is defined as the ratio of the actual distance between two points on the surface of the target body to the distance between the corresponding points on the map.  The map_scale references | km pixel$^{-1}$ | real | MAP |
| POSITI-VE_LONGITUDE_DIR ECTION | The positive_longitude_direction element identifies the direction of longitude (e.g. EAST, WEST) for a planet. The IAU definition for direction of positive longitude is adopted. | | string | MAP |
| SPHERI-CAL_AZIMUTH | One of three Euler angles (the others are center_latitude and center_longitude) that define the pre-mapping orientation of the planetary sphere for any spherical projection. | | real | MAP |
| DTM_RANGE | indicates at which minimum or maximum value the elevations in the DTM raster file have been cut-off | | real (2) | H |





| Name | Definition | Dimension | Type | Label Group |
|------|------------|-----------|------|-------------|
| DTM_A_AXIS_RADIUS | The DTM_A_AXIS_RADIUS element provides the value of the (+X) semi-axis length of the triaxial ellipsoid surface used as reference for DTM data. | km | real | DIGITAL_TERRAIN_MODEL |
| DTM_B_AXIS_RADIUS | The DTM_B_AXIS_RADIUS element provides the value of the (+Y) semi-axis length of the triaxial ellipsoid surface used as reference for DTM data. | km | real | DIGITAL_TERRAIN_MODEL |
| DTM_C_AXIS_RADIUS | The DTM_C_AXIS_RADIUS element provides the value of the (+Z) semi-axis length of the triaxial ellipsoid surface used as reference for DTM data. | km | real | DIGITAL_TERRAIN_MODEL |
| DTM_OFFSET | The DTM_OFFSET element provides the constant value by which a stored elevation value is shifted or displaced. | m | real | DIGITAL_TERRAIN_MODEL |
| DTM_SCALING_FACTOR | The DTM_SCALING_FACTOR element provides the constant value by which the stored elevation is multiplied | | real | DIGITAL_TERRAIN_MODEL |
| DTM_DESC | The DTM_DESC provides a free form, unlimited length character string that describes the DTM data. | | | DIGITAL_TERRAIN_MODEL |

**Table B2.** Description of metadata (keyword-value pairs) contained in the HRSC-AX image
headers.

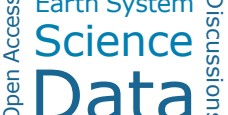

# Appendix C: Metadata description and pictures of climate, soil and permafrost stations, and instruments

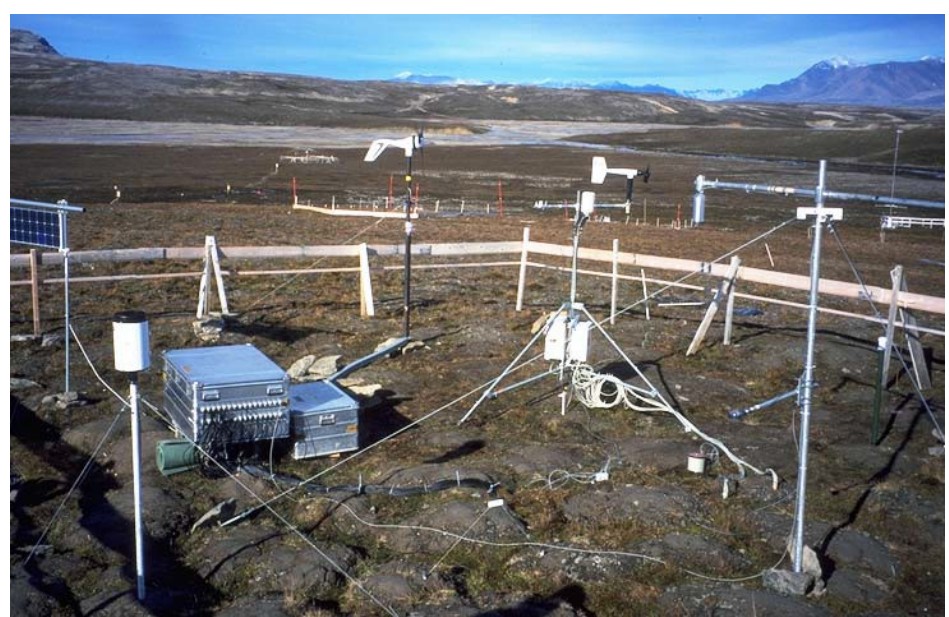

**Figure C1.** Bayelva climate station set up, Sept. 1998–Aug. 2009, UTM: 33 N 432100 8762992. Picture taken after installation in August 1998.





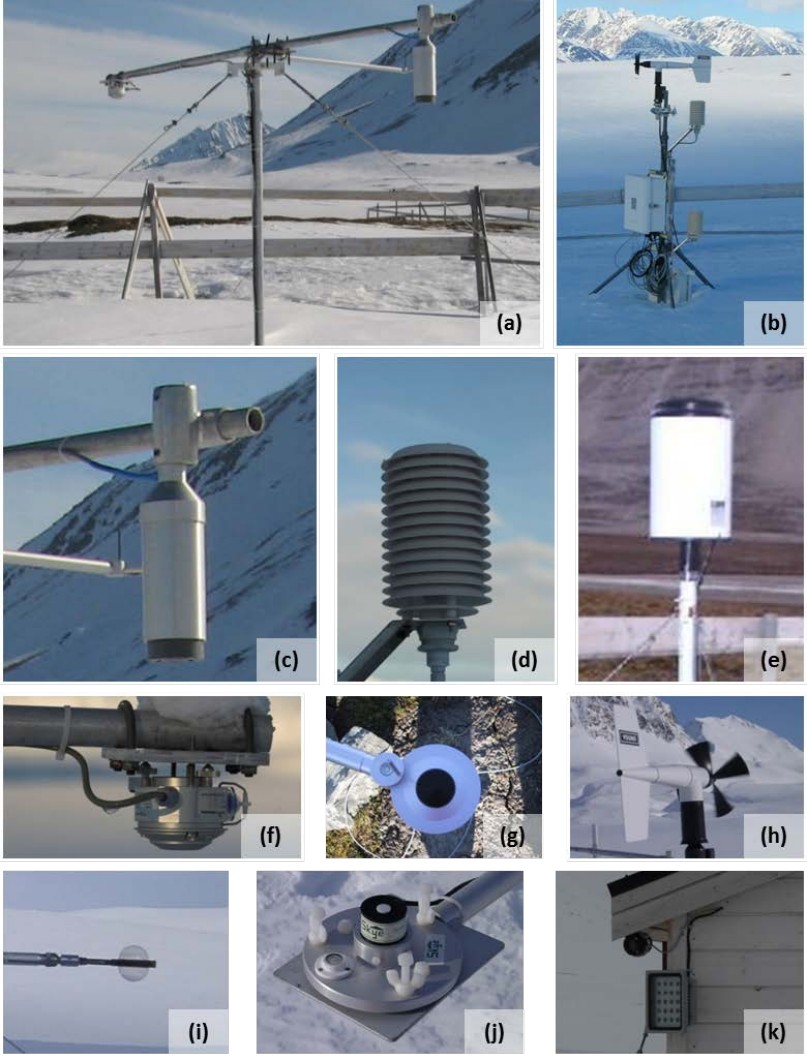

**Figure C2.** Climate station, Bayelva, Sept. 1998–Aug. 2009. **(a)** radiation/snow tower, **(b)** meteorological tower, **(c)** snowheight (from Sep 1999: 2 m. from Jul 2001, 1.6 m, from Sep 2003:1.45 m), **(d)** temperature/relative humidity (2 m) and temperature (1 m), **(e)** raingauge (1.68 m top of bucket above ground Level), **(f)** pyrgeometer (from May 2003: 2 m; from Sep 2003: 2 m) installed in May 2002, **(g)** net radiometer (from Apr 2000: 1.14 m; from Sep 2003: 1.6 m), **(h)** windspeed/-direction (3 m), **(i)** net radiation (from Apr 2000:1.14 m, from Sep 2003: 1.6 m), **(j)** pyranometer (2 m), **(k)** camera (2 m) with flash since Aug 2013; sensor details can be found in Table 2.



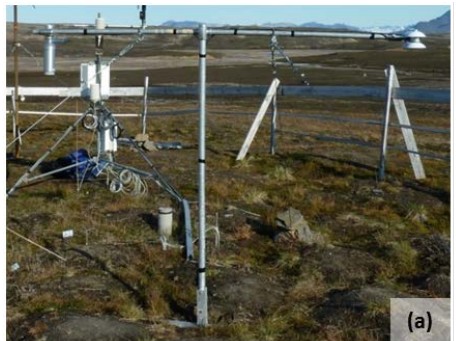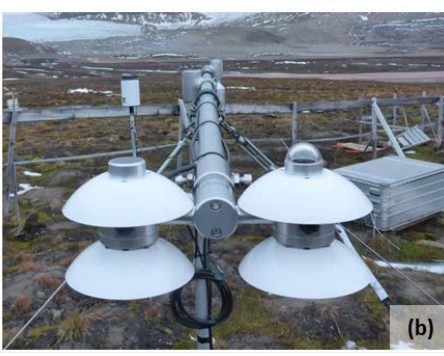


**Figure C3.** Climate station, Bayelva, Oct. 2009–present (with sensor heights), UTM: 33 N 432100 8762992. **(a)** radiation/snow tower, **(b)** four components radiation (1.56 m).

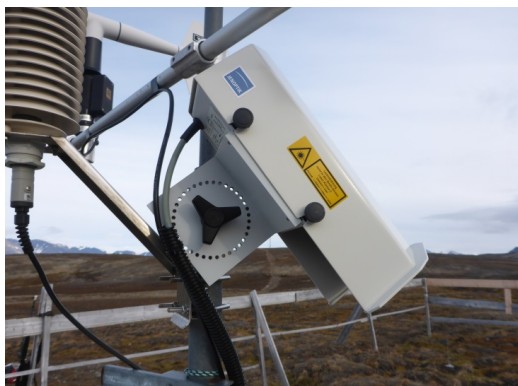

**Figure C4.** Laser snow depth sensor attached to the climate station, Bayelva, Aug. 2013– present, UTM: 33 N 432100 8762992.

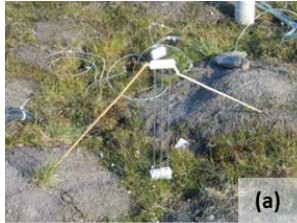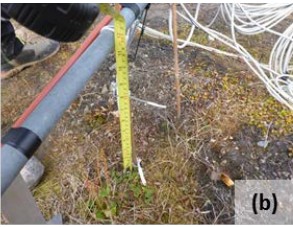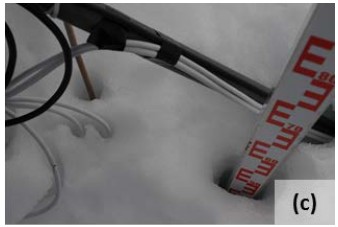

**Figure C5. (a)** Vertically installed time-domain reflectometry (TDR) air/ snow TDR probe (50 cm length) above the ground surface during summer at the Bayelva climate station (1999). During winter (fully snow covered), the dielectric number of snow is recorded. **(b)** PT100 temperature sensors in air (summer) at 4 and 20 cm above ground surface, **(c)** PT100



temperature sensors in snow (winter) with air gaps around cables and sticks; the TDR probe is completely covered with snow and not visible from the surface.






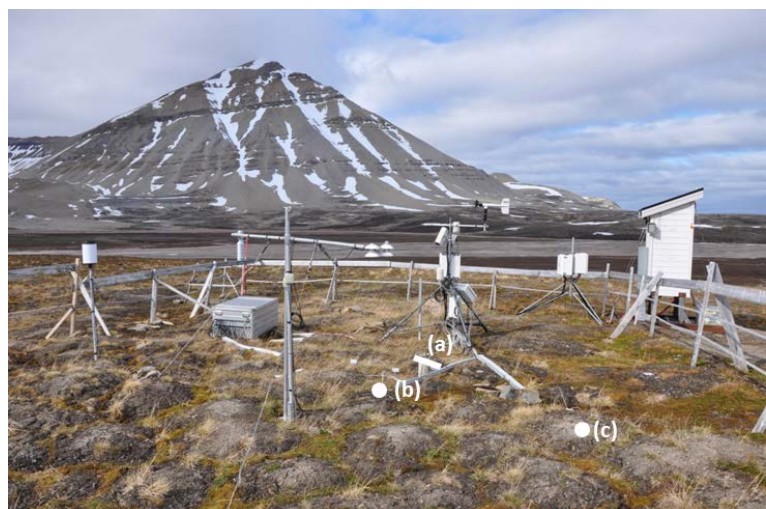

**Figure C6.** Location of soil profiles, Bayelva, 1998, 1999 and 2009, UTM: 33 N 432100 8762992. **(a)** 2D soil profile and station location 1998 (installed August 1998), **(b)** 1D soil station 1999 (installed July 1999), **(c)** 1D soil profile and station 2009 (installed August 2009).

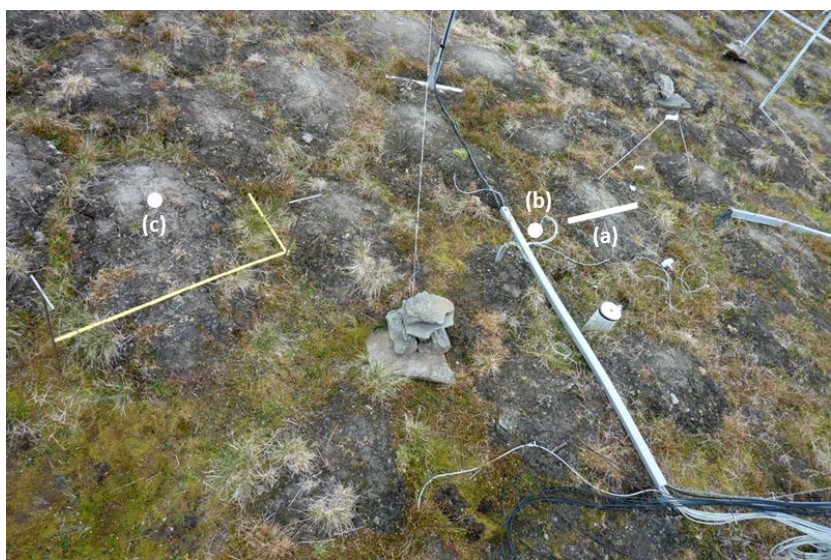

**Figure C7.** Soil profile and station location **(a)** 1998, **(b)** 1999, **(c)** 2009. Profile (b)  upper temperature sensors were affected by reindeer disturbance in September 2003. Furthermore,



the temperature sensors were affected by continuous frost heave by about 10 cm between
2003 and 2015.

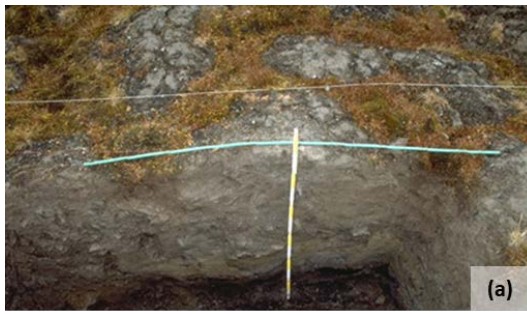
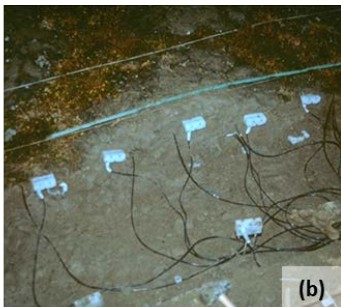

**Figure C8.** Bayelva soil station, Sept. 1998–January 2012, UTM: 33 N 432100 8762992. **(a)**
soil profile in non-sorted circle (down to 120 cm below ground surface), **(b)** TDR and
temperature probes. Soil texture and C, N, S data are listed in table F1.

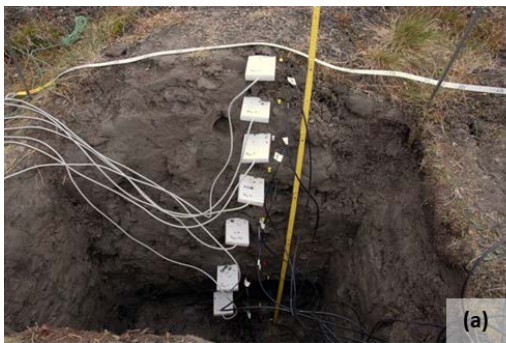
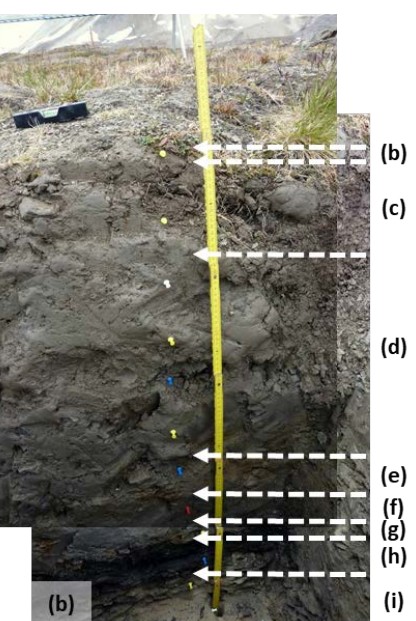

**Figure C9.** Bayelva soil station, Aug 2009–present, UTM: 33 N 432100 8762992. **(a)** TDR
and temperature probe installation; **b:** Soil characteristics of the non-sorted circle; **(b)** moss
layer (0–1 cm), **(c)** loam with roots (1–15 cm), **(d)** loam with few rocks (15–55 cm), **(e)** loam
with rocks (55–65 cm), **(f)** dark loam with few rocks (65–77 cm), **(g)** rocky layer with loam



(89–95 cm), **(h)** coal layer (81–95 cm), **(i)** rocky layer (95–119 cm). Soil data for texture and
C, N, S is listed in table F1.

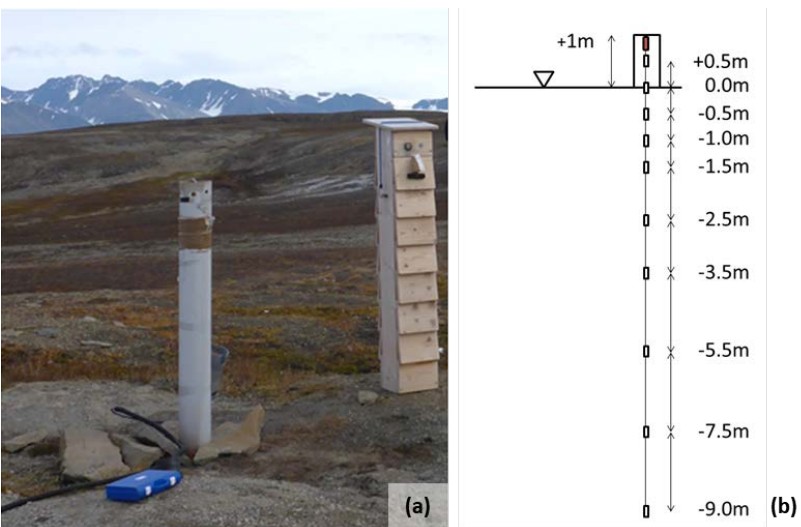

**Figure C10.** Borehole, Bayelva, drilled in March 2009 (total depth of 9.3 m) and
instrumented in Aug 2009–present, UTM: 33 N 432118 8762978.  (a) Borehole site with
PVC casing outside the borehole and wooden ventilation shield removed; setup (b) with a
Geoprecision temperature chain with probes at depths of 0.5 above ground surface, and 0, 0.5,
1.0, 1.5, 2.5, 3.5, 5.5, 7.5, 9 m below ground surface was used May 2009–September 2015
and May 2016–present.



# Appendix D: Calculation and correction of soil and meteorological parameters

**D1 Calculation of soil volumetric liquid water content using TDR**

The apparent dielectric numbers were converted into liquid water content ($\theta_l$) using the semi-empirical mixing model in (Roth et al., 1990). Frozen soil is treated as a four-phase porous medium composed of a solid (soil) matrix and interconnected pore spaces filled with water, ice and air.

The TDR method measures the ratio of apparent to physical probe rod length ($\frac{L_a}{L}$) which gives the square root of dielectric number ($\varepsilon_b$).

The bulk dielectric number is then calculated from the volumetric fractions and the dielectric numbers of the four phases (details in Appendix A) using

$$\varepsilon_b = [\theta_l \varepsilon_l^{\alpha} + \theta_i \varepsilon_i^{\alpha} + \theta_s \varepsilon_s^{\alpha} + \theta_a \varepsilon_a^{\alpha}]^{\frac{1}{\alpha}} \qquad (D1)$$

A value of 0.5 was used for $\alpha$.

It is not possible to distinguish between changes in the liquid water content and changes in the ice content with only one measured parameter ($\varepsilon_b$). Equation D1 was therefore rewritten in terms of the total water content ($\theta_{tot}$) and the porosity ($\Phi$) as

$$\theta_i = \theta_{tot} - \theta_l \qquad (D2)$$

We remark that equation D2 assumes the densities of liquid and frozen water to be the same. This is clearly wrong for free phases and probably also in the pore space of soils. However, the density ratio can be absorbed into the dielectric number $\varepsilon_i$, which we do in the following. The resulting fluctuation of $\varepsilon_i$ is presumed to be small compared to other uncertainties.

$$\theta_s = 1 - \phi \qquad (D3)$$

and




$$\theta_a = \phi - \theta_l - \theta_i = \phi - \theta_{tot} \qquad (D4)$$

to obtain the equation

$$\varepsilon_b = [\theta_l \varepsilon_l^\alpha + (\theta_{tot} - \theta_l)\varepsilon_i^\alpha + (1 - \phi)\varepsilon_s^\alpha + (\phi - \theta_{tot})\varepsilon_a^\alpha]^{\frac{1}{\alpha}} \qquad (D5)$$

For temperatures above a threshold freezing temperature ($T > T_f$), all water is assumed to be unfrozen ($\theta_{tot}$ equals $\theta_l$). Equation D5 then reduces to:

$$\theta_l(T) = \frac{\varepsilon_b^\alpha - \varepsilon_s^\alpha + \phi(\varepsilon_s^\alpha - \varepsilon_a^\alpha)}{\varepsilon_l^\alpha - \varepsilon_a^\alpha} \quad if \; T > T_f \qquad (D6)$$

For temperatures equal to or below the threshold freezing temperature ($T \leq T_f$) it was assumed

that the total water content ($\theta_{tot}$) remained constant and only the ratio between volumetric liquid water content ($\theta_l$) and volumetric ice content ($\theta_i$) changed. This is a rather bold assumption as freezing can lead to high gradients of matric potential, as well as to moisture redistribution. However, since the dielectric number of ice is much smaller than the dielectric number of liquid water, the error in liquid water content measurements is still acceptable

(which is not the case for ice content measurements). Under these assumptions we obtained the following equation for calculating the liquid water content of a four-phase mixture:

$$\theta_l(T \leq T_f) = \frac{\varepsilon_b^\alpha - \varepsilon_s^\alpha + \phi(\varepsilon_s^\alpha - \varepsilon_a^\alpha) + \theta_{tot}(\varepsilon_a^\alpha - \varepsilon_i^\alpha)}{\varepsilon_l^\alpha - \varepsilon_i^\alpha} \qquad (D7)$$

The error of the volumetric water content measurements using TDR probes was estimated to be between 2 and 5 %, and the precision to be better than 0.5 % (Boike and Roth, 1997).

The availability of reliable temperature data is crucial in this approach. The liquid water

content is first calculated for all times when the soil temperature was above the freezing threshold, using Equation D5. When the soil temperature was below the freezing threshold the water content immediately prior to the onset of freezing was determined and used as total water content ($\theta_{tot}$) for calculating the liquid water content during the frozen interval with Equation D7.




Since water in a porous medium does not necessarily freeze at 0 °C but at a temperature that

depends on the soil type and water content, estimating the threshold temperature is a crucial

part of this approach. If the freezing characteristic curve is known for the material then the

threshold temperature can be determined from the soil liquid water content. To avoid

interpretations of frequent freezing and thawing due to soil temperature measurement errors,

short-term temperature fluctuations were smoothed by calculating the mean of a moving

window with an adjustable width. The smoothed temperatures were then used to trigger the

switch from one equation to the other, rather than using the original temperature time series.

**D2 Calculation of soil temperatures using raw data and the Steinhart-Hart equation**

Temperatures were calculated from the raw voltage data using the Steinhart-Hart equation

(Steinhart and Hart, 1968) and sensor calibration at 0 °C (Appendix C). The thermistors were

calibrated at 0 °C prior to installation using a de-ionized water-ice mixture, from which a

thermistor-specific offset ($\delta_O$) for the Steinhart-Hart equation was obtained using

$$\frac{1}{T} = 1.28 * 10^{-3} + 2.37 * 10^{-4} * \ln(R_T - \delta_o) + 9.06 * 10^{-8} * (\ln(R_T - \delta_o))^3 \qquad \text{(D8)}$$

where $R_T$ is the measured resistance and $\delta_o$ is the resistance offset at 0 °C. The estimated pre-

cision was 2.4 x $10^{-4}$ °C at 0 °C with an absolute error of less than 0.1 °C, in the temperature

range from -40 to +40 °C.

**D3 Correction of net radiation correction for wind speed**

**03 Sep 1998–18 Apr 2000**

Campbell Scientific Ltd. Q-7 Net Radiometer running from 01 Sep 1998–18 Apr 2000.

For positive fluxes (NetRad_raw > 0 W m$^{-2}$) the correction formula was:





$$NetRad = NetRad\_raw * 1.159 * \left(1 + \left(\frac{0.066 * 0.2 * wind\_v}{0.066 + (0.2 * wind\_v)}\right)\right) \quad (D9)$$

For negative fluxes (NetRad_raw $< 0$ W m$^{-2}$) the correction formula was:

$$NetRad = NetRad\_raw * 0.9065 * (0.00174 * wind\_v + 0.99755) \quad (D10)$$

**18 Apr 2000–20 May 2002**

Kipp & Zonen NR-LITE Net Radiometer running from 18 Apr 2000–20 May 2002.

The correction formula was:

$$NetRad = NetRad\_raw * 68.027 * (1 + (0.0082 * wind\_v)) \quad (D11)$$

**20 May 2002–15 Sep 2003**

Campbell Scientific Ltd. Q-7 Net Radiometer running from 20 May 2002–15 Sep 2003.

For positive fluxes (NetRad_raw $> 0$ W m$^{-2}$) the correction formula was:

$$NetRad = NetRad\_raw * 7.71 * \left(1 + \left(\frac{0.066 * 0.2 * wind\_v}{0.066 + (0.2 * wind\_v)}\right)\right) \quad (D12)$$

For negative fluxes (NetRad_raw $< 0$ W m$^{-2}$) the correction formula was:

$$NetRad = NetRad\_raw * 11.66 * ((0.00174 * wind\_v) + 0.99755) \quad (D13)$$

**15 Sep 2003–15 Aug 2009**

Kipp & Zonen NR-LITE Net Radiometer running from 15 Sep 2003–14 Aug 2009.

The correction formula was:

$$NetRad = NetRad\_raw * 68.0272 * (1 + (0.0082 * wind\_v)) \quad (D14)$$

**D4 Snow depth correction for air temperature**

The raw distance $D_{sn\_raw}$ obtained from the SR50 sonic sensor (Campbell Scientific Ltd.) is

calculated using the speed of sound at 0°C and is corrected with the air temperature at 2m

height  using the formula provided by the manufacturer:



$$Dsn = Dsn\_raw * \sqrt{\frac{T}{273.15}} \qquad (D15)$$





## Appendix E: Description and data of snow profiles

| Location | Date | Air temperature | Total snow depth | Minimum height above ground surface | Maximum height above ground surface | Snow temperature | Dielectric number | Density | Grain shape | Grain size | Hand hardness index |
|---|---|---|---|---|---|---|---|---|---|---|---|
| | [YYYY-MM-DD] | [°C] | [cm] | [cm] | [cm] | [°C] | | [g/cm³] | | [mm] | [1-6] |
| Bayelva | 2000-04-20 | | 80 | 0 | 0 | -9.6 | | | | | |
| | 2000-04-20 | | 80 | 0 | 10 | | | | DH | 3–4 | 4 |
| | 2000-04-20 | | 80 | 10 | 10 | -10 | | | | | |
| | 2000-04-20 | | 80 | 10 | 29 | | | 403 | | 1–3 | 4 |
| | 2000-04-20 | | 80 | 20 | 20 | -10.9 | | | | | |
| | 2000-04-20 | | 80 | 29 | 52 | | | 390 | PPip | 2 | 4 |
| | 2000-04-20 | | 80 | 30 | 30 | -11.4 | | | | | |
| | 2000-04-20 | | 80 | 40 | 40 | -12 | | | | | |
| | 2000-04-20 | | 80 | 50 | 50 | -12.6 | | | | | |
| | 2000-04-20 | | 80 | 52 | 80 | | | 354 | | <1 | 3 |
| | 2000-04-20 | | 80 | 55 | 55 | -13.7 | | | | | |
| | 2000-04-20 | | 80 | 60 | 60 | -14.3 | | | | | |
| | 2000-04-20 | | 80 | 65 | 65 | -15 | | | | | |
| | 2000-04-20 | | 80 | 70 | 70 | -15.5 | | | | | |
| | 2000-04-20 | | 80 | 75 | 75 | -15.3 | | | | | |
| | 2000-04-20 | | 80 | 80 | 80 | -13.6 | | | | | |
| Bayelva | 2006-05-11 | | 37 | 0 | 8 | | | | IFbi | | 6 |
| | 2006-05-11 | | 37 | 8 | 19 | | | | | 1–2 | |
| | 2006-05-11 | | 37 | 10 | 10 | -1.9 | | | | | |

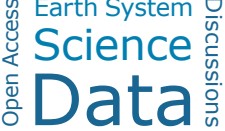

| Location | Date | Air temperature | Total snow depth | Minimum height above ground surface | Maximum height above ground surface | Snow temperature | Dielectric number | Density | Grain shape | Grain size | Hand hardness index |
|---|---|---|---|---|---|---|---|---|---|---|---|
| | [YYYY-MM-DD] | [°C] | [cm] | [cm] | [cm] | [°C] | | [g/cm³] | | [mm] | [1-6] |
| | 2006-05-11 | | 37 | 13 | 13 | -2.2 | 3.2 | | | | |
| | 2006-05-11 | | 37 | 18 | 18 | -2.3 | | | | | |
| | 2006-05-11 | | 37 | 19 | 19 | | | | IFil | | 6 |
| | 2006-05-11 | | 37 | 20 | 27 | | | | | | |
| | 2006-05-11 | | 37 | 23 | 23 | -2.8 | 2.2 | | | | |
| | 2006-05-11 | | 37 | 25 | 25 | | | 333 | | | |
| | 2006-05-11 | | 37 | 27 | 27 | | 2.1 | | | | |
| | 2006-05-11 | | 37 | 28 | 28 | -4.0 | | | | | |
| | 2006-05-11 | | 37 | 33 | 33 | -4.6 | 2.2 | | | | |
| | 2006-05-11 | | 37 | 37 | 37 | -4.7 | 2.2 | | | | |
| Ny-Ålesund | 2006-05-12 | -2.7 | 30 | 0 | 11 | | | | IFbi | | 6 |
| | 2006-05-12 | -2.7 | 30 | 11 | 24 | -0.6 | 2.3 | | | | |
| | 2006-05-12 | -2.7 | 30 | 12 | 12 | | | | | | |
| | 2006-05-12 | -2.7 | 30 | 16 | 16 | -1.4 | 2.2 | 304 | | | |
| | 2006-05-12 | -2.7 | 30 | 21 | 21 | -2.3 | 2.2 | | | | |
| | 2006-05-12 | -2.7 | 30 | 24 | 30 | | | | | | |
| | 2006-05-12 | -2.7 | 30 | 26 | 26 | -3.0 | 2.2 | 324 | | | |
| | 2006-05-12 | -2.7 | 30 | 30 | 30 | -2.7 | 2.1 | | | | |
| Bayelva | 2016-05-09 | | 64 | 0 | 15 | | | | IFbi | | 6 |
| (left) | 2016-05-09 | | 64 | 15 | 15 | -2.6 | | | | | |
| (left) | 2016-05-09 | | 64 | 18 | 18 | | 1.3 | | | | |
| (left) | 2016-05-09 | | 64 | 19 | 19 | | 1.3 | | | | |
| (left) | 2016-05-09 | | 64 | 20 | 20 | -2.2 | | | | | |





| Location | Date | Air temperature | Total snow depth | Minimum height above ground surface | Maximum height above ground surface | Snow temperature | Dielectric number | Density | Grain shape | Grain size | Hand hardness index |
|---|---|---|---|---|---|---|---|---|---|---|---|
| | [YYYY-MM-DD] | [°C] | [cm] | [cm] | [cm] | [°C] | | [g/cm³] | | [mm] | [1-6] |
| (left) | 2016-05-09 | | 64 | 23 | 27 | | | 275 | | | |
| (left) | 2016-05-09 | | 64 | 24 | 28 | | | | IFil | | 6 |
| (left) | 2016-05-09 | | 64 | 25 | 25 | -2 | 1.2 | | | | |
| (left) | 2016-05-09 | | 64 | 30 | 30 | -1.8 | 1.2 | | | | |
| (left) | 2016-05-09 | | 64 | 35 | 35 | -1.3 | 1.3 | 297 | | | |
| (left) | 2016-05-09 | | 64 | 37 | 41 | | | | | | |
| (left) | 2016-05-09 | | 64 | 40 | 40 | -0.9 | 1.2 | | | | |
| (left) | 2016-05-09 | | 64 | 43 | 47 | | | | IFil | | 6 |
| (left) | 2016-05-09 | | 64 | 45 | 45 | -0.5 | 1.2 | | | | |
| (left) | 2016-05-09 | | 64 | 45 | 49 | | | 335 | | | |
| (left) | 2016-05-09 | | 64 | 50 | 50 | -0.1 | 1.2 | | | | |
| (left) | 2016-05-09 | | 64 | 53 | 57 | | | 361 | | | |
| (left) | 2016-05-09 | | 64 | 55 | 55 | 0.1 | 1.3 | | | | |
| (left) | 2016-05-09 | | 64 | 55 | 56 | | | | IFil | | 6 |
| (left) | 2016-05-09 | | 64 | 60 | 60 | 0.1 | 1.4 | | | | |
| (left) | 2016-05-09 | | 64 | 60 | 64 | | | 388 | | | |
| Bayelva | 2016-05-09 | | 64 | 0 | 15 | | | | IFbi | | 6 |
| (right) | 2016-05-09 | | 64 | 15 | 15 | -2.6 | | | | | |
| (right) | 2016-05-09 | | 64 | 20 | 20 | -2.2 | | | | | |
| (right) | 2016-05-09 | | 64 | 23 | 27 | | | 294 | | | |
| (right) | 2016-05-09 | | 64 | 24 | 28 | | | | IFil | | 6 |
| (right) | 2016-05-09 | | 64 | 25 | 25 | -2 | 1.3 | | | | |
| (right) | 2016-05-09 | | 64 | 30 | 30 | -1.8 | 1.2 | | | | |



| Location | Date | Air temperature | Total snow depth | Minimum height above ground surface | Maximum height above ground surface | Snow temperature | Dielectric number | Density | Grain shape | Grain size | Hand hardness index |
|---|---|---|---|---|---|---|---|---|---|---|---|
| | [YYYY-MM-DD] | [°C] | [cm] | [cm] | [cm] | [°C] | | [g/cm³] | | [mm] | [1-6] |
| (right) | 2016-05-09 | | 64 | 35 | 35 | -1.3 | 1.3 | | | | |
| (right) | 2016-05-09 | | 64 | 37 | 41 | | | 268 | | | |
| (right) | 2016-05-09 | | 64 | 40 | 40 | -0.9 | 1.2 | | | | |
| (right) | 2016-05-09 | | 64 | 43 | 47 | | 1.2 | | IFil | | 6 |
| (right) | 2016-05-09 | | 64 | 45 | 45 | -0.5 | | | | | |
| (right) | 2016-05-09 | | 64 | 45 | 49 | | | 319 | | | |
| (right) | 2016-05-09 | | 64 | 50 | 50 | -0.1 | 1.5 | | | | |
| (right) | 2016-05-09 | | 64 | 53 | 57 | | | 405 | | | |
| (right) | 2016-05-09 | | 64 | 55 | 55 | 0.1 | | | | | |
| (right) | 2016-05-09 | | 64 | 55 | 56 | | 1.5 | | IFil | | 6 |
| (right) | 2016-05-09 | | 64 | 60 | 60 | 0.1 | 1.4 | | | | |
| (right) | 2016-05-09 | | 64 | 60 | 64 | | | 372 | IFbi | | 6 |
| Bayelva | 2016-05-10 | 1 | 75 | 0 | 20 | | | | | | |
| (left) | 2016-05-10 | 1 | 75 | 20 | 44 | | | | | | 3 |
| (left) | 2016-05-10 | 1 | 75 | 22 | 22 | -2.5 | | | | | |
| (left) | 2016-05-10 | 1 | 75 | 24 | 28 | | | 390 | | | |
| (left) | 2016-05-10 | 1 | 75 | 25 | 25 | -2.4 | 1.3 | | | | |
| (left) | 2016-05-10 | 1 | 75 | 28 | 34 | | | 359 | | | |
| (left) | 2016-05-10 | 1 | 75 | 30 | 30 | -2.3 | 1.3 | | | | |
| (left) | 2016-05-10 | 1 | 75 | 30 | 34 | | 1.3 | 295 | | | |
| (left) | 2016-05-10 | 1 | 75 | 35 | 35 | -1.9 | 1.3 | | | | |
| (left) | 2016-05-10 | 1 | 75 | 38 | 42 | | | 289 | | | |
| (left) | 2016-05-10 | 1 | 75 | 40 | 40 | -1.6 | 1.3 | | | | |



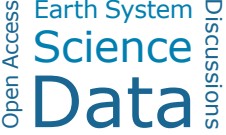

| Location | Date | Air temperature | Total snow depth | Minimum height above ground surface | Maximum height above ground surface | Snow temperature | Dielectric number | Density | Grain shape | Grain size | Hand hardness index |
|---|---|---|---|---|---|---|---|---|---|---|---|
| | [YYYY-MM-DD] | [°C] | [cm] | [cm] | [cm] | [°C] | | [g/cm³] | | [mm] | [1-6] |
| (left) | 2016-05-10 | 1 | 75 | 40 | 44 | | | | IFil | | 6 |
| (left) | 2016-05-10 | 1 | 75 | 44 | 50 | | | | IFil | | 6/2 |
| (left) | 2016-05-10 | 1 | 75 | 45 | 45 | -1.1 | | | | | |
| (left) | 2016-05-10 | 1 | 75 | 46 | 50 | | | 400 | | | |
| (left) | 2016-05-10 | 1 | 75 | 48 | 50 | | | | IFil | | 6 |
| (left) | 2016-05-10 | 1 | 75 | 50 | 50 | -0.9 | 1.3 | | | | |
| (left) | 2016-05-10 | 1 | 75 | 50 | 64 | | | | | | 1 |
| (left) | 2016-05-10 | 1 | 75 | 52 | 58 | | | 245 | | | |
| (left) | 2016-05-10 | 1 | 75 | 55 | 55 | -0.6 | 1.3 | | | | |
| (left) | 2016-05-10 | 1 | 75 | 58 | 62 | | | 275 | | | |
| (left) | 2016-05-10 | 1 | 75 | 60 | 60 | -0.1 | 1.2 | | | | |
| (left) | 2016-05-10 | 1 | 75 | 62 | 65 | | | | IFil | | 6 |
| (left) | 2016-05-10 | 1 | 75 | 64 | 68 | | | 438 | | | |
| (left) | 2016-05-10 | 1 | 75 | 65 | 65 | 0.0 | 1.4 | | | | |
| (left) | 2016-05-10 | 1 | 75 | 65 | 75 | | | | | | 3 |
| (left) | 2016-05-10 | 1 | 75 | 68 | 72 | | | 453 | | | |
| (left) | 2016-05-10 | 1 | 75 | 70 | 70 | 0.1 | 1.5 | | | | |
| (left) | 2016-05-10 | 1 | 75 | 72 | 72 | | 1.4 | | | | |
| (left) | 2016-05-10 | 1 | 75 | 72 | 76 | | | 414 | | | |
| (left) | 2016-05-10 | 1 | 75 | 73 | 73 | | 1.4 | | | | |
| (left) | 2016-05-10 | 1 | 75 | 74 | 74 | | 1.4 | | | | |
| (left) | 2016-05-10 | 1 | 75 | 75 | 75 | 0.1 | | | | | |
| Bayelva | 2016-05-10 | 1 | 75 | 0 | 20 | | | | IFbi | | 6 |



| Location | Date | Air temperature | Total snow depth | Minimum height above ground surface | Maximum height above ground surface | Snow temperature | Dielectric number | Density | Grain shape | Grain size | Hand hardness index |
|---|---|---|---|---|---|---|---|---|---|---|---|
| | [YYYY-MM-DD] | [°C] | [cm] | [cm] | [cm] | [°C] | | [g/cm³] | | [mm] | [1-6] |
| (right) | 2016-05-10 | 1 | 75 | 20 | 44 | | | | | | 3 |
| (right) | 2016-05-10 | 1 | 75 | 22 | 22 | -2.5 | | | | | |
| (right) | 2016-05-10 | 1 | 75 | 23 | 23 | | 1.4 | | | | |
| (right) | 2016-05-10 | 1 | 75 | 24 | 28 | | | 340 | | | |
| (right) | 2016-05-10 | 1 | 75 | 25 | 25 | -2.4 | | | | | |
| (right) | 2016-05-10 | 1 | 75 | 28 | 34 | | | 357 | | | |
| (right) | 2016-05-10 | 1 | 75 | 30 | 30 | -2.3 | 1.3 | | | | |
| (right) | 2016-05-10 | 1 | 75 | 30 | 34 | | | 362 | | | |
| (right) | 2016-05-10 | 1 | 75 | 35 | 35 | -1.9 | 1.3 | | | | |
| (right) | 2016-05-10 | 1 | 75 | 38 | 42 | | | 290 | | | |
| (right) | 2016-05-10 | 1 | 75 | 40 | 40 | -1.6 | 1.2 | | | | |
| (right) | 2016-05-10 | 1 | 75 | 40 | 44 | | | | IFil | | 6 |
| (right) | 2016-05-10 | 1 | 75 | 44 | 50 | | | | IFil | | 6/2 |
| (right) | 2016-05-10 | 1 | 75 | 45 | 45 | -1.1 | 1.3 | 393 | | | |
| (right) | 2016-05-10 | 1 | 75 | 46 | 50 | | | | | | |
| (right) | 2016-05-10 | 1 | 75 | 48 | 50 | | | | IFil | | 6 |
| (right) | 2016-05-10 | 1 | 75 | 50 | 50 | -0.9 | 1.2 | | | | |
| (right) | 2016-05-10 | 1 | 75 | 50 | 64 | | | | | | 1 |
| (right) | 2016-05-10 | 1 | 75 | 52 | 58 | | | 244 | | | |
| (right) | 2016-05-10 | 1 | 75 | 55 | 55 | -0.6 | 1.2 | | | | |
| (right) | 2016-05-10 | 1 | 75 | 58 | 62 | | | 316 | | | |
| (right) | 2016-05-10 | 1 | 75 | 60 | 60 | -0.1 | 1.2 | | | | |
| (right) | 2016-05-10 | 1 | 75 | 62 | 65 | | | | IFil | | 6 |

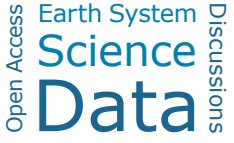

| Location | Date | Air temperature | Total snow depth | Minimum height above ground surface | Maximum height above ground surface | Snow temperature | Dielectric number | Density | Grain shape | Grain size | Hand hardness index |
|---|---|---|---|---|---|---|---|---|---|---|---|
| | [YYYY-MM-DD] | [°C] | [cm] | [cm] | [cm] | [°C] | | [g/cm³] | | [mm] | [1-6] |
| (right) | 2016-05-10 | 1 | 75 | 64 | 68 | | | 395 | | | |
| (right) | 2016-05-10 | 1 | 75 | 65 | 65 | 0.0 | 1.3 | | | | |
| (right) | 2016-05-10 | 1 | 75 | 65 | 75 | | | | | | 3 |
| (right) | 2016-05-10 | 1 | 75 | 68 | 72 | | | 421 | | | |
| (right) | 2016-05-10 | 1 | 75 | 70 | 70 | 0.1 | 1.4 | | | | |
| (right) | 2016-05-10 | 1 | 75 | 72 | 72 | | | | | | |
| (right) | 2016-05-10 | 1 | 75 | 72 | 76 | | | 392 | | | |
| (right) | 2016-05-10 | 1 | 75 | 73 | 73 | | 1.4 | | | | |
| (right) | 2016-05-10 | 1 | 75 | 74 | 74 | | | | | | |
| (right) | 2016-05-10 | 1 | 75 | 75 | 75 | 0.1 | | | | | |

**Table E1.** Snow data obtained from manual probing of snow profiles in spring 2000 (Bayelva), 2006 (Bayelva and Ny-Ålesund), and 2016 (Bayelva). Data include stratigraphy, temperature, dielectric number, density and information on dominant grain shape, type, and size, and hand hardness index according to international snow classification (Fierz et al., 2009). Left and right refer to the measurement positions within the profile shown in Figure E3.





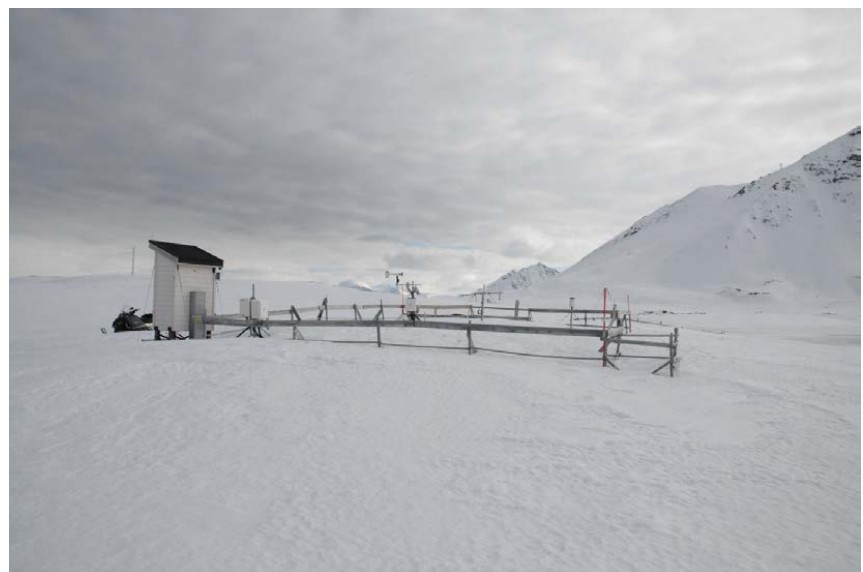

**Figure E10.** The Bayelva site covered by snow, 7 May 2016.

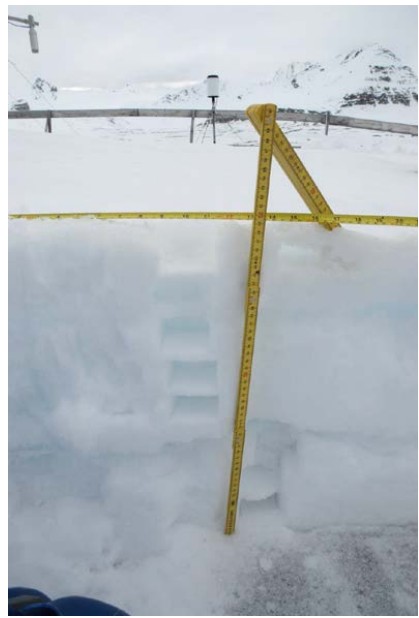

**Figure E11.** Snow profile taken on 9 May 2016, inside the Bayelva fenced area. The
pockets/holes are the result of snow removal using a snow density cutter (100 cm³). Snow



depth is 47 cm above a basal ice layer at the point of measurement but ranges between 47 and 49 cm. Measurements were taken to the left of the yellow folding rule (left and right profiles in Table E1). Ice layers are present 24–28 cm, 43–47 cm, and 55–56 cm above the basal ice layer, which is 15 cm thick. Further data on temperatures, dielectric numbers, and stratigraphy can be found in Table E1.

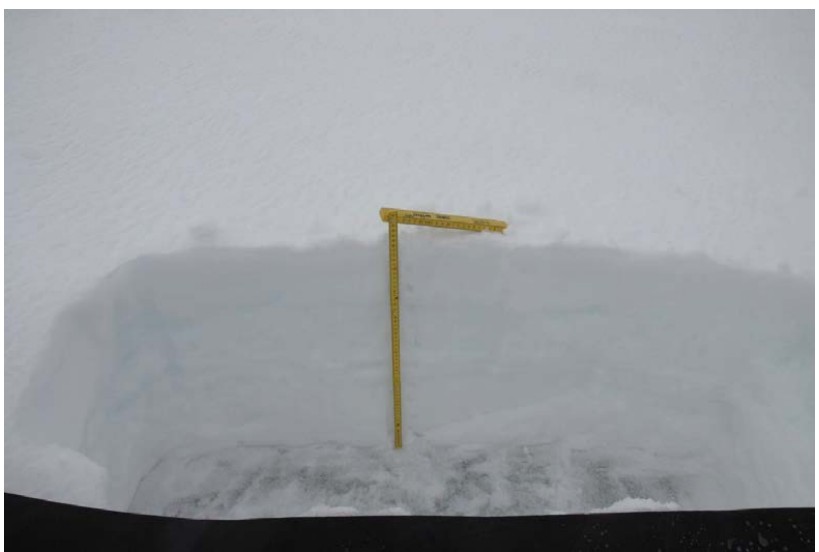


**Figure E12.** Snow pit, 10 May 2016. The basal ice layer is visible at the bottom of snow layer. Two profiles were measured one to the left and one to the right from the yellow meter stick (profile data named left and right in Table E1). Further data on temperatures, dielectric numbers, and stratigraphy can be found in table E1.





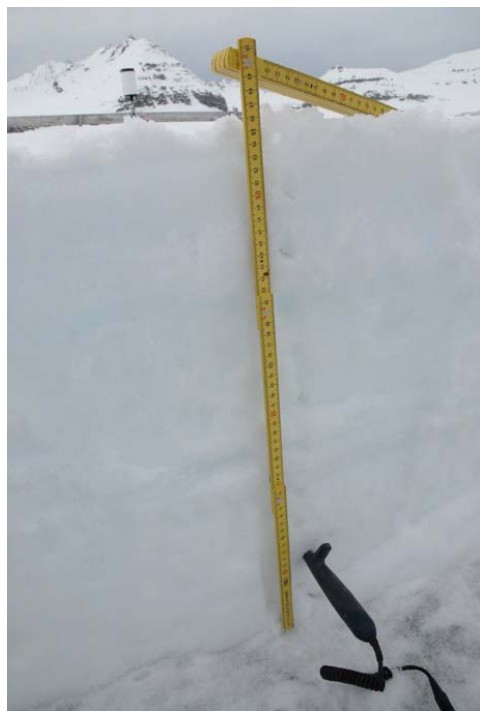

**Figure E13.** Measuring snow temperature in profile, 10 May 2016. Data can be found in table E1.



## Appendix F: Description and data of soil profiles

| Sample-ID | Date [YYYY] | Profile number | Depth below surface [cm] | Layer thickness [cm] | TOC [wt%] | SOCC [kg/m²] | CD$_b$ [kg/m³] | C [wt%] | N [wt%] | S [wt%] | Dry bulk density [g/cm³] | Average dry bulk density [g/cm³] | Grain size class |
|---|---|---|---|---|---|---|---|---|---|---|---|---|---|
| 1/26 | 1998 | 1 | 11 | 11 | 1.3 | 2.5 | 22.8 | 1.3 | < 0.1 | < 0.05 | | 1.7 | sZ |
| 1/50 | 1998 | 1 | 35 | 24 | 1.1 | 4.5 | 18.7 | 1.1 | < 0.1 | < 0.05 | | 1.7 | sM |
| 1/65 | 1998 | 1 | 50 | 15 | 0.7 | | | 0.8 | < 0.1 | < 0.05 | | 1.7 | |
| 1/76 | 1998 | 1 | 61 | 11 | 3.0 | 5.6 | 50.5 | 2.8 | < 0.1 | < 0.05 | | 1.7 | M |
| 1/84 | 1998 | 1 | 69 | 8 | 21.2 | | | 23.8 | 0.4 | 0.4 | | 1.7 | |
| 1/95 | 1998 | 1 | 80 | 11 | 4.4 | | | 4.3 | 0.1 | 0.3 | | 1.7 | |
| 2/12 | 1998 | 2 | 6 | 6 | 1.1 | 1.1 | 18.2 | 1.2 | < 0.1 | < 0.05 | | 1.7 | sZ |
| 2/37 | 1998 | 2 | 31 | 25 | 1.4 | 5.7 | 23.0 | 1.3 | < 0.1 | < 0.05 | | 1.7 | sZ |
| 2/57 | 1998 | 2 | 51 | 20 | 1.2 | 4.2 | 20.9 | 1.2 | < 0.1 | < 0.05 | | 1.7 | sM |
| 2/69 | 1998 | 2 | 63 | 12 | 1.1 | 2.2 | 18.2 | 1.1 | < 0.1 | < 0.05 | | 1.7 | M |
| 2/78 | 1998 | 2 | 72 | 9 | 2.9 | 4.4 | 48.6 | 2.8 | 0.1 | 0.1 | | 1.7 | M |
| 2/86 | 1998 | 2 | 80 | 8 | 29.4 | | | 34.3 | 0.6 | 0.6 | | 1.7 | |
| 2/88 | 1998 | 2 | 82 | 2 | 4.1 | | | 4.1 | 0.1 | 0.1 | | 1.7 | |
| 2/92 | 1998 | 2 | 86 | 4 | 3.8 | 2.6 | 63.9 | 3.7 | 0.1 | 0.2 | | 1.7 | C |
| 3/12 | 1998 | 3 | 10 | 10 | 0.7 | 1.2 | 12.2 | 0.8 | < 0.1 | < 0.05 | | 1.7 | sM |
| 3/41 | 1998 | 3 | 39 | 29 | 0.6 | 3.1 | 10.5 | 0.7 | < 0.1 | < 0.05 | | 1.7 | M |
| 3/72 | 1998 | 3 | 70 | 31 | 1.0 | | | 1.1 | < 0.1 | < 0.05 | | 1.7 | |
| 3/77 | 1998 | 3 | 75 | 5 | 2.7 | 2.3 | 46.4 | 2.7 | < 0.1 | < 0.05 | | 1.7 | M |
| 3/82 | 1998 | 3 | 80 | 5 | 5.1 | 4.3 | 85.9 | 6.6 | 0.2 | 0.2 | | 1.7 | C |




| Sample-ID | Date [YYYY] | Profile number | Depth below surface [cm] | Layer thickness [cm] | TOC [wt%] | SOCC [kg/m²] | $CD_b$ [kg/m³] | C [wt%] | N [wt%] | S [wt%] | Dry bulk density [g/cm³] | Average dry bulk density [g/cm³] | Grain size class |
|---|---|---|---|---|---|---|---|---|---|---|---|---|---|
| 3/92 | 1998 | 3 | 91 | 11 | 6.5 | 12.1 | 110.0 | 6.4 | 0.2 | 0.5 | | 1.7 | C |
| 4/16 | 1998 | 4 | 10 | 10 | 1.3 | 2.2 | 21.8 | 1.3 | <0.1 | <0.05 | | 1.7 | sM |
| 4/39 | 1998 | 4 | 33 | 23 | 1.0 | 4.1 | 17.7 | 1.0 | <0.1 | <0.05 | | 1.7 | M |
| 4/63 | 1998 | 4 | 57 | 24 | 0.8 | 3.1 | 12.9 | 0.8 | <0.1 | <0.05 | | 1.7 | M |
| 4/75 | 1998 | 4 | 69 | 12 | 0.7 | 1.3 | 11.2 | 0.8 | <0.1 | <0.05 | | 1.7 | M |
| 4/88 | 1998 | 4 | 82 | 13 | 3.0 | 6.6 | 51.0 | 2.9 | 0.1 | 0.1 | | 1.7 | M |
| 4/94 | 1998 | 4 | 88 | 6 | 3.7 | 3.7 | 62.1 | 3.4 | 0.1 | 0.1 | | 1.7 | M |
| 5/24 | 1998 | 5 | 11 | 11 | 2.0 | 3.8 | 34.3 | 2.0 | 0.1 | <0.05 | | 1.7 | sZ |
| 5/38 | 1998 | 5 | 25 | 14 | 1.2 | 2.8 | 19.7 | 1.0 | <0.1 | <0.05 | | 1.7 | sM |
| 5/63 | 1998 | 5 | 50 | 25 | 1.2 | 5.0 | 19.9 | 1.1 | <0.1 | <0.05 | | 1.7 | M |
| 5/75 | 1998 | 5 | 60 | 12 | 1.2 | | | 1.2 | <0.1 | <0.05 | | 1.7 | |
| 5/83 | 1998 | 5 | 70 | 8 | 1.3 | 1.8 | 21.9 | 1.8 | <0.1 | <0.05 | | 1.7 | M |
| 5/96 | 1998 | 5 | 83 | 13 | 3.3 | 7.3 | 56.3 | 3.7 | <0.1 | 0.1 | | 1.7 | sM |
| 277, 299, 442 | 2007 | 1 | 0 | 0 | | | | 1.3 | <0.1 | <0.05 | 0.8 | 1.5 | sZ |
| 432, 448, 470 | 2007 | 1 | 15 | 15 | 1.1 | 2.8 | 18.6 | | | | 1.7 | 1.5 | |
| 259, 280, 295 | 2007 | 1 | 30 | 15 | 0.7 | 1.9 | 12.8 | | | | 1.8 | 1.5 | |
| 267, 276, 453 | 2007 | 1 | 45 | 15 | 2.2 | 5.1 | 33.9 | | | | 1.5 | 1.5 | |
| 454, 484, 505 | 2007 | 1 | 60 | 15 | 0.8 | 2.0 | 13.6 | | | | 1.7 | 1.5 | |
| 666 | 2007 | 2 | 15 | 15 | | | | 2.8 | <0.1 | <0.05 | 1.8 | 1.7 | sZ |
| 460 | 2007 | 2 | 30 | 15 | 1.5 | 3.5 | 23.6 | | | | 1.6 | 1.7 | |
| 654 | 2007 | 2 | 45 | 15 | 1.4 | 3.4 | 22.7 | | | | 1.6 | 1.7 | |
| 483 | 2007 | 2 | 60 | 15 | | | | 2.3 | <0.1 | <0.05 | 1.7 | 1.7 | sZ |
| 290 | 2007 | 2 | 70 | 10 | 1.2 | 1.9 | 19.1 | | | | 1.6 | 1.7 | |
| 480 | 2007 | 2 | 82 | 12 | 1.2 | 2.7 | 22.1 | | | | 1.8 | 1.7 | |





| Sample-ID | Date | Profile number | Depth below surface | Layer thickness | TOC | SOCC | $CD_b$ | C | N | S | Dry bulk density | Average dry bulk density | Grain size class |
|---|---|---|---|---|---|---|---|---|---|---|---|---|---|
| | [YYYY] | | [cm] | [cm] | [wt%] | [kg/m²] | [kg/m³] | [wt%] | [wt%] | [wt%] | [g/cm³] | [g/cm³] | |
| 532 | 2007 | 2 | 95 | 13 | | | | 2.9 | <0.1 | <0.05 | 1.7 | 1.7 | sZ |
| 298 | 2007 | 2 | 105 | 10 | 1.3 | 2.2 | 21.8 | | | | 1.7 | 1.7 | |
| 593 | 2007 | 2 | 110 | 5 | | | | 2.4 | <0.1 | <0.05 | 1.6 | 1.7 | sZ |
| 782 | 2007 | 3 | 15 | 15 | | | | 2.0 | 0.1 | <0.05 | 1.4 | 1.5 | sZ |
| 288 | 2007 | 3 | 30 | 15 | 1.1 | 2.2 | 15.0 | | | | 1.4 | 1.5 | |
| 513 | 2007 | 3 | 40 | 10 | 1.7 | 2.4 | 23.6 | | | | 1.4 | 1.5 | |
| 208 | 2007 | 3 | 50 | 10 | 1.0 | 1.5 | 15.3 | 2.0 | <0.1 | <0.05 | 1.5 | 1.5 | sZ |
| 478 | 2007 | 3 | 65 | 15 | | | | 1.5 | <0.1 | 0.1 | 1.4 | 1.5 | sZ |
| 469 | 2007 | 3 | 80 | 15 | 0.5 | 1.1 | 7.6 | | | | 1.5 | 1.5 | |
| 292 | 2007 | 3 | 100 | 20 | 0.9 | 2.8 | 14.0 | | | | 1.6 | 1.5 | |
| 270 | 2007 | 3 | 125 | 25 | | | | 0.9 | <0.1 | <0.05 | 1.8 | 1.5 | sZ |
| | 2009 | 1 | 5 | 5 | | | | | | | 2.1 | 2.2 | |
| | 2009 | 1 | 19 | 19 | 1.3 | 5.6 | 29.5 | 1.8 | <0.1 | 0.3 | 2.2 | 2.2 | sZ |
| | 2009 | 1 | 30 | 11 | 15.7 | 37.7 | 345.8 | 15.4 | 0.4 | 1.0 | | 2.2 | zS |
| | 2009 | 1 | 35 | 5 | 51.2 | 55.9 | 1126.7 | 45.7 | 1.1 | 1.9 | 2.2 | 2.2 | Z |
| | 2009 | 1 | 50 | 15 | 2.2 | 7.4 | 49.5 | 2.4 | 0.1 | 0.2 | | 2.2 | sZ |
| | 2009 | 1 | 60 | 10 | 1.5 | 3.3 | 33.3 | 1.8 | <0.1 | 0.2 | | 2.2 | sZ |
| | 2009 | 1 | 70 | 10 | 0.8 | 1.7 | 17.3 | 1.0 | <0.1 | 0.2 | | 2.2 | sZ |
| | 2009 | 1 | 80 | 10 | 1.5 | 3.3 | 33.5 | 1.7 | 0.1 | 0.2 | | 2.2 | sZ |
| | 2009 | 1 | 90 | 10 | 1.0 | 2.3 | 22.8 | 1.2 | <0.1 | 0.2 | | 2.2 | sZ |
| | 2009 | 1 | 100 | 10 | 0.9 | 1.9 | 19.4 | 1.0 | <0.1 | 0.2 | | 2.2 | sZ |
| | 2009 | 1 | 110 | 10 | 0.9 | 2.1 | 20.7 | 1.1 | <0.1 | 0.2 | | 2.2 | sZ |
| | 2009 | 1 | 120 | 10 | 1.1 | 2.3 | 23.5 | 1.2 | <0.1 | 0.2 | | 2.2 | sZ |

**Table F1.** Soil data from the soil pits in 1998, 2007, and 2009. The location of the soil profiles (including pictures and location) is given in Appendix C. Grain size class according to Folk (1954) with S: sand, s: sandy, Z: silt, z: silty, M: mud, m: muddy, C: clay, and c: clayey. In 2009 at depths of 30 and 35 cm high Total Organic Content (TOC)-, Soil Organic Carbon Content (SOCC)-, bulk Carbon Density (CD$_{bulk}$)- and Carbon (C)-values possible due to natural coal in the soil profile. The soil organic carbon content ($SOCC$) [kg m$^{-2}$] has been calculated using the following formula:

$$SOCC = TOC * \overline{\rho_{bulk}} * z = \frac{CD_{bulk}}{z} \qquad \text{(F1)}$$

The bulk carbon density ($CD_{bulk}$) [kg m$^{-3}$] has been calculated using the bulk soil density $\rho_{bulk}$ and the following formula:

$$CD_{bulk} = TOC * \overline{\rho_{bulk}} = SOCC * z \qquad \text{(F2)}$$





## Appendix G: Names of the variables and units for data files

| variable | columnname | unit |
|---|---|---|
| air/snow temperature | Tair_(height in cm) | °C |
| relative humidity | RH_(height in cm) | % |
| incoming shortwave radiation | SwIn | W m⁻² |
| outgoing shortwave radiation | SwOut | W m⁻² |
| incoming longwave radiation | LwIn | W m⁻² |
| outgoing longwave radiation | LwOut | W m⁻² |
| net radiation | RadNet | W m⁻² |
| wind speed | Vwind_(height in cm) | m s⁻¹ |
| wind direction | Dirwind_(height in cm) | ° |
| wind direction standard deviation | Dirwind_sd_(height in cm) | ° |
| soil/permafrost temperature | Ts_(depth in cm) | °C |
| soil bulk electrical conductivity | Cond_(depth in cm) | S m⁻¹ |
| dielectric number | E2_(depth in cm) | – |
| soil volumetric  liquid water content | Vwc_(depth in cm) | – |
| ground heat flux | G | W m⁻² |
| precipitation (liquid) | Prec | mm |
| snow depth | Dsn | m |

**Table G1.** Overview of all variables provided as time series. The variables electric conductivity, dielectric number, soil temperature, and volumetric soil liquid water content include a second number or letter in the column name. In these cases, the first number is the distance in the 2D profile and the second is the depth. The single letters [a, b, c, d] refer to the different 1D profiles. Vertically oriented probes are marked with letter [v]. Additional Level 2 data is provided for the variables snow depth, soil temperature, and volumetric soil liquid water content, which is indicated by "_lv2" in the column names. If an air temperature sensor is covered by snow and thus measures snow temperature, this is indicated by a flag in the data.





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
