# Peer review of "A 20-year record (1998-2017) of permafrost, active layer, and meteorological conditions at a High Arctic permafrost research site (Bayelva, Spitsbergen): an opportunity to validate remote sensing data and land surface, snow, and"

_Earth System Science Data, 2017_

## Referee Comment (RC1) · Anonymous Referee #1 · 28 Nov 2017

I have read this manuscript with great interest. Climate change is of critical societal concern and is currently a central theme for a number of scientific disciplines. Observations and models suggest most rapid warming at high latitudes and the presence of permafrost makes these amongst the most sensitive global environments. The 20-year record of permafrost, active layer, and meteorological conditions presented by Boike

et al. from Bayelva near Ny-Ålesund is a unique high-quality datset that will be very useful for many scientists and projects in the future. This is a clearly written paper and the overall structure of the article is well structured. The main sources of errors and uncertainty are given and discussed. The data set is made easy accessible. The manuscript should be accepted and needs only some minor revisions:

Title: Consider a shorter title; only use e.g the first part: A 20-year record (1998-2017) of permafrost, active layer, and meteorological conditions at a High Arctic permafrost research site (Bayelva, Spitsbergen)

In the abstract (L36) and other parts of the manuscript (e.g. L104, L520) the term "climate warming" and "warming of air temperatures" are used. Better use "global warming", "atmospheric temperature rise" or so instead of "climate warming"/"warming climate". The term "climate" is defined as a statistical average of meteorological conditions and as such cannot "warm" (the expression is popular but not really scientifically correct).

L63: After IPY new updates on changes in permafrost temperature have been published and presented in recent peer-reviewed assessments, like SWIPA2017. In addition to Romanovsky et al. (2010) I therefore suggest to add e.g. Romanovsky et al. (2017). Romanovsky V, Isaksen K, Drozdov D, Anisimov O, Instanes A, Leibman M, McGuire AD, Shiklomanov N, Smith S, Walker D, 2017. Changing permafrost and its impacts. In: Snow, Water, Ice and Permafrost in the Arctic (SWIPA) 2017. pp. 65-102. Arctic Monitoring and Assessment Programme (AMAP), Oslo, Norway.

L208: Replace "1989" with "1898". Consider also to include that a meteorological station was established in Ny-Ålesund already in 1969.

L470: Replace "thermometer chain" with "thermistor chain" or "thermistor string"

L 479-481: Please explain more in detail why you think there is an air exchange within the casing in the uppermost 1.5 m. There are several other boreholes with similar

setup and such information is important.

Figure 1: Please include exact coordinates of the Bayelva site in e.g. the figure caption.

[Figure]

---

## Referee Comment (RC2) · Anonymous Referee #2 · 29 Nov 2017

Good dataset.

Major questions:

I do not quite understand why you only list the usefulness of the dataset for RS validation and permafrost modelling?

[Figure]

Why not at least include information on the overall results that the dataset provides such as active layer thickness (ALT) and its variations and one the permafrost thermal state?

Why not provide information if there are some CALM measurements and what they show compared to your dataset on ALT?

Do you have any stratigraphical information from the drilling of the borehole (would strengthen the use of the dataset in many ways) ? and some info on the drilling method (hammer or coring) ?

Otherwise only smaller detailed questions:

Line 50: 'The data set also includes a high resolution digital elevation model that can be used together with the snow physical information for snow pack modeling'. This is the second mentioning of the high res. DEM in the abstract.

Line 60: (the "active layer"), I would use the active layer as this is standard scientific terminology within permafrost science.

Line 61-63: 'Thermal degradation of permafrost over the last few decades has been reported from many circum-Arctic boreholes (Romanovsky et al., 2010)'. Well not many. Most boreholes do not actually have very long records, so please include the number or say some.

Line 70: 'and temperature'. This is a listing of additional parameters to temperature, so this should not be here as well.

Line 71: 'can blanket the permafrost surface' normally snow is isolating the permafrost

Line 75: 'resulting datasets' are some dataset not resulting..I do not quite follow the use of resulting datasets..why not simply say datasets ?

Line 82: 'has been investigated by the AWI'. A site cannot be investigated, but used for research..

Line 101-102: 'The West Spitsbergen Ocean Current, a branch of the North Atlantic Current, warms this area to an average air temperature of about $-13°$C in January and $+5°$C in July'. Please add reference and time period.

Line 102-103: 'It also provides about 400 mm of precipitation annually, which falls mostly as snow between September and May. An ocean current cannot provide precipitation.

Line 104: 'Significant warming of air temperatures'. Air temperature either increases or decreases, but do not warm or cool.

Line 106-108: 'This warming is also reflected in the permafrost temperatures, as recorded from deep boreholes (up to 102 m deep) in the mountains at Janssonhaugen'. Do you mean that the permafrost has warmed down to 102 m due to which air temperature increase?

Figure 1: On the overview figure the arrow point in the sea? Could point at the site, or make a dot of the site, which then shows on the next top right figure. C and d show the area in summer and winter simply.

Figure 2 Suggest to change the figure text to: Topographical map (a) and overview of the Bayelva study site (b) in Ny-Ålesund, Svalbard. (a) Aerial orthophoto (20 cm/px) with topographic information (2.5 m contour lines) derived from a Digital Elevation Model (DEM) with a cell size of 0.5 m. The locations of the instruments and sampling sites are marked by coloured symbols. The site is located on a small hill; white areas are snow fields remaining late in the season. The orthophoto and DEM were obtained with an HRSC-AX camera in August 2008 (data and metadata for the high resolution digital elevation model covering the entire area shown in Figure 1b are provided in Appendix B).

Line 141-143: 'The hill consists mainly of rock but is partly covered by a mixture of sediments that consist of glacial till, together with fine grained glacio-fluvial sediments

and clays from the last glacial advance' could you make a reference to this important stratigraphical information please.

Line 146; 'below the maximum height for marine sedimentation' you must mean below the upper marine limit?

Line 155: 'These non-sorted circles formed under localised favourable conditions following the last glacial period'. Please provide reference, or explain how you know this timing.

Line 171: 'The depth of the snow cover', The snow depth..

Figure 3: Please make sure to print these figures horizontally then much easier to directly read and probably also to compare the 3 parts on one page. What is the snow temperature in the snow free period? probably best to remove..Precipitation is normally shown upwards not downwards. Soil temperature is actually the active layer temperatures, which would fit better with permafrost temperatures below.

Line 208-209: 'Climate records covering a longer period of time (since 1989) are available from the airport in Longyearbyen....' They go much further back more than 100 yrs.

Line 222: 'PT00 temperature sensors'..must be PT100 temperature sensors.

Line 258: argely – largely

Line 259-260: The Q7 sensor was destroyed by reindeer in September 2003 and was not replaced. When was the reindeer fence put up and why did it not work?

Line 265: 'ground Level' – ground level.

Line 280: (about 85 m; Figure 2), (about 85 m away ?...

Line 284-285:'To obtain the snow depth the distance of the sensor from the ground surface was recorded annually and subtracted from 285 the corrected distance

data'. . .What about frost lifting of the measuring pole, is this taken into consideration ?

Line 296-298: 'It should be noted that two of the sensors (an SR50 sensor and the laser sensor) were located within 5 m of each other, while the second SR50 (at the eddy covariance site) was located 85 m away, downhill from the Bayelva site (Figure 2)' This is repeating already provided information.

Line 303: to monitor the timing and pattern of snow melt. . .must be for all snow cover changes not only melt

Figure 4. 'Bayelva soil temperature trends for three depths from 1998- September 2017 using Level 2 soil temperatures'. So this is active layer and top permafrost temperatures, would be good to use this terminology. 'a) yearly trends at all three depths of 4, 58, 138 cms are 0.18°C/year (standard error of trends: ± 0.07, 0.06, 0.05 °C/570 year) b) winter trends (months December, January, February) are 0.25, 0.25, 0.23°C/winter (standard error of trends: ± 0.12, 0.11, 0.07 °C/winter) and c) summer trends (months June, July, August) are 0.08, 0.1, 0.12°C/summer (standard error of trend ± 0.05, 0.04, 0.03 °C/summer). Years in which data gaps of more than 48 hours exceeded 5% of data were included. Data gaps were interpolated linearly'. Suggestion for changes: a) the mean 20 year temperature increase in the active layer at 4 and 58 cm depths and in the top permafrost at 138 cm was 0.18°C/year, the mean 20 year winter temperature increases were. . ..., while the mean 20 year summer increases were. . ..

Line 341: 'stockpiled separate'- stockpiled separately

Line 378: 'After installation, the sensors cannot be re-calibrated' – why not? Could be dug up for recalibration..

Line 383-384: 'Subsequently, temperature readings from nine of the 32 sensors subsequently started to drift and' please not more than one subsequently..

Line 396-397: 'increased noise Levels' increased noise levels..

Line 407: 'the permanently frozen soil'- the permafrost
Figure 4 interpretation...you list means for the entire period, but very interesting there was a large increase before 2002 and then more stable conditions since then..would be good to comment on this

Line 424: were installed were installed..only once..

Line 451: The lack of calibration

Line 467-468: Ten thermometers were installed, one above the ground surface and nine from the surface down to 9 m depth..you mean a thermistor string !

Line 488: data Levels – data levels

Line 502: soil temperature – sediment temperature..no information given on soil development, so sediment would be the most natural to use in this connection.

Line 508-509: The data from this Bayelva site have been widely used for the development and evaluation of land surface models. Include references.

Line 519: Permafrost around the Arctic is thawing and warming – permafrost around the Arctic is warning and thawing.

Line 521: soil temperature – you need to specify is this is sediment or mixtures on rock and sediment?

Line 523: (Figure 4 h). Such a figure does not exist (Fig. 4 a, b and c only)! Line 523-524: Interannual to sub-decadal variability is evident in the data and results mostly from differences during the winter months. But you do not at all explain these here, which is a pity, so why include this sentence?

Table 2. Permafrost temperature is Active layer and permafrost temperature.

---

## Author Comment (AC1) · 23 Dec 2017

Dear reviewer,

thank you very much for your detailed comments which helped to clarify the manuscript. We have been through them in detail and made amends as requested. We provide

a point to point response below to your comments (AC), as well as changes in the manuscript (CM). We also provide a revised manuscript in "tracked changes mode".

On behalf of the authors,

Julia Boike

Please also note the supplement to this comment:
https://www.earth-syst-sci-data-discuss.net/essd-2017-100/essd-2017-100-AC1-supplement.pdf

**Supplement:**

Author comments on https://doi.org/10.5194/essd-2017-100-RC1, 2017

**A 20-year record (1998–2017) of permafrost, active layer, and meteorological conditions at a High Arctic permafrost research site (Bayelva, Spitsbergen): an opportunity to validate remote sensing data and land surface, snow, and permafrost models**
by Julia Boike et al.

**Anonymous Referee #1**

**RC**: Referee comment | **AC**: Author comment | **CM**: Change in the manuscript

**RC1.01:** Title: Consider a shorter title; only use e.g the first part: A 20-year record (1998-2017) of permafrost, active layer, and meteorological conditions at a High Arctic permafrost research site (Bayelva, Spitsbergen)
**AC1.01:** We adopted the amendment proposed by the referee.
**CM1.01:** We changed the title as follows: A 20-year record (1998-2017) of permafrost, active layer, and meteorological conditions at a High Arctic permafrost research site (Bayelva, Spitsbergen): an opportunity to validate remote sensing data and land surface, snow, and permafrost models

**RC1.02:** In the abstract (L36) and other parts of the manuscript (e.g. L104, L520) the term "climate warming" and "warming of air temperatures" are used. Better use "global warming", "atmospheric temperature rise" or so instead of "climate warming"/"warming climate". The term "climate" is defined as a statistical average of meteorological conditions and as such cannot "warm" (the expression is popular but not really scientifically correct)
**AC1.02:** We adopted the amendments proposed by the referee.
**CM1.02:** We changed the named passages as follows: L36: global warming; L104: of air temperatures rise; L106: This rise; L521: soil temperature data at all depths have been rising

**RC1.03:** L63: After IPY new updates on changes in permafrost temperature have been published and presented in recent peer-reviewed assessments, like SWIPA2017. In addition to Romanovsky et al. (2010) I therefore suggest to add e.g. Romanovsky et al. (2017). Romanovsky V, Isaksen K, Drozdov D, Anisimov O, Instanes A, Leibman M, McGuire AD, Shiklomanov N, Smith S, Walker D, 2017. Changing permafrost and its impacts. In: Snow, Water, Ice and Permafrost in the Arctic (SWIPA) 2017. pp. 65-102. Arctic Monitoring and Assessment Programme (AMAP), Oslo, Norway.
**AC1.03:** We included the reference proposed by the referee.
**CM1.03:** We changed the named sentence as follows: Thermal degradation of permafrost over the last few decades are reported from ten available circum-Arctic boreholes (Romanovsky et al., 2010) with a recent update by Romanovsky et al. (2017).

**RC1.04:** L208: Replace "1989" with "1898". Consider also to include that a meteorological station was established in Ny-Ålesund already in 1969.

**AC1.04:** We adopted the replacement proposed by the referee.

**CM1.04:** We changed the named passage as follows: Climate records covering a longer period of time  are available from  Svalbard since 1898, with the establishment of a permanent weather station in 1911 (Førland et al. 2011).

**RC1.05:** L470: Replace "thermometer chain" with "thermistor chain" or "thermistor string"

**AC1.05:** We adopted the replacement proposed by the referee.

**CM1.05:** We changed the named passage as follows: the thermistor chain

**RC1.06:** L 479-481: Please explain more in detail why you think there is an air exchange within the casing in the uppermost 1.5 m. There are several other boreholes with similar setup and such information is important.

**AC1.06**: We provide further information about the differences in temperature in the two figures below. The temperature data show that the differences between borehole and active layer profile can be up to several degrees during winter (Figures R1.01. & R1.02. below). The temperatures in the borehole (50 cm) during winter are colder in comparison to the soil profile (55 cm) and follow closely the air temperature signal. This effect is still clearly observed at about 150 cm depth in winter. We assume that differences in snow cover are in part responsible for the changes between borehole and profile temperatures: the snow cover is thinner at the borehole location compared to the active layer profile observed from aerial photographs. A recently submitted study to *The Cryosphere* (Gouttevin et al., Observation and modelling of snow at a polygonal tundra permafrost site: spatial variability and thermal implications) showed that spatial variability in snow (incl. both depth and structure) could lead to soil temperature differences up to 6°C at 50 cm depth. Furthermore, differences in ground properties (texture, soil volumetric water content) between the borehole and active layer profile could also be responsible for the differences in temperatures.

Since borehole drilling, pipe fitting, refilling of the pipe (no filling or addition of liquid) is different between borehole sites, we cannot give a universal statement, but rather suggest using caution when interpreting the upper 2 m of temperature.

[Figure]

[Figure]

**Figure R1.01. Upper:** Daily mean temperature in the borehole and active layer profile at depths of 50 and 55 cm, and air temperature. **Lower:** Daily mean temperature in the borehole and active layer profile at depths of 150 and 141 cm, and air temperature.

[Figure]

**Figure R1.02.** Comparison between mean borehole temperature and active layer profile on February 6, 2013.

**CM1.06:** We added the following information to the named passage as follows:  Caution is warranted when using the borehole temperatures in the uppermost 1–2 m, since they may be affected by air movement in the borehole.

**RC1.07:** Figure 1: Please include exact coordinates of the Bayelva site in e.g. the figure caption.

**AC1.07:** We adopted the amendment proposed by the referee.

[revised manuscript text omitted]
 between –-17.0 °C to -3.8 °C in January and from 4.6 to 6.9 °C in July (18-year period: 01 August 1993 to 31 July 2011; -(Maturilli et al., 2013)). It also provides about 400 mm of precipitation annually, which falls mostly as snow between September and May. Significant  air temperatures rise has been detected since 1960, which has generally been attributed to changes in the radiation budget and atmospheric circulation patterns (Førland et al., 2012; Hanssen-Bauer and Førland, 1998). This rise is also reflected in the permafrost temperatures, as recorded from deep boreholes  in the mountains at Janssonhaugen (Isaksen et al., 2007a; Isaksen et al., 2001; Isaksen et al., 2007b). For the 6-year period from 1999 to 2005 the borehole temperatures have increased: at 25 m by 0.26 °C, at 30 m by 0.19 °C and at 40 m by between 0.07 and 0.09 °C –(Romanovsky et al., 2017). Continuous permafrost underlies the un-glaciated coastal areas to a depth of about 100 m and the active layer thickness (ALT) at the end of summer ranges between 1 and 2 m (Humlum, 2005). ALT data are available from the Global Terrestrial Network for Permafrost (GTNP) database (http://gtnpdatabase.org/) for several locations on Svalbard, but for different time periods and distances between 25 and 115 km away from Bayelva. The site with the longest record, Janssonhaugen, is located 115 km to the southeast and had an average thaw depth of 175 cm for 1998–2013. The region has experienced increases in cloudiness, precipitation, and the number and intensity of cyclones in recent years, especially during the winter months (Hanssen-Bauer and Førland, 1998; Sepp and Jaagus, 2011). The increase in cloudiness (Maturilli and Kayser, 2016) has led to an increase in incoming long-wavelength radiation,

resulting in a major change to the winter radiation budget for this region, as measured at the German-French (AWIPEV) research station, which is located within the village of Ny-Ålesund (Maturilli et al., 2014). This research station carries out long term monitoring of radiation (Baseline Surface Radiation Network: http://bsrn.awi.de/) and meteorological data 130 (http://www.awipev.eu/).

The data presented herein were collected from the High Arctic Bayelva River catchment area (Figure 1), away from Ny-Ålesund. A legacy from past mining activities is the physical disturbance of the ground in and around Ny-Ålesund (for example, compaction and reworking of the soil). Traffic in the village (people, cars, snow mobiles) also affects the surface 135 conditions, especially in winter. The Bayelva catchment area lies between two mountains (Zeppelinfjellet and Scheteligfjellet), with the glacial Bayelva River originating from the two branches of the Brøggerbreen glacier. The terrain flattens out to the north of the Bayelva site and the Bayelva River flows into the Kongsfjorden fjord and the Arctic Ocean about 1 km from the site (Figure 1). Over the past three decades the Bayelva catchment area has been the 140 focus of intensive investigations into fluvial hydrology (sediment transfer and geochemistry; (Hodson et al., 2002)), soil and permafrost conditions (Boike et al., 2008b; Roth and Boike, 2001; Westermann, 2010; Westermann et al., 2011b), the surface energy balance (Boike et al., 2003b; Westermann et al., 2009), and the micrometeorological processes controlling the surface gas and energy exchanges (Lloyd, 2001b; Lloyd et al., 2001; Lüers et al., 2014). A 145 permafrost maximum, minimum and average temperature with depth (trumpet curve) was shown for data for one year (August 2009–August 2010) by Boike et al., (2012).The data from this Bayelva site have also been used in earth system modeling (Ekici et al., 2014; Ekici et al., 2015). Nearby investigations by Japanese and Italian researchers include vegetation analysis with respect to periglacial and glacial landforms and topography (Cannone et al.,

150 2004; Ohtsuka et al., 2006), and investigations into the plant and plot scale dependence of $CO_2$ emissions on biotic and abiotic factors at the start and end of the growing season (Cannone et al., 2016; Uchida et al., 2006).

The Bayelva site is located on top of the Leirhaugen hill (25 m a.s.l.), on permafrost patterned ground (Figure 2). The hill consists mainly of rock but is partly covered by a mixture of

155 sediments that consist of glacial till, together with fine grained glacio-fluvial sediments and clays from the last glacial advance, likely deposited by the Kongsfjorden glacier (J. Tolgensbakk, personal information). According to the geological map, the site is part of the Brøggerbreen Formation (lithography sand-stone, shale, conglomerate), but the gray color of the sediments suggests that the material was deposited by the Kongsfjorden glacier and not

160 the adjacent Brøggerbreen glacier, which deposits redder material. Thus, the site is part of the Kongsfjorden Formation, with the main lithography sandstone, shale, coal (Norwegian Polar Institute online geological maps, accessed December 2017 ).  
[revised manuscript text omitted]

A 9.3 m borehole was drilled using a rotary drill rig on March 30, 2009 and cased with PVC. No stratigraphic information or drill core was recovered.  A thermistor string was installed, with one sensor above the ground surface and nine from the surface down to 9 m depth (at 0, 0.5, 1.0, 1.5, 2.5, 3.5, 5.5, 7.5, 9 m below ground surface, Appendix C, Figure C10). The casing was left open (not refilled) so that the thermistor chain could be retrieved or replaced. Because of instrument failure, thermometers were retrieved and replaced several times. From August 2009, a Geoprecision-M datalogger and sensor chain were used. Temperatures were recorded at hourly intervals, with no averaging; no data was recorded between 30 January 2011 and 29 March 2011, due to a low battery voltage. Geoprecision claims an accuracy of +/-0.05 C (at 0°C) and a resolution of 0.01°C, suitable for measurements in the range from -50 to +120°C. However, comparison measurements using an PT100 thermometer at the same depths in the borehole showed a deviation of up to 0.2°C. In 2014, a wooden shield was installed over the casing to prevent warming due to radiation and to facilitate natural ventilation (Appendix C, Figure C10).  Caution is warranted when using

505 the borehole temperatures in the uppermost 1–2 m, since they may be affected by air movement in the borehole. These data are flagged in the data series.

**4    Data quality control and availability**

[revised manuscript text omitted]

565  acknowledge the financial support provided through the European Union's FP7-ENV PAGE21 project under contract number GA282700. We thank the two anonymous reviewers whose comments helped to improve this manuscript.

[Figure]

Figure 1. (a) Location of the Bayelva study site on Spitsbergen, western Svalbard at 78°55'15.6"N 11°49'58.5"E. (b) The site is located about 3 km from Ny-Ålesund in the Bayelva River catchment area, between two mountains (Zeppelinfjellet and Scheteligfjellet), and in front of the Brøggerbreen glacier. Aerial image captured in August 2008 using a high resolution HRSC-AX camera (Hauber et al., 2011a; Hauber et al., 2011b): data and metadata for the high resolution the HRSC-AX image covering the entire area shown in Figure 1b are provided in Appendix B (c) & (d). The area of the site under summer conditions (August 2008) and spring conditions (April 2016).

[Figure]

[Figure]

**Figure 2.** Topographical map (a) and overview of the Bayelva study site (b) in Ny-Ålesund, Svalbard. **(a)** Aerial orthoimage (20 cm/px) with topographic information (2.5 m contour lines) derived from a  Digital Elevation Model (DEM) with a cell size of 0.5 m. The locations of the instrumental and sampling sites are marked by coloured symbols. The site is located on a small hill; white areas are snow fields remaining late in the season. The orthophoto and DEM were obtained with an HRSC-AX camera in August 2008 (data and metadata for the

high resolution DEM covering the entire area shown in Figure 1b are provided in Appendix B). **(b)** Most of the instruments are within a fenced-off area to protect them from reindeer damage. Note that the eddy covariance station and permafrost borehole are located outside the fenced area.

[Figure]

590

**Figure 3.** Time series of Bayelva data provided in this paper . a-f: meteorological data, g-l: soil/subsurface data, m-p: snow data.  The data are organized following the structure of Appendix G. Further details on the sensors and periods of operation are given in Table 2. This plot (divided in three parts for better visibility) is also provided in Appendix H.

[Figure]

**Figure 4.** Bayelva mean 20-year  temperature  increases for three depths from  –September 1998 – September 2017 using Level 2 soil temperatures. a)  in the active layer  (4 and 58 cm) and in the top of permafrost (138 cm) :  0.18 °C/year (standard error of trends: ±–0.07 °C/year) b) in winter  (December, January, February):  0.25 ±0.12, 0.25 ±0.11, and 0.23 ±0.07°C/year, respectively, and c) in summer  (months June, July, August):  0.08 ±0.05, 0.10 ±0.04, and 0.12 ±0.03°C/ year, respectively.  Years in which data gaps of more than 48 hours exceeded 5% of data were included; data gaps were interpolated linearly.

[revised manuscript text omitted]

Norwegian Polar Institute: Svalbardkartet Geology Map 207, http://svalbardkartet.npolar.no/html5/index.html?viewer=svalbardkartet.html5, last access: 22 December 2017, 2016.

Ohtsuka, T., Adachi, M., Uchida, M., and Nakatsubo, T.: Relationships between vegetation types and soil properties along a topographical gradient on the northern coast of the Brøgger Peninsula, Svalbard, Polar Biosciene, 19, 63-72, 2006.

Overduin, P. P. and Kane, D. L.: Frost Boils and Soil Ice Content: Field Observations, Permafr. Periglac. Proc., 17, 291–307, 2006.

Romanovsky, V., Isaksen, K., Drozdov, D., Anisimov, O., Instanes, A., Leibman, M., McGuire, A. D., Shiklomanov, N., Smith, S., and Walker, D.: Changing permafrost and its impacts. In: Snow, Water, Ice and Permafrost in the Arctic (SWIPA) 2017, Arctic Monitoring and Assessment Programme (AMAP), Oslo, Norway, 2017.

Romanovsky, V. E., Smith, S. L., and Christiansen, H. H.: Permafrost thermal state in the polar Northern Hemisphere during the international polar year 2007–2009: a synthesis, Permafr. Periglac. Proc., 21, 106-116, 2010.

Roth, K. and Boike, J.: Quantifying the Thermal Dynamics of a Permafrost Site Near Ny-Ålesund, Svalbard, Water Resour. Res., 37, 2901–2914, 2001.

Roth, K., Schulin, R., Flühler, H., and Attinger, W.: Calibration of Time Domain Reflectometry for Water Content Measurement Using a Composite Dielectric Approach, Water Resour. Res., 26, 2267-2273, 1990.

Schneebeli, M., Coleou, C., Touvier, F., and Lesaffre, B.: Measurement of density and wetness in snow using time-domain reflectometry, Ann. Glaciol., 26, 69–72, 1998.

Scholten, F. and Gwinner, G.: Operational Parallel Processing in Digital Photogrammetry - Strategy and Results using Different Multi-Line Cameras, 20th International Congress for Photogrammetry and Remote Sensing, Istanbul (Turkey), 408–413, 2004.

Scholten, F., Gwinner, K., Roatsch, T., Matz, K.-D., Wählisch, M., Giese, B., Oberst, J., Jaumann, R., Neukum, G., and HRSC Co-I-Team: Mars Express HRSC Data Processing - Methods and Operational Aspects, Photogramm. Eng. Remote Sensing, 71, 1143–1152, 2005.

Sepp, M. and Jaagus, J.: Changes in the activity and tracks of Arctic cyclones, Climatic Change, 105, 577-595, 2011.

Steinhart, J. S. and Hart, S. R.: Calibration curves for thermistors, Deep Sea Research and Oceanographic Abstracts, 15, 497-503, 1968.

Stern, L.: Hydraulic and Thermal properties of Permafrost in Svalbard, Analysis and Modelling using GEOtop, Master of Science, Department of Physics and Astronomy, University of Heidelberg, 73 pp., 2017.

Uchida, M., Kishimoto, A., Muraoka, H., Nakatsubo, T., Kanda, H., and Koizumi, H.: Seasonal shift in factors controlling net ecosystem production in a high Arctic terrestrial ecosystem, J. Plant Res., 123, 1-7, 2009.

Uchida, M., Nakatsubo, T., Kanda, H., and Koizumi, H.: Estimations of the annual primary production of the lichen Cetrariella delisei in a glacier foreland in the High Arctic, Ny-Ålesund, Svalbard, Polar Research, 25, 39-49, 2006.

Weismüller, J., Wollschläger, U., Boike, J., Pan, X., Yu, Q., and Roth, K.: Modeling the thermal dynamics of the active layer at two contrasting permafrost sites on Svalbard and on the Tibetan Plateau, The Cryosphere, 5, 741-757, 2011.

Westermann, S.: Sensitivity of Permafrost, Doctor of Natural Sciences PhD thesis, Combined Faculties for the Natural Sciences and for Mathematics, Ruperto-Carola University, Heidelberg, 174 pp., 2010.

Westermann, S., Boike, J., Guglielmin, M., Gisnås, K., and Etzelmüller, B.: Snow melt monitoring near Ny-Ålesund, Svalbard, using Automatic Camera Systems. PANGAEA, 2015.

Westermann, S., Boike, J., Langer, M., Schuler, T. V., and Etzelmüller, B.: Modeling the impact of wintertime rain events on the thermal regime of permafrost, The Cryosphere, 5, 945-959, 2011a.

Westermann, S., Langer, M., and Boike, J.: Spatial and temporal variations of summer surface temperatures of high-arctic tundra on Svalbard -- Implications for MODIS LST based permafrost monitoring, Remote Sensing of Environment, 115, 908–922, 2011b.

Westermann, S., Lüers, J., Langer, M., Piel, K., and Boike, J.: The annual surface energy budget of a high-arctic permafrost site on Svalbard, Norway, The Cryosphere, 3, 245-263, 2009.

1095    Wewel, F., Scholten, F., and Gwinner, K.: High Resolution Stereo Camera (HRSC) – Multispectral 3D-Data Acquisition and Photogrammetric Data Processing, Canadian Journal of Remote Sensing, 26, 466–474, 2000.

Yen, Y.-C.: Review of thermal properties of snow, ice and sea ice, CRREL, Hanover,  NH, USA, 34 pp., 1981.

1100    Yoshitake, S., Uchida, M., Ohtsuka, T., Kanda, H., Koizumi, H., and Nakatsubo, T.: Vegetation development and carbon storage on a glacier foreland in the High Arctic, Ny-Ålesund, Svalbard, Polar Science, 5, 391–397, 2011.

---

## Author Comment (AC2) · 23 Dec 2017

Dear reviewer,

thank you very much for your detailed comments which helped to clarify the manuscript. We have been through them in detail and made amends as requested. We provide

a point to point response below to your comments (AC), as well as changes in the manuscript (CM). We also provide a revised manuscript in "tracked changes mode".

On behalf of the authors,

Julia Boike

Please also note the supplement to this comment:
https://www.earth-syst-sci-data-discuss.net/essd-2017-100/essd-2017-100-AC2-supplement.pdf

[Figure]

**Supplement:**

Author comments on https://doi.org/10.5194/essd-2017-100-RC2, 2017

**A 20-year record (1998–2017) of permafrost, active layer, and meteorological conditions at a High Arctic permafrost research site (Bayelva, Spitsbergen): an opportunity to validate remote sensing data and land surface, snow, and permafrost models**
by Julia Boike et al.

**Anonymous Referee #2**

**RC**: Referee comment | **AC**: Author comment | **CM**: Change in the manuscript

**RC2.01:** I do not quite understand why you only list the usefulness of the dataset for RS validation and permafrost modelling?
**AC2.01:** We changed the title by using only the first part of it.
**CM2.01:** We changed the title as follows: A 20-year record (1998-2017) of permafrost, active layer, and meteorological conditions at a High Arctic permafrost research site (Bayelva, Spitsbergen): an opportunity to validate remote sensing data and land surface, snow, and permafrost models

**RC2.02:** Why not at least include information on the overall results that the dataset provides such as active layer thickness (ALT) and its variations and one the permafrost thermal state?
**AC2.02:** Our detailed temperature profile does not extend to the maximum active layer thaw depth (deepest sensor at 1.41 m), and the permafrost borehole location is not resolved enough in terms of sensor resolution (the active layer thaw depth falls between two borehole temperature sensors). We cannot measure ALT directly using a thaw probe since it is physically not possible to penetrate the soil with a probe at this site.
Thus, we have modelled the active layer thaw depth using the Stefan model and supply this figure below (Figure R2.01). Since these data are modelled though and require further explanation, we show this information only in the reply.

[Figure]

**Figure R2.01.** ALT at the Bayelva site over 17 years of data using Stefan model (in blue). A linear fit (in red) is added to show the increase in the active layer thickness of 4±0.5 cm per year. The ALT exceeded 2 m depth for the first time in 2016.

**CM2.02:** We added following information about the permafrost thermal state in the paper: **2. Site description**: A permafrost maximum, minimum and average temperature with depth (trumpet curve) was shown for data for one year (August 2009–August 2010) by Boike et al. (2012). **5. Outlook:** The record from 2009–2017 of permafrost temperature measurements shows a zero-amplitude depth of 5.5 m, at a mean temperature of -2.8°C. **Abstract:** The mean permafrost temperature is -2.8°C, with a zero-amplitude depth at 5.5 m (2009–2017).

**RC2.03:** Why not provide information if there are some CALM measurements and what they show compared to your dataset on ALT?
**AC2.03:** We do not measure ALT at this site using the CALM method, since the soil cannot be easily penetrated with a thaw depth probe and ALT is therefore prone to a systematic underestimation. CALM data are available from locations too far away to be relevant (>25 km away in different settings).

CALM data are available for the following locations:
1. Kafføyra: 25 km directly south of Bayelva, at 78°41'N 11°50'E, measurements available for 1999–2005, average ALT 110 cm.
2. Kapp Linne: 94 km south-east of Bayelva, at 78°03'N 13°37'E, measurements available for 1990–2011, ALT of 95 cm.
3. Janssonhaugen: 115 km south-east of Bayelva, at 78°10'N 16°28'E, measurements available for 1998–2013, ALT of 175 cm.
4. UNISCLAM: 115 km south-east of Bayelva, at 78°1210'N 16°28'E, measurements available for 2000–2015, ALT of 100 cm.
5. Calypsostranda: 162 km south-east of Bayelva, 77°34'N 14°30'E, measurements available for 1990–2009, ALT of 155 cm.

As it can be seen, i) there is no CALM grid established close to our site, ii) ALT is a local characteristic of the permafrost and differ from each location on Svalbard.

**CM2.03:** We added in **2. Site description:** ALT data are available from the Global Terrestrial Network for Permafrost database (http://gtnpdatabase.org/) for several locations on Svalbard, but for different time periods and distances between 25 and 115 km away from Bayelva. The site with the longest record, Janssonhaugen, is located 115 km to the southeast and had an average thaw depth of 175 cm for 1998–2013.

**RC2.04:** Do you have any stratigraphical information from the drilling of the borehole (would strengthen the use of the dataset in many ways) ? and some info on the drilling method (hammer or coring) ?

**AC2.04:** We do not have any stratigraphic information from the drilling. Drilling was done using a rotary drill rig (no core was obtained) and the material was removed from the borehole with compressed air. Visual inspection showed a different material with greater depth potentially indicating bedrock, but we have no confirmation.

**CM2.04:** We added the following information: A 9.3 m borehole was drilled using a rotary drill rig on 30 March 2009 and cased with PVC. No stratigraphic information or drill core was recovered.

**RC2.05:** Line 50: 'The data set also includes a high resolution digital elevation model that can be used together with the snow physical information for snow pack modeling'. This is the second mentioning of the high res. DEM in the abstract.

**AC2.05:** We adopted the amendment proposed by the referee.

**CM2.05:** We changed the named passages as follows: Additional data include a high resolution digital elevation model that can be used together with the snow physical information for snow pack modeling and a panchromatic image.;

**RC2.06:** Line 60: (the "active layer"), I would use the active layer as this is standard scientific terminology within permafrost science.

**AC2.06:** We adopted the amendment proposed by the referee.

**CM2.06:** We changed the named passage as follows: the active layer

**RC2.07:** Line 61-63: 'Thermal degradation of permafrost over the last few decades has been reported from many circum-Arctic boreholes (Romanovsky et al., 2010)'. Well not many. Most boreholes do not actually have very long records, so please include the number or say some.

**AC2.07:** We agree with the comment by the reviewer. According to the GTN-P network, there are only a few sites with long term borehole data available in Arctic permafrost areas, five of them located in Alaska: WestDock, Deadhorse, FranklBluffs, Gulakana, Chandalar Shelf, Alert, Table Mt., Norman Wells, Dionisiy, Umaybyt, Bolvansky, Tiksi_stone_ridge, Juvvasshoe, Janssonhaugen, Urengoy, Tarfalaryggen. An update of the temperature inventory reporting continuous increase of borehole temperatures has been recently compiled in the SWIPA report (Romanovsky V, Isaksen K, Drozdov D, Anisimov O, Instanes A, Leibman M, McGuire AD, Shiklomanov N, Smith S, Walker D, 2017. Changing permafrost and its impacts. In: Snow, Water, Ice and Permafrost in the Arctic (SWIPA) 2017. pp. 65-102. Arctic Monitoring and Assessment Programme (AMAP), Oslo, Norway.)

**CM2.07:** We changed the named sentence as follows: Thermal degradation of permafrost over the last few decades has been reported from ten available circum-Arctic boreholes (Romanovsky et al., 2010)' and recently been updated by Romanovsky et al. (2017).
Addition to the references:
Romanovsky V, Isaksen K, Drozdov D, Anisimov O, Instanes A, Leibman M, McGuire AD, Shiklomanov N, Smith S, Walker D, 2017. Changing permafrost and its impacts. In: Snow, Water, Ice and Permafrost in the Arctic (SWIPA) 2017. pp. 65-102. Arctic Monitoring and Assessment Programme (AMAP), Oslo, Norway

**RC2.08:** Line 70: 'and temperature'. This is a listing of additional parameters to temperature, so this should not be here as well.
**AC2.08:** We adopted the amendment proposed by the referee.
**CM2.08:** We changed the named passage as follows: various subsurface state variables (for example, volumetric liquid water content )

**RC2.09:** Line 71: 'can blanket the permafrost surface' normally snow is isolating the permafrost
**AC2.09:** We exchanged the word "blanket" with "insulate" since the word "blanket" seemed to cause misunderstanding.
**CM2.09:** We changed the named passage as follows: The seasonal snow cover in Arctic permafrost regions insulates the permafrost surface for many months of the year.

**RC2.10:** Line 75: 'resulting datasets' are some dataset not resulting..I do not quite follow the use of resulting datasets..why not simply say datasets ?
**AC2.10:** We adopted the amendment proposed by the referee.
**CM2.10:** We changed the named passage as follows: the  datasets

**RC2.11:** Line 82: 'has been investigated by the AWI'. A site cannot be investigated, but used for research.
**AC2.11:** We adopted the correction proposed by the referee.
**CM2.11:** We changed the named passage as follows: has been used for research by AWI

**RC2.12:** Line 101-102: 'The West Spitsbergen Ocean Current, a branch of the North Atlantic Current, warms this area to an average air temperature of about −13_C in January and +5_C in July'. Please add reference and time period.
**AC2.12:** We added the information provided in the publication by Maturilli et al. (2013).
**CM2.12:** We changed the named passage as follows: The West Spitsbergen Ocean , a branch of the North Atlantic , warms this area to an average air temperature of between -17.0 °C to -3.8 °C in January and from 4.6 to 6.9 °C in July (18-year period: 01 August 1993 to 31 July 2011; Maturilli et al. 2013).

**RC2.13:** Line 102-103: 'It also provides about 400 mm of precipitation annually, which falls mostly as snow between September and May. An ocean current cannot provide precipitation.
**AC2.13**: We adopted the suggestion proposed by the referee and deleted the words "current".
**CM2.13:** See comment CM2.12.

**RC2.14:** Line 104: 'Significant warming of air temperatures'. Air temperature either increases or decreases, but do not warm or cool.

**AC2.14:** We adopted the correction proposed by the referee.

**CM2.14:** We changed the named passage as follows: air temperature rise

**RC2.15:** Line 106-108: 'This warming is also reflected in the permafrost temperatures, as recorded from deep boreholes (up to 102 m deep) in the mountains at Janssonhaugen'. Do you mean that the permafrost has warmed down to 102 m due to which air temperature increase?

**AC2.15:** The warming has taken place down to 60 m depth, with statistically significant trends. For the 6 year period from 1999-2005 the borehole temperatures have been reported to increase: 25m (0.26°C), 30m (0.19°C) and 40 m (0.07–0.09°C).

**CM2.15:** We changed the sentence as follows: This warming is also reflected in permafrost temperatures, as in the deep borehole (up to 102 m deep) in the mountains at Janssonhaugen. For the 6-year period from 1999 to 2005 the borehole temperatures have increased: at 25m by 0.26°C, at 30m by 0.19°C and at 40 m by between 0.07 and 0.09°C (Romanovsky et al. (2017).

**RC2.16:** Figure 1: On the overview figure the arrow point in the sea? Could point at the site, or make a dot of the site, which then shows on the next top right figure. C and d show the area in summer and winter simply.

**AC2.16:** We changed the figure as suggested by the referee.

**CM2.16:** Figure 1c&d now show red dots at the sites.

**RC2.17:** Figure 2 Suggest to change the figure text to: Topographical map (a) and overview of the Bayelva study site (b) in Ny-Ålesund, Svalbard. (a) Aerial orthophoto (20 cm/px) with topographic information (2.5 m contour lines) derived from a Digital Elevation Model (DEM) with a cell size of 0.5 m. The locations of the instruments and sampling sites are marked by coloured symbols. The site is located on a small hill; white areas are snow fields remaining late in the season. The orthophoto and DEM were obtained with an HRSC-AX camera in August 2008 (data and metadata for the high resolution digital elevation model covering the entire area shown in Figure 1b are provided in Appendix B).

**AC2.17:** We adopted the amendments proposed by the referee.

**CM2.17:** We changed the text of Figure 2 as follows: Detailed images of the Bayelva study site on Spitsbergen. Topographic map (a) and overview of the Bayelva study site (b) in Ny-Ålesund, Svalbard. (a) Aerial orthoimage (20 cm/px) with topographic information (2.5 m contour lines) derived from a  Digital Elevation Model (DEM) with a cell size of 0.5 m. The locations of the instrumental and sampling sites are marked by coloured symbols. The site is located on a small hill; white areas are snow fields remaining late in the season. The orthophoto and DEM were obtained with an HRSC-AX camera in August 2008 (data and metadata for the high resolution DEM covering the entire area shown in Figure 1b are provided in Appendix B). (b) Most of the instruments are within a fenced-off area to protect them from reindeer damage. Note that the eddy covariance station and permafrost borehole are located outside the fenced area.

**RC2.18:** Line 141-143: 'The hill consists mainly of rock but is partly covered by a mixture of sediments that consist of glacial till, together with fine grained glacio-fluvial sediments and clays from the last glacial advance' could you make a reference to this important stratigraphical information please.

**AC2.18:** We received this information from J. Tolgensbakk and included the following sentences in the paper by Boike et al. (2008):"Leirhaugen hill is mainly composed of rock, but partly covered by a mixture of sediments: glacial till, finer glacio-fluvial sediments and clay formed by the last glacial advance (Tolgensbakk, personal communication). The gray color of the sediments suggests that the material was deposited by the Kongsfjorden glacier and not the adjacent Brøggerbreen glacier, which deposits redder material. Marine sedimentation could also have contributed since the hill is located below the marine limit (about 38 m).

**CM2.18:** We changed the sentences as follows: The hill consists mainly of rock but is partly covered by a mixture of sediments that consist of glacial till, together with fine grained glacio-fluvial sediments and clays from the last glacial advance, likely deposited by the Kongsfjorden glacier (J. Tolgensbakk, personal information). According to the geological map, the site is part of the Brøggerbreen Formation (lithography sandstone, shale, conglomerate), but the gray color of the sediments suggests that the material was deposited by the Kongsfjorden glacier and not the adjacent Brøggerbreen glacier, which deposits redder material. Thus, the site is part of the Kongsfjorden Formation, with the main lithography sandstone, shale, coal (Norwegian Polar Institute online geological maps, accessed December 2017).

The following reference has been added in the reference list:

Norwegian Polar Institute: Svalbardkartet Geology Map 207, http://svalbardkartet.npolar.no/html5/index.html?viewer=svalbardkartet.html5, last access: 22 December 2017, 2016.

**RC2.19:** Line 146; 'below the maximum height for marine sedimentation' you must mean below the upper marine limit?

**AC2.19:** We adopted the amendment proposed by the referee.

**CM2.19:** We changed the named passage as follows: which is below the upper marine limit

**RC2.20:** Line 155: 'These non-sorted circles formed under localised favourable conditions following the last glacial period'. Please provide reference, or explain how you know this timing.

**AC2.20:** These features are part of the postglacial landcover. We have no more precise ages available for these features, since this would be difficult in these cryoturbated features. Thus, we assume that these features have to be postglacial.

**CM2.20:** No changes in the paper.

**RC2.21:** Line 171: 'The depth of the snow cover', The snow depth..

**AC2.21:** We adopted the amendment proposed by the referee.

**CM2.21:** We changed the named passage as follows: The snow depth varies

**RC2.22:** Figure 3: Please make sure to print these figures horizontally then much easier to directly read and probably also to compare the 3 parts on one page. What is the snow temperature in the snow free period? probably best to remove..Precipitation is normally shown

upwards not downwards. Soil temperature is actually the active layer temperatures, which would fit better with permafrost temperatures below.

**AC2.22:** We adopted all suggestions by the reviewer. Temperatures above 0°C have been removed (snow free period), precipitation is plotted upwards. We also changed the term soil temperatures to active layer temperatures. A revised figure 3 now shows all data horizontally and on one page. We also provide the time series data on three pages in an additional Appendix H.

**CM2.22:** See revised Figure 3 and new Appendix H. Please note that high resolution figures (extending the entire pages) will be provided with the final version of this paper.

**RC2.23:** Line 208-209: 'Climate records covering a longer period of time (since 1989) are available from the airport in Longyearbyen. . ..' They go much further back more than 100 yrs.

**AC2.23:** We adopted the correction proposed by the referee.

**CM2.23:** We changed the named passage as follows: Climate records covering a longer period of time (since 1989) are available from the airport in Longyearbyen a longer period of time Svalbard since 1898, with the establishment of a permanent weather station in 1911 (Førland et al. 2011).

**RC2.24:** Line 222: 'PT00 temperature sensors'..must be PT100 temperature sensors.

**AC2.24:** We adopted the correction proposed by the referee.

**CM2.24:** We changed the named passage as follows: The PT100 temperature sensors

**RC2.25:** Line 258: argely – largely

**AC2.25:** We adopted the correction proposed by the referee.

**CM2.25:** We changed the named passage as follows: largely

**RC2.26:** Line 259-260: The Q7 sensor was destroyed by reindeer in September 2003 and was not replaced. When was the reindeer fence put up and why did it not work?

**AC2.26:** The reindeer fence was set up with the original instrumentation in 1998. We do not know how the reindeer could still make it into the fence and suspect that they squeezed in between the wooden bars. Since 2017, this fence was newly constructed and improved.

**CM2.26:** We added a sentence: The fence was improved in 2017.

**RC2.27:** Line 265: 'ground Level' – ground level.

**AC2.27:** We adopted the correction proposed by the referee.

**CM2.27:** We changed the named passage as follows: ground level

**RC2.28:** Line 280: (about 85 m; Figure 2), (about 85 m away ?...

**AC2.28:** We adopted the amendment proposed by the referee.

**CM2.28:** We changed the named passage as follows: (about 85 m away; Figure 2)

**RC2.29:** Line 284-285:'To obtain the snow depth the distance of the sensor from the ground surface was recorded annually and subtracted from the corrected distance data'. . .What about frost lifting of the measuring pole, is this taken into consideration ?

**AC2.29:** We now report an error of about 2.5 cm due to limits of measurement accuracies. This includes lifting of the measuring pole (not likely), and subsiding of the subsurface (due to volumetric decrease of soil upon thawing) which we discuss in the paper.

**CM2.29:** Table 2 includes the revised error of 2.5 for the ultrasonic snow/surface measurement.

**RC2.30:** Line 296-298: 'It should be noted that two of the sensors (an SR50 sensor and the laser sensor) were located within 5 m of each other, while the second SR50 (at the eddy covariance site) was located 85 m away, downhill from the Bayelva site (Figure 2)' This is repeating already provided information.

**AC2.30:** We agree and deleted this sentence.

**CM2.30:**

**RC2.31:** Line 303: to monitor the timing and pattern of snow melt. . .must be for all snow cover changes not only melt

**AC2.31:** We agree and changed the sentence.

**CM2.31:** We changed the named passage as follows: to monitor the timing and pattern of snow  cover changes.

**RC2.32:** Figure 4. 'Bayelva soil temperature trends for three depths from 1998- September 2017 using Level 2 soil temperatures'. So this is active layer and top permafrost temperatures, would be good to use this terminology. 'a) yearly trends at all three depths of 4, 58, 138 cms are 0.18_C/year (standard error of trends: ± 0.07, 0.06, 0.05 _C/570 year) b) winter trends (months December, January, February) are 0.25, 0.25, 0.23_C/winter (standard error of trends: ± 0.12, 0.11, 0.07 _C/winter) and c) summer trends (months June, July, August) are 0.08, 0.1, 0.12_C/summer (standard error of trend ± 0.05, 0.04, 0.03 _C/summer). Years in which data gaps of more than 48 hours exceeded 5% of data were included. Data gaps were interpolated linearly'. Suggestion for changes: a) the mean 20 year temperature increase in the active layer at 4 and 58 cm depths and in the top permafrost at 138 cm was 0.18_C/year, the mean 20 year winter temperature increases were. . ..., while the mean 20 year summer increases were. . ..

**AC2.32:** We adopted the amendments proposed by the referee.

**CM2.32:** We changed the text of Figure 4 as follows:

**Figure 4**. Bayelva mean 20-year temperature increases for three depths from September 1998 – September 2017 using Level 2 soil temperatures. a) in the active layer (4 and 58 cm) and in the top of permafrost (138 cm): 0.18 ±0.07 °C/year, b) in winter (December, January, February): 0.25 ±0.12, 0.25 ±0.11, and 0.23 ±0.07°C/year, respectively, and c) in summer (months June, July, August): 0.08 ±0.05, 0.10 ±0.04, and 0.12 ±0.03°C/year, respectively. Years in which data gaps of more than 48 hours exceeded 5% of data were included; data gaps were interpolated linearly.

**RC2.33:** Line 341: 'stockpiled separate'- stockpiled separately

**AC2.33:** We adopted the correction proposed by the referee.

**CM2.33:** We changed the named passage as follows: stockpiled separately

**RC2.34:** Line 378: 'After installation, the sensors cannot be re-calibrated' – why not? Could be dug up for recalibration..

**AC2.34:** It is not practical to dig up the sensors for recalibration, certainly not without a lasting physical disturbance of the profile. These sensors were inserted into the undisturbed side wall of the soil pit.

**CM2.34:** We added the following information: After installation, the sensors cannot be re-calibrated, certainly not without a lasting physical disturbance of the profile.

**RC2.35:** Line 383-384: 'Subsequently, temperature readings from nine of the 32 sensors subsequently started to drift and' please not more than one subsequently..

**AC2.35:** We adopted the correction proposed by the referee.

**CM2.35:** We changed the named passage as follows: Subsequently, temperature readings from nine of the 32 sensors  started to drift

**RC2.36:** Line 396-397: 'increased noise Levels' increased noise levels..

**AC2.36:** We adopted the correction proposed by the referee.

**CM2.36:** We changed the named passage as follows: increased noise levels

**RC2.37:** Line 407: 'the permanently frozen soil' - the permafrost

**AC2.37:** We adopted the amendment proposed by the referee.

**CM2.37:** We changed the named passage as follows: into the permafrost

**RC2.38:** Figure 4 interpretation. . .you list means for the entire period, but very interesting there was a large increase before 2002 and then more stable conditions since then..would be good to comment on this

**AC2.38:** See also comment AC2.48 below. Since we cannot go into further analysis in this ESSD paper, we will not provide further analysis and interpretation of data at this point. This is planned for a follow up publication.

**CM2.38:** No change in the manuscript.

**RC2.39:** Line 424: were installed were installed..only once..

**AC2.39:** We adopted the correction proposed by the referee.

**CM2.39:** We changed the named passage as follows: probes  were installed in

**RC2.40:** Line 451: The lack of calibration

**AC2.40:** We adopted the correction proposed by the referee.

**CM2.40:** We changed the named passage as follows: The lack of calibration

**RC2.41:** Line 467-468: Ten thermometers were installed, one above the ground surface and nine from the surface down to 9 m depth..you mean a thermistor string !

**AC2.41:** We adopted the amendment proposed by the referee.

**CM2.41:** We changed the sentence as follows: A thermistor string was installed, with one sensor above the ground surface and nine from the surface down to 9 m depth.

**RC2.42:** Line 488: data Levels – data levels

**AC2.42:** We adopted the correction proposed by the referee.

**CM2.42:** We changed the named passage as follows: data levels

**RC2.43:** Line 502: soil temperature – sediment temperature..no information given on soil development, so sediment would be the most natural to use in this connection.

**AC2.43:** We supply information on important soil characteristics, such as stratigraphy, texture, bulk density, carbon content, porosity, cryoturbation, vegetation in Table 1, as well as in the appendices C and D. We would like to keep the term "soil" since it is distinguishable from the initial source sediment (glacial-fluvial) as a result of additions, losses, transfers, and transformations of energy and matter.

**CM2.43:** No change in the manuscript.

**RC2.44:** Line 508-509: The data from this Bayelva site have been widely used for the development and evaluation of land surface models. Include references.

**AC2.44:** We include the reference by: Ekici et al. (2014, 2015), Chadburn et al. (2015, 2017).

**CM2.44:** We changed the named passage as follows: The data from this Bayelva site have been widely used for the development and evaluation of land surface models (e.g. Ekici et al. 2014, 2015; Chadburn et al. 2015, 2017).

**RC2.45:** Line 519: Permafrost around the Arctic is thawing and warming – permafrost around the Arctic is warning and thawing.

**AC2.45:** We adopted the amendment proposed by the referee.

**CM2.45:** We changed the named passage as follows: Permafrost around the Arctic is warming and thawing

**RC2.46:** Line 521: soil temperature – you need to specify is this is sediment or mixtures on rock and sediment?

**AC2.46:** We have provided a detailed information on the soil (stratigraphy, texture, components, physical properties) in appendix C and F. Please also find reply to comment AC2.43.

**CM2.46:** No change in the manuscript.

**RC2.47:** Line 523: (Figure 4 h). Such a figure does not exist (Fig. 4 a, b and c only)!

**AC2.47:** We fixed the error mentioned by the referee.

**CM2.47:** We changed the named passage as follows: (Figure 3h)

**RC2.48:** Line 523-524: Interannual to sub-decadal variability is evident in the data and results mostly from differences during the winter months. But you do not at all explain these here, which is a pity, so why include this sentence?

**AC2.48:** This paper is a contribution to ESSD where further analysis and interpretation of data is not foreseen. We would like to keep this sentence since this it provides the goals for a follow up analysis of these data.

**CM2.48:** No change in the manuscript.

**RC2.49:** Table 2. Permafrost temperature is Active layer and permafrost temperature.

**AC2.49:** We agree and have adapted the wording as suggested in Table 2, as well in the figures and figure captions.

**CM2.49:** See revised Table 2 and Figure 3.

[revised manuscript text omitted]
 between -–17.0 °C to -3.8 °C in January and from 4.6 to 6.9 °C in July (18-year period: 01 August 1993 to 31 July 2011; (Maturilli et al., 2013)). It also provides about 400 mm of precipitation annually, which falls mostly as snow between September and May. Significant  air temperature rise has been detected since 1960, which has generally been attributed to changes in the radiation budget and atmospheric circulation patterns (Førland et al., 2012; Hanssen-Bauer and Førland, 1998). This rise is also reflected in the permafrost temperatures, as recorded from deep boreholes  in the mountains at Janssonhaugen (Isaksen et al., 2007a; Isaksen et al., 2001; Isaksen et al., 2007b). For the 6-year period from 1999 to 2005 the borehole temperatures have increased: at 25 m by 0.26 °C, at 30 m by 0.19 °C and at 40 m by between 0.07 and 0.09 °C (Romanovsky et al., 2017). Continuous permafrost underlies the un-glaciated coastal areas to a depth of about 100 m and the active layer thickness (ALT) at the end of summer ranges between 1 and 2 m (Humlum, 2005). ALT data are available from the Global Terrestrial Network for Permafrost (GTNP) database (http://gtnpdatabase.org/) for several locations on Svalbard, but for different time periods and distances between 25 and 115 km away from Bayelva. The site with the longest record, Janssonhaugen, is located 115 km to the southeast and had an average thaw depth of 175 cm for 1998–2013. The region has experienced increases in cloudiness, precipitation, and the number and intensity of cyclones in recent years, especially during the winter months (Hanssen-Bauer and Førland, 1998; Sepp and Jaagus, 2011). The increase in cloudiness (Maturilli and Kayser, 2016) has led to an increase in incoming long-wavelength radiation,

resulting in a major change to the winter radiation budget for this region, as measured at the German-French (AWIPEV) research station, which is located within the village of Ny-Ålesund (Maturilli et al., 2014). This research station carries out long term monitoring of radiation (Baseline Surface Radiation Network: http://bsrn.awi.de/) and meteorological data (http://www.awipev.eu/).

The data presented herein were collected from the High Arctic Bayelva River catchment area (Figure 1), away from Ny-Ålesund. A legacy from past mining activities is the physical disturbance of the ground in and around Ny-Ålesund (for example, compaction and reworking of the soil). Traffic in the village (people, cars, snow mobiles) also affects the surface conditions, especially in winter. The Bayelva catchment area lies between two mountains (Zeppelinfjellet and Scheteligfjellet), with the glacial Bayelva River originating from the two branches of the Brøggerbreen glacier. The terrain flattens out to the north of the Bayelva site and the Bayelva River flows into the Kongsfjorden fjord and the Arctic Ocean about 1 km from the site (Figure 1). Over the past three decades the Bayelva catchment area has been the focus of intensive investigations into fluvial hydrology (sediment transfer and geochemistry; (Hodson et al., 2002)), soil and permafrost conditions (Boike et al., 2008b; Roth and Boike, 2001; Westermann, 2010; Westermann et al., 2011b), the surface energy balance (Boike et al., 2003b; Westermann et al., 2009), and the micrometeorological processes controlling the surface gas and energy exchanges (Lloyd, 2001b; Lloyd et al., 2001; Lüers et al., 2014). A permafrost maximum, minimum and average temperature with depth (trumpet curve) was shown for data for one year (August 2009–August 2010) by Boike et al., (2012). The data from this Bayelva site have also been used in earth system modeling (Ekici et al., 2014; Ekici et al., 2015). Nearby investigations by Japanese and Italian researchers include vegetation analysis with respect to periglacial and glacial landforms and topography (Cannone et al.,

150 2004; Ohtsuka et al., 2006), and investigations into the plant and plot scale dependence of $CO_2$ emissions on biotic and abiotic factors at the start and end of the growing season (Cannone et al., 2016; Uchida et al., 2006).

The Bayelva site is located on top of the Leirhaugen hill (25 m a.s.l.), on permafrost patterned ground (Figure 2). The hill consists mainly of rock but is partly covered by a mixture of

155 sediments that consist of glacial till, together with fine grained glacio-fluvial sediments and clays from the last glacial advance, likely deposited by the Kongsfjorden glacier (J. Tolgensbakk, personal information). According to the geological map, the site is part of the Brøggerbreen Formation (lithography sand-stone, shale, conglomerate), but the gray color of the sediments suggests that the material was deposited by the Kongsfjorden glacier and not

160 the adjacent Brøggerbreen glacier, which deposits redder material. Thus, the site is part of the Kongsfjorden Formation, with the main lithography sandstone, shale, coal (Norwegian Polar Institute online geological maps, accessed December 2017 ).  
[revised manuscript text omitted]

A 9.3 m borehole was drilled using a rotary drill rig on March 30, 2009 and cased with PVC. No stratigraphic information or drill core was recovered. Ten thermometers were A thermistor string was installed, with one sensor above the ground surface and nine from the surface down to 9 m depth (at 0, 0.5, 1.0, 1.5, 2.5, 3.5, 5.5, 7.5, 9 m below ground surface, Appendix C, Figure C10). The casing was left open (not refilled) so that the thermometerthermistor chain could be retrieved or replaced. Because of instrument failure, thermometers were retrieved and replaced several times. From August 2009, a Geoprecision-M datalogger and sensor chain were used. Temperatures were recorded at hourly intervals, with no averaging; no data was recorded between 30 January 2011 and 29 March 2011, due to a low battery voltage. Geoprecision claims an accuracy of +/-0.05 C (at 0°C) and a resolution of 0.01°C, suitable for measurements in the range from -50 to +120°C. However, comparison measurements using an PT100 thermometer at the same depths in the borehole showed a deviation of up to 0.2°C. In 2014, a wooden shield was installed over the casing to prevent warming due to radiation and to facilitate natural ventilation (Appendix C, fFigure C10). It is recommended that the temperature data from sensors installed above ground and down to a depth of 1.5 m should be used with caution because of air exchange within the casing. Caution is warranted when using

485

490

495

500

505   the borehole temperatures in the uppermost 1–2 m, since they may be affected by air

movement in the borehole. These data are flagged in the data series.

**4   Data quality control and availability**

[revised manuscript text omitted]

565 acknowledge the financial support provided through the European Union's FP7-ENV PAGE21 project under contract number GA282700. We thank the two anonymous reviewers whose comments helped to improve this manuscript.

[Figure]

Figure 1. (a) Location of the Bayelva study site on Spitsbergen, western Svalbard at 78°55'15.6"N 11°49'58.5"E. (b) The site is located about 3 km from Ny-Ålesund in the Bayelva River catchment area, between two mountains (Zeppelinfjellet and Scheteligfjellet), and in front of the Brøggerbreen glacier. Aerial image captured in August 2008 using a high resolution HRSC-AX camera (Hauber et al., 2011a; Hauber et al., 2011b): data and metadata for the high resolution the HRSC-AX image covering the entire area shown in Figure 1b are provided in Appendix B (c) & (d). The area of the site under summer conditions (August 2008) and spring conditions (April 2016).

[Figure]

[Figure]

**Figure 2.** Topographical map (a) and overview of the Bayelva study site (b) in Ny-Ålesund, Svalbard. **(a)** Aerial orthoimage (20 cm/px) with topographic information (2.5 m contour lines) derived from a  Digital Elevation Model (DEM) with a cell size of 0.5 m. The locations of the instrumental and sampling sites are marked by coloured symbols. The site is located on a small hill; white areas are snow fields remaining late in the season. The orthophoto and DEM were obtained with an HRSC-AX camera in August 2008 (data and metadata for the

high resolution DEM covering the entire area shown in Figure 1b are provided in Appendix B). **(b)** Most of the instruments are within a fenced-off area to protect them from reindeer damage. Note that the eddy covariance station and permafrost borehole are located outside the fenced area.

[Figure]

590

**Figure 3.** Time series of Bayelva data provided in this paper . a-f: meteorological data, g-l: soil/subsurface data, m-p: snow data.  The data are organized following the structure of Appendix G. Further details on the sensors and periods of operation are given in Table 2. This plot (divided in three parts for better visibility) is also provided in Appendix H.

[Figure]

**Figure 4.** Bayelva mean 20-year  temperature  increases for three depths from  September 1998 – September 2017 using Level 2 soil temperatures. a)  in the active layer  (4 and  58 cm) and in the top of permafrost ( 138 cm:  0.18 °C/year (standard error of trends: ±0.07 °C/year) b) in winter  (December, January, February):  0.25 ±0.12, 0.25 ±0.11, and 0.23 ±0.07°C/year, respectively, and c) in summer  (months June, July, August):  0.08 ±0.05, 0.10 ±0.04, and 0.12 ±0.03°C/year, respectively.  Years in which data gaps of more than 48 hours exceeded 5% of data were included; data gaps were interpolated linearly.

| Variable | Value | Source |
|---|---|---|
| **Surface characteristic** | | |
| Summer albedo | 0.15 | Westermann et al. (2009) |
| Summer Bowen ratio | 1 (0.25–2) | Westermann et al. (2009) |
| Summer roughness length [mm] | 1 (by fitting an energy balance model)

 7 (eddy covariance) | Westermann (2010) |
| **Snow properties** | | |
| Snow albedo | Average for snow covered period prior to melt: 0.81 (2009–2016) | This paper |
| End of the snow ablation | 8 June–11 July | This paper (1999–2016) |
| Maximum snow depth (end of season before ablation) [m] | range: 0.65–1.42 m

 average 0.9 | This paper (1999–2016) |
| End of season snow density [kg m$^{-3}$] | $350 \pm 50$ | Westermann et al. (2009)

 (Boike et al., 2003a)Boike et al. (2003)

 Appendix E for individual profiles |
| Snow thermal conductivity [W m$^{-1}$ K$^{-1}$] | 0.34 | Computed from snow density of 370 kg m$^{-3}$ using Yen (1981) |
| Snow thermal conductivity [W m$^{-1}$ K$^{-1}$] | $0.45 \pm 0.15$ | Westermann et al. (2009) |
| **Soil properties** | | |
| Organic layer thickness | 0–5 cm (bare to vegetated areas; up to 15 cm in wetter areas) | This paper |
| Thawed soil thermal conductivity [W m$^{-1}$ K$^{-1}$] | $1.3 \pm 0.4$ at volumetric soil liquid water content 20%–40%

 (soil profile 1998: profile average) | Westermann et al. (2009) |

[revised manuscript text omitted]

Norwegian Polar Institute: Svalbardkartet Geology Map 207, http://svalbardkartet.npolar.no/html5/index.html?viewer=svalbardkartet.html5, last access: 22 December 2017, 2016.

Ohtsuka, T., Adachi, M., Uchida, M., and Nakatsubo, T.: Relationships between vegetation types and soil properties along a topographical gradient on the northern coast of the Brøgger Peninsula, Svalbard, Polar Biosciene, 19, 63-72, 2006.

Overduin, P. P. and Kane, D. L.: Frost Boils and Soil Ice Content: Field Observations, Permafr. Periglac. Proc., 17, 291–307, 2006.

Romanovsky, V., Isaksen, K., Drozdov, D., Anisimov, O., Instanes, A., Leibman, M., McGuire, A. D., Shiklomanov, N., Smith, S., and Walker, D.: Changing permafrost and its impacts. In: Snow, Water, Ice and Permafrost in the Arctic (SWIPA) 2017, Arctic Monitoring and Assessment Programme (AMAP), Oslo, Norway, 2017.

Romanovsky, V. E., Smith, S. L., and Christiansen, H. H.: Permafrost thermal state in the polar Northern Hemisphere during the international polar year 2007–2009: a synthesis, Permafr. Periglac. Proc., 21, 106-116, 2010.

Roth, K. and Boike, J.: Quantifying the Thermal Dynamics of a Permafrost Site Near Ny-Ålesund, Svalbard, Water Resour. Res., 37, 2901–2914, 2001.

Roth, K., Schulin, R., Flühler, H., and Attinger, W.: Calibration of Time Domain Reflectometry for Water Content Measurement Using a Composite Dielectric Approach, Water Resour. Res., 26, 2267-2273, 1990.

Schneebeli, M., Coleou, C., Touvier, F., and Lesaffre, B.: Measurement of density and wetness in snow using time-domain reflectometry, Ann. Glaciol., 26, 69–72, 1998.

Scholten, F. and Gwinner, G.: Operational Parallel Processing in Digital Photogrammetry - Strategy and Results using Different Multi-Line Cameras, 20th International Congress for Photogrammetry and Remote Sensing, Istanbul (Turkey), 408–413, 2004.

Scholten, F., Gwinner, K., Roatsch, T., Matz, K.-D., Wählisch, M., Giese, B., Oberst, J., Jaumann, R., Neukum, G., and HRSC Co-I-Team: Mars Express HRSC Data Processing - Methods and Operational Aspects, Photogramm. Eng. Remote Sensing, 71, 1143–1152, 2005.

Sepp, M. and Jaagus, J.: Changes in the activity and tracks of Arctic cyclones, Climatic Change, 105, 577-595, 2011.

Steinhart, J. S. and Hart, S. R.: Calibration curves for thermistors, Deep Sea Research and Oceanographic Abstracts, 15, 497-503, 1968.

Stern, L.: Hydraulic and Thermal properties of Permafrost in Svalbard, Analysis and Modelling using GEOtop, Master of Science, Department of Physics and Astronomy, University of Heidelberg, 73 pp., 2017.

Uchida, M., Kishimoto, A., Muraoka, H., Nakatsubo, T., Kanda, H., and Koizumi, H.: Seasonal shift in factors controlling net ecosystem production in a high Arctic terrestrial ecosystem, J. Plant Res., 123, 1-7, 2009.

Uchida, M., Nakatsubo, T., Kanda, H., and Koizumi, H.: Estimations of the annual primary production of the lichen Cetrariella delisei in a glacier foreland in the High Arctic, Ny-Ålesund, Svalbard, Polar Research, 25, 39-49, 2006.

Weismüller, J., Wollschläger, U., Boike, J., Pan, X., Yu, Q., and Roth, K.: Modeling the thermal dynamics of the active layer at two contrasting permafrost sites on Svalbard and on the Tibetan Plateau, The Cryosphere, 5, 741-757, 2011.

Westermann, S.: Sensitivity of Permafrost, Doctor of Natural Sciences PhD thesis, Combined Faculties for the Natural Sciences and for Mathematics, Ruperto-Carola University, Heidelberg, 174 pp., 2010.

Westermann, S., Boike, J., Guglielmin, M., Gisnås, K., and Etzelmüller, B.: Snow melt monitoring near Ny-Ålesund, Svalbard, using Automatic Camera Systems. PANGAEA, 2015.

Westermann, S., Boike, J., Langer, M., Schuler, T. V., and Etzelmüller, B.: Modeling the impact of wintertime rain events on the thermal regime of permafrost, The Cryosphere, 5, 945-959, 2011a.

Westermann, S., Langer, M., and Boike, J.: Spatial and temporal variations of summer surface temperatures of high-arctic tundra on Svalbard -- Implications for MODIS LST based permafrost monitoring, Remote Sensing of Environment, 115, 908–922, 2011b.

Westermann, S., Lüers, J., Langer, M., Piel, K., and Boike, J.: The annual surface energy budget of a high-arctic permafrost site on Svalbard, Norway, The Cryosphere, 3, 245-263, 2009.

1095    Wewel, F., Scholten, F., and Gwinner, K.: High Resolution Stereo Camera (HRSC) – Multispectral 3D-Data Acquisition and Photogrammetric Data Processing, Canadian Journal of Remote Sensing, 26, 466–474, 2000.

Yen, Y.-C.: Review of thermal properties of snow, ice and sea ice, CRREL, Hanover, NH, USA, 34 pp., 1981.

1100    Yoshitake, S., Uchida, M., Ohtsuka, T., Kanda, H., Koizumi, H., and Nakatsubo, T.: Vegetation development and carbon storage on a glacier foreland in the High Arctic, Ny-Ålesund, Svalbard, Polar Science, 5, 391–397, 2011.